# Highly durable crack sensor integrated with silicone rubber cantilever for measuring cardiac contractility

Dong-Su Kim[1,6], Yong Whan Choi[2,6], Arunkumar Shanmugasundaram[1], Yun-Jin Jeong[1], Jongsung Park[1], Nomin-Erdene Oyunbaatar[1], Eung-Sam Kim[3,4], Mansoo Choi[5] & Dong-Weon Lee[1,4]*

To date, numerous biosensing platforms have been developed for assessing drug-induced cardiac toxicity by measuring the change in contractile force of cardiomyocytes. However, these low sensitivity, low-throughput, and time-consuming processes are severely limited in their real-time applications. Here, we propose a cantilever device integrated with a poly-dimethylsiloxane (PDMS)-encapsulated crack sensor to measure cardiac contractility. The crack sensor is chemically bonded to a PDMS thin layer that allows it to be operated very stably in culture media. The reliability of the proposed crack sensor has been improved dramatically compared to no encapsulation layer. The highly sensitive crack sensor continuously measures the cardiac contractility without changing its gauge factor for up to 26 days (>5 million heartbeats), while changes in contractile force induced by drugs are monitored using the crack sensor-integrated cantilever. Finally, experimental results are compared with those obtained via conventional optical methods to verify the feasibility of building a contraction-based drug-toxicity testing system.

[1] MEMS and Nanotechnology Laboratory, School of Mechanical Systems Engineering, Chonnam National University, Gwangju 61186, Republic of Korea. [2] Division of Mechanical Convergence Engineering, College of MICT Convergence Engineering, Silla University, Busan 46958, Republic of Korea. [3] Department of Biological Sciences, Chonnam National University, Gwangju 61186, Republic of Korea. [4] Center for Next-Generation Sensor Research and Development, Chonnam National University, Gwangju 61186, Republic of Korea. [5] Global Frontier Center for Multiscale Energy Systems, Department of Mechanical and Aerospace Engineering, Seoul National University, Seoul 08826, Republic of Korea. [6] These authors contributed equally: Dong-Su Kim, Yong Whan Choi. *email: mems@jnu.ac.kr

Over the years, several in vitro methods have been proposed for assessing the drug-induced cardiac toxicity by measuring the contractile force change of cardiomyocytes[1–7]. Investigating the physiology of the cultured cardiomyocytes by deformation of the sensing platform such as microposts and/or cantilever[8–13] is considered as a promising method owing to its ease of use, low fabrication cost, high sensitivity, and possibility to employ for high-throughput application. Feinberg et al. reported the polydimethylsiloxane (PDMS) thin film for measuring the contraction force of the cardiomyocytes[14]. The deformation of the PDMS thin film owing to the contraction force of the cardiomyocytes was measured by image analysis. Ribeiro et al. quantitatively measured the contractility of a single cardiomyocyte using PDMS micro-post arrays[8]. Although PDMS micro-posts can measure the contractility of individual cells, the deformation of the micro-posts caused by cardiac contractility is very small. Moreover, it is not easy to grow cells on top of microposts. To overcome the limitations of micro-posts cantilever-type sensor structures were proposed and utilized for the same purpose. Recently, we have proposed various types of cantilever device for investigating the drug-induced changes in the cardiomyocytes[11–13]. Although all these investigations have been intensively studied, the optical-based data acquisition techniques are intrinsically data intensive. In addition, it is tough to parallelize measurements in multiwell plate formats as imaging each well is a time-consuming process, and it is not convenient to employ for rapidly analyzing drug-induced cardiotoxicity effects.

Several nonoptical methods also have been developed to measure the physiology of the cardiomyocytes. For instance, Bielawski et al. proposed a magnet-integrated post and giant magnetoresistive (GMR) sensor array to measure the contraction force of the cardiomyocytes[15]. The GMR sensor can measure the large displacement of the post owing to the contraction force of the cardiomyocytes. However, the use of GMR in the high-throughput analysis is limited due to the interference of the adjacent magnetic field. Mannhardt et al. proposed the piezo actuator-based measurement system for high-throughput real-time analysis of the heart rate, contraction, and the relaxation rate of the engineered heart muscle (EHM)[16]. Recently, strain sensors have been used to measure the contractile force of cardiomyocytes, where sensitivity is determined by the gauge factor (GF) of the strain sensor[17,18]. In general, the GF is defined as the ratio of relative change in electrical resistance to the mechanical strain ($GF = (\Delta R/R_0)/\varepsilon$) of the strain sensor[17]. The strain caused by the mechanical deformation of a thin metal film can be converted into an electrical signal, such as resistance, and analyzed using integrated sensors. Lind et al. established cardiac microphysiological devices via multi-material three-dimensional (3D) printing[17]. They fabricated the cantilever structure, strain sensor, groove, and culture well all at once. The cantilever integrated with a carbon black strain sensor can measure the contraction/relaxation of cardiomyocytes in real-time. However, the sensor exhibits low sensitivity due to the low GF of the integrated carbon black strain sensor which is similar to that of the commercial strain sensors. Our previous study also demonstrated the piezoresistive sensor-integrated PDMS cantilever for measuring the contraction force of the cardiomyocytes[18]. However, proposed measurement system showed less sensitivity due to a low GF (<3) of the metal strain sensor. Therefore, developing the highly sensitive sensor to continuously measure the contraction force and detect rare events such as changes in the regularity of the contraction force of drug-treated cardiomyocytes are imperative for the next generation high-throughput drug screening platform.

High-sensitivity sensors based on silicone nanowires, graphene or carbon nanotube composites, and crack sensors, etc. were employed in the field of micro- and nano-electromechanical system[19–21]. Among them, crack-based sensors have received considerable attention owing to its flexibility, durability, and ultrahigh mechanosensitivity. The crack-based sensor was reported in 2014 by Kang et al.[21]. The authors developed the crack-based sensor by depositing platinum (Pt) on a poly(urethane acrylate) (PUA) and applying a 2% strain to the substrate. The developed crack-based sensor exhibits excellent sensitivity in the air with a GF higher than 2000. The potential importance of the crack sensors was demonstrated in various applications requiring ultrahigh displacement sensitivity[21–23]. However, to date, there is no experimental demonstration of the crack sensor for measuring the physiology of cardiomyocytes. The limited success of the crack sensor in the biomedical drug screening application could be due to the continuous exposure of the metal layer to the conductive culture medium, which reduces the long-term durability of the sensor. In addition, the leached metal ions are harmful to the cardiomyocyte's health[24].

With this research background and considering the advantages of the crack sensor, herein, we propose a highly sensitive crack sensor-integrated silicone rubber cantilever for real-time analysis of maturation and drug-induced changes in cardiac contractility. The durability of the crack sensor in the cell culture medium greatly improved by the PDMS-encapsulation layer. The mechanical durability of the crack sensor was substantially improved due to the chemical bonding between the crack sensor and the protection layer. The crack-based sensor made of Pt was chemically bonded with a PDMS thin layer by depositing an adhesion layer ($SiO_2$: 2 nm) on the Pt. The plasma bonding process performed at a low vacuum and room temperature not only does not affect the function of the conductive layer but also avoids contamination of the cantilever surface. The fabricated crack sensor exhibited a high GF of $9 \times 10^6$ at a strain of 1% even after the formation of the encapsulation layer. The durability (26 days, >5 million heartbeats) of the sensor was also confirmed in various solutions, such as DI water and culture media. After various basic experiments, the changes in the contractile force of cardiomyocytes induced by various cardiovascular drugs, namely verapamil, quinidine, and isoproterenol were evaluated using silicone rubber cantilever integrated with the PDMS-encapsulated crack sensor. The experimental findings of the crack sensor were compared with the data obtained from the laser vibrometer, and the results are consistent with each other. The proposed crack-based sensor-integrated silicone rubber cantilever arrays are expected to be applicable in various fields, such as cardiac toxicity tests in the initial stage of the development of drugs, owing to its excellent sensitivity, reversibility, reproducibility, and stability over extended periods in a culture medium.

## Results

**Principle and preliminary crack sensor characteristics**. The proposed cantilever integrated with the PDMS-encapsulated crack sensor consists of a silicone rubber cantilever, a PDMS thin film, and a glass body. Fig. 1a, b shows dimensions of the various layer and a schematic of the crack sensor-integrated silicone rubber cantilever, respectively. The highly sensitive strain sensor based on metal cracks was formed on the silicone rubber cantilever, allowing us to precisely monitor the strain changes caused by the mechanical contraction of cultured cardiomyocytes on the cantilever. Besides, Au patterns formed on the glass substrate were electrically connected to other Pt patterns formed on the silicone rubber via plasma bonding. The use of the glass body with the Au patterns significantly improved the electrical reliability of the fabricated cantilever sensor. Figure 1c shows a schematic of nano-patterns formed on the cantilever surface to align cardiomyocytes along the groove direction and the principle

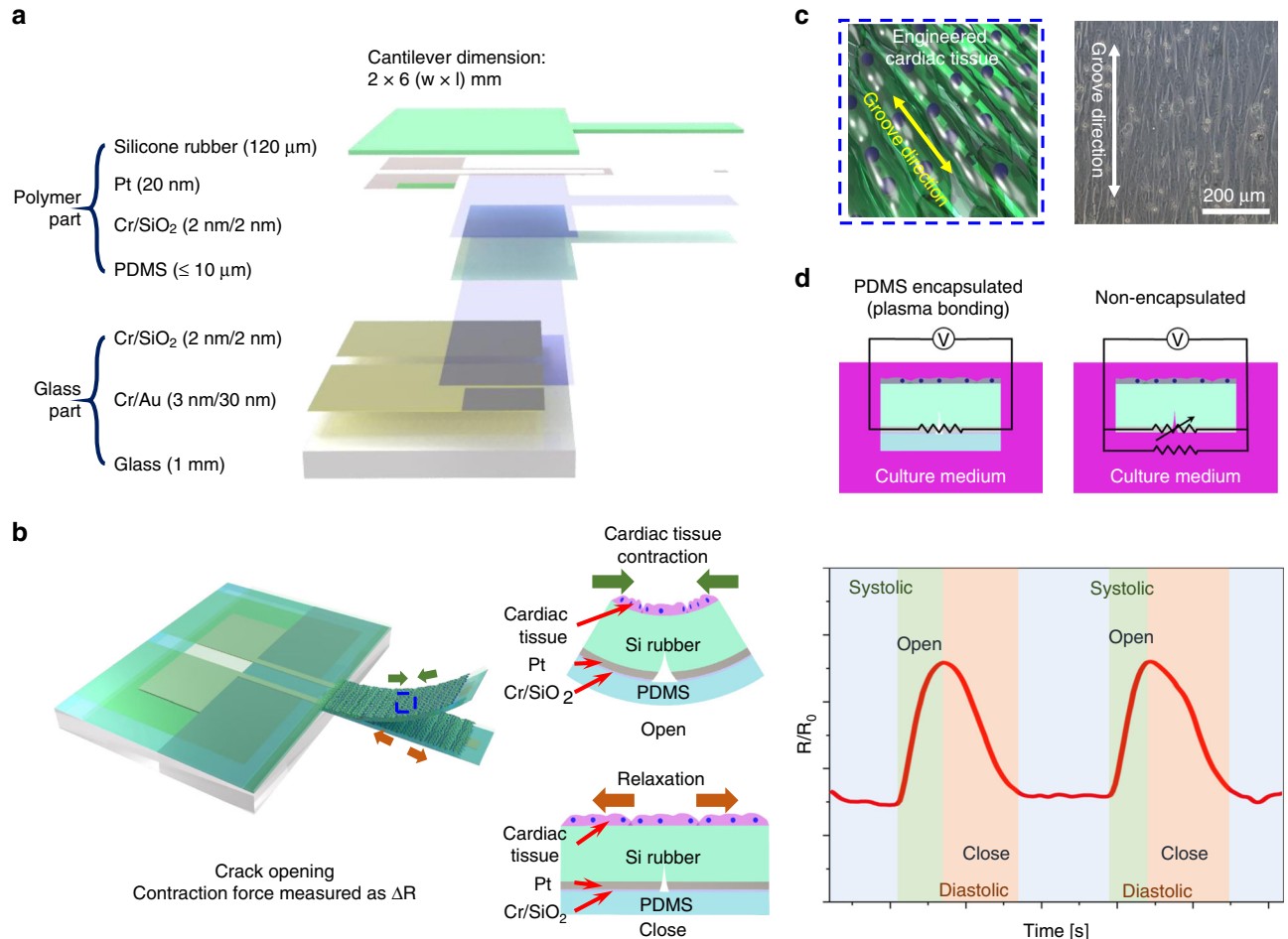

**Fig. 1 Concept of the highly durable crack sensor working in culture media. a** Dimensions of the PDMS-encapsulated crack sensor-integrated silicone rubber cantilever. **b** Schematic of a silicone rubber cantilever composed of various layers and operation principle of the cantilever sensor-integrated with a PDMS-encapsulated crack sensor to measure the contractile force of cardiomyocytes in liquids. **c** Nano-patterns used to align the cardiomyocytes on the cantilever surface. **d** Circuit diagrams of two different crack sensors operating in culture media.

of the cantilever sensor used to measure the contractile force of cardiomyocytes. Figure 1d shows equivalent circuit diagrams of two different crack sensors operating in culture media. The electrical pathway of conventional crack sensors consists of a parallel resistor with cracks and liquid, whereas the current flow in the proposed crack sensor only occurs through cracks generated on the cantilever.

Furthermore, as for the previous PDMS-coating method proposed by Hong et al., the protection layer is physically combined with the crack sensor via spin-coating, which also affects the long-term durability of the crack sensor[25]. However, the proposed crack sensor was chemically bonded to a PDMS thin film via plasma bonding and can be used reliably in various ionic liquids. The use of an intermediate $Cr/SiO_2$ layer on the sensing layer significantly improved the adhesion force between Pt and PDMS. Supplementary Fig. 1 and Supplementary Note 1 briefly explained the sensing mechanism of crack sensor[21–23]. The optical images of the PDMS-encapsulated crack sensor before and after stretching and its corresponding change in resistance ratio as a function of applied strain are shown in Supplementary Figs. 2 and 3 and briefly described in Supplementary Note 2 and 3, respectively.

The stability of the proposed PDMS-encapsulated crack sensor was investigated under various conditions, such as different humidity, temperatures, and culture media. The changes in electrical resistance of the crack sensor with and without PDMS

encapsulation were evaluated at different times. Generally, the metal at the edges of the cracks may be damaged because of the delamination of the metallic thin film from the cantilever surface when used in conditions of repeated tensile strain, temperature changes, and liquid environments. Repeated exposure to harsh environments can result in an intensive stress concentration between cracks owing to different Poisson's ratios and thermal expansion[26]. The PDMS used as the encapsulation layer has high resistance to liquids because of its hydrophobic characteristics. In addition, the PDMS was chemically bonded to the crack sensor via the $O_2$ plasma treatment; it was robust enough to withstand a tensile strength of 2–5 bar[27]. Experiments were conducted in environments with varying humidity to monitor the changes in resistance of the crack sensor due to changes in the dielectric constant of the medium between the cracks. The PDMS-encapsulated crack sensor showed stable behavior in the humidity range of 45–95% compared with the nonencapsulated crack sensor. The PDMS-encapsulation layer was preventing vaporized water molecules from penetrating into cracks. The crack sensor without the PDMS-encapsulation layer allows water molecules to permeate through its cracks, causing irregular cracks opening and closing, results in unstable resistance changes of ±4% (Fig. 2a–c).

Supplementary Fig. 4 and Supplementary Note 4 briefly describes the temperature-dependent deformation of the cantilever composed of the silicone rubber cantilever, silicone rubber,

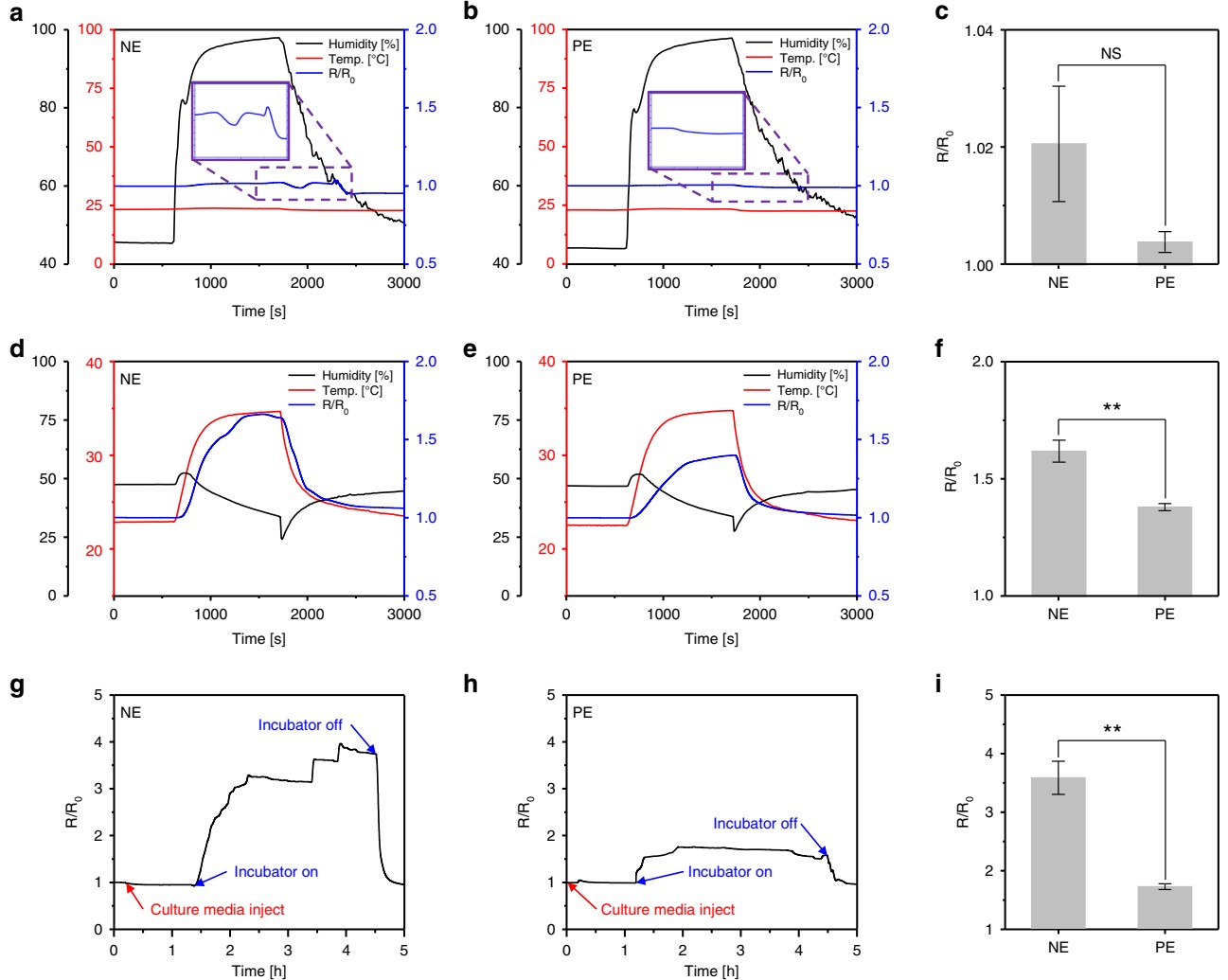

**Fig. 2 Change in resistance of the crack sensors with and without PDMS encapsulation. a–c** Performances of the crack sensors as a function of humidity changes (45–95%). **d–f** Effect of temperature. **g–i** Stability of the PDMS-encapsulated crack sensor in a stage top incubator at 37 °C. (NE and PE represent nonencapsulation and PDMS encapsulation). **c, f, i** Error bars are mean ± s.d. ($n = 5$, $*p < 0.05$, $**p < 0.01$).

and PDMS analyzed using the commercial finite element analysis program (ANSYS). Supplementary Fig. 5 and Supplementary Note 5 describes the displacement change of the cantilever as a function of temperature. In comparison with the simulation results, small differences were negligible, and no further displacement changes of the cantilever were observed when the temperature was kept constant (Supplementary Fig. 5b). Change in the resistance of the PDMS-encapsulated crack sensor at different temperatures and displacement of the PDMS-encapsulated crack sensor-integrated silicone rubber cantilever as a function of time under a custom stage top incubator at 40 °C are shown in Supplementary Fig. 6 and Supplementary Note 6. The temperature characteristics of the crack sensor, according to encapsulation, are shown in Fig. 2d–f. For the nonencapsulated crack sensor, the resistance change rate was ~170% at 35 °C. However, the PDMS-encapsulated crack sensor exhibited relatively stable changes in resistance with a change of ~120% at 35 °C. This can be explained by the fact that the crack sensor was protected from sudden temperature changes by using a polymer with a low thermal conductivity coefficient. Besides, because the proposed crack-based cantilever sensor used for measuring the contractility of cardiomyocytes is to be used in an incubator environment in which temperature and humidity are maintained, changes in the initial resistance can be neglected (Fig. 2g–i). The

nonencapsulated crack sensor initial resistance value increased approximately four times as the temperature increased from room temperature to 38 °C, as required by the stage top incubator. Exposing this crack sensor to electrolyte solutions had a significant impact on its reliability because its resistance value changed rapidly even in the culture media with a constant temperature. In contrast, the PDMS-encapsulated crack sensor saturated with a resistance change similar to that of merely increasing the temperature in air. The hydrophobic characteristics and mechanical sealing effect of the PDMS-encapsulation layer prevent the penetration of the culture media into the cracks even after the operating temperature was increased. Therefore, the crack-based cantilever sensor with the chemically bonded PDMS-encapsulation layer exhibited improved durability in a variety of environments in terms of humidity, temperature, and culture media.

Figure 3a, b shows optical and SEM images of the fabricated crack-based cantilever sensor and a cantilever integrated with the conventional piezo-resistive sensor. A force was applied to the free end of both cantilevers using a motorized stage to reproduce a contraction and relaxation behavior similar to that of cardiomyocytes. Figure 3c, d shows the change in the resistance ratio of the proposed PDMS-encapsulated crack sensor and piezo-resistive sensor as a function of displacement. The crack-based cantilever

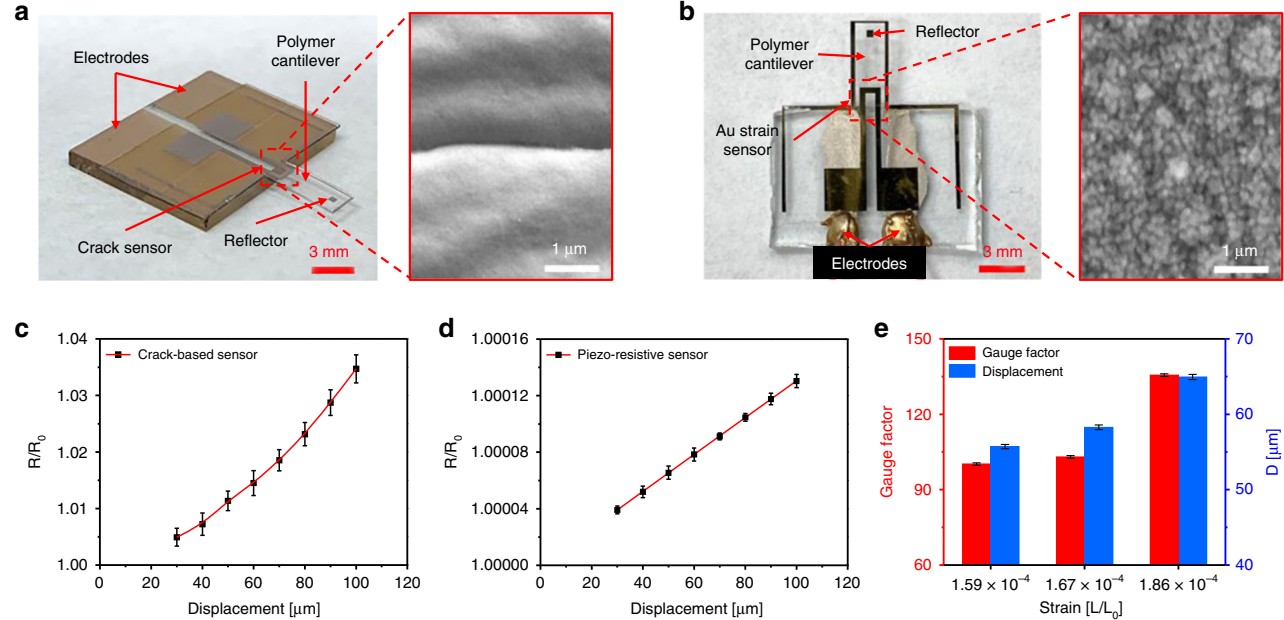

**Fig. 3 Change in resistance of the crack sensor and conventional strain sensor at different displacement. a, b** Optical, and SEM images of the PDMS-encapsulated crack sensor and Au strain sensor integrated on a silicone rubber cantilever. **c, d** Change in sensor resistance ratio of the proposed crack sensor and piezo-resistive sensor as a function of displacement with the ramp of 10 μm. **e** The averaged gauge factor of the crack sensor at the different applied strain. **c–e** Error bars are mean ± s.d. ($n = 5$). D represents displacement.

sensor exhibited a resistance change of ~15 Ω (GF = 166.6) when a displacement of 100 μm (strain = ~$3 \times 10^{-4}$) was applied to the cantilever with a cycle time of 3 s (Supplementary Fig. 7a). On the other hand, the piezo-resistive sensor showed a very low-value resistance change of ~16.7 mΩ (GF = 1.07) at 100 μm (Supplementary Fig. 7b). The proposed crack sensor has ~583 times higher signal-to-noise ratio than the piezo-resistive sensor, allowing accurate analysis of changes in the cardiomyocyte contractile force (Supplementary Fig. 8 and Supplementary Note 7). The fabricated PDMS-encapsulated crack sensor showed ~900-times better sensitivity than the piezo-resistive sensor in the same strain range. Figure 3e shows the experimental results of cardiomyocytes measurements using the PDMS-encapsulated crack sensor in culture media. The GF of the crack sensor is ~156 times higher than the piezo-resistive sensor at 0.03% strain (Supplementary Fig. 9 and Supplementary Note 8). Therefore, the crack sensor is highly sensitive enough to detect even the smaller variation in the displacement of the cantilever caused by the cultured cardiomyocytes contractile force. A 100 μm displacement was applied to the integrated silicone rubber cantilever using a motorized stage and measured the displacement and corresponding change in resistance ratio using a laser vibrometer and crack sensor. The laser vibrometer and crack sensor showed a response time of 121 and 123 ms and remained constant output after the displacement was maintained (Supplementary Fig. 10a, b and Supplementary Note 9). It is confirmed that this method can be used to analyze cardiomyocyte contractility and heart rate. The reproducibility of the proposed crack sensor and its arrays has been verified at the different applied strain in the range of 0–0.7% and 0.03% strain (Supplementary Figs. 11, 12 and Supplementary Notes 10, 11).

**Long-term stability of the PDMS-encapsulated crack sensor.** The long-lasting durability of the crack sensors is critical for in vitro assays. Figure 4a, b shows the change in resistance ratio and corresponding displacements of two PDMS-encapsulated crack sensors with different cardiomyocytes densities. The optical images of the cultured cardiomyocytes and real-time traces of the cantilever owing to the contractility of cardiomyocytes are shown in Supplementary Figs. 13 and 14. As the density of cells increases, the displacement of cantilever increases, and change in resistance increases accordingly. As observed, the measurement of cantilever displacement can be measured stably for 11 days in culture medium even at a displacement of 5 μm or less (Fig. 4a and Supplementary Fig. 15). Figure 4b shows the real-time traces of contraction and relaxation characteristics of cardiomyocytes measured at day 26 using the cantilever with higher cell density. The cantilever displacement owing to the relative contraction force generated by cardiomyocytes increased with an increasing culture period (Fig. 4c, d). The beating rate and rise time of cardiomyocytes significantly decreased at a higher culture period (Fig. 4e, f) indicative of more maturation of cardiomyocytes. Supplementary Fig. 16 shows the changes in resistance over 26 days after culturing the cardiomyocytes on the proposed crack-based cantilever sensors. The beating rate of cardiomyocytes was fast (averaging 4 Hz) at an early stage and then stabilized after 5 days of incubation. Whenever the culture medium was changed every three days after cell seeding, temporary changes in temperature induced a fast-beating rate (5 Hz or more) and an unstable beating for a certain period. This abnormal beating of cardiomyocytes in contraction was normalized by stabilizing the temperature of the culture media. The fabricated crack sensor responded quickly, even at a rapid beating rate of ~6 Hz (systolic 82 ms and diastolic 92 ms). The crack-based cantilever sensor showed a significantly stable output for 26 days even for abnormal beating due to changes in the external environment (Supplementary Fig. 17). Most importantly, no change in GF due to fatigue fracture of the PDMS-encapsulated crack sensor was observed even after 5 million instances of repeated operation. An array of crack sensors was also fabricated and evaluated to validate the possibility of high-efficiency drug toxicity screening capability. Supplementary Fig. 18 and Supplementary Note 12 briefly explains the repeatability and reproducibility of crack sensor arrays under cell culture medium.

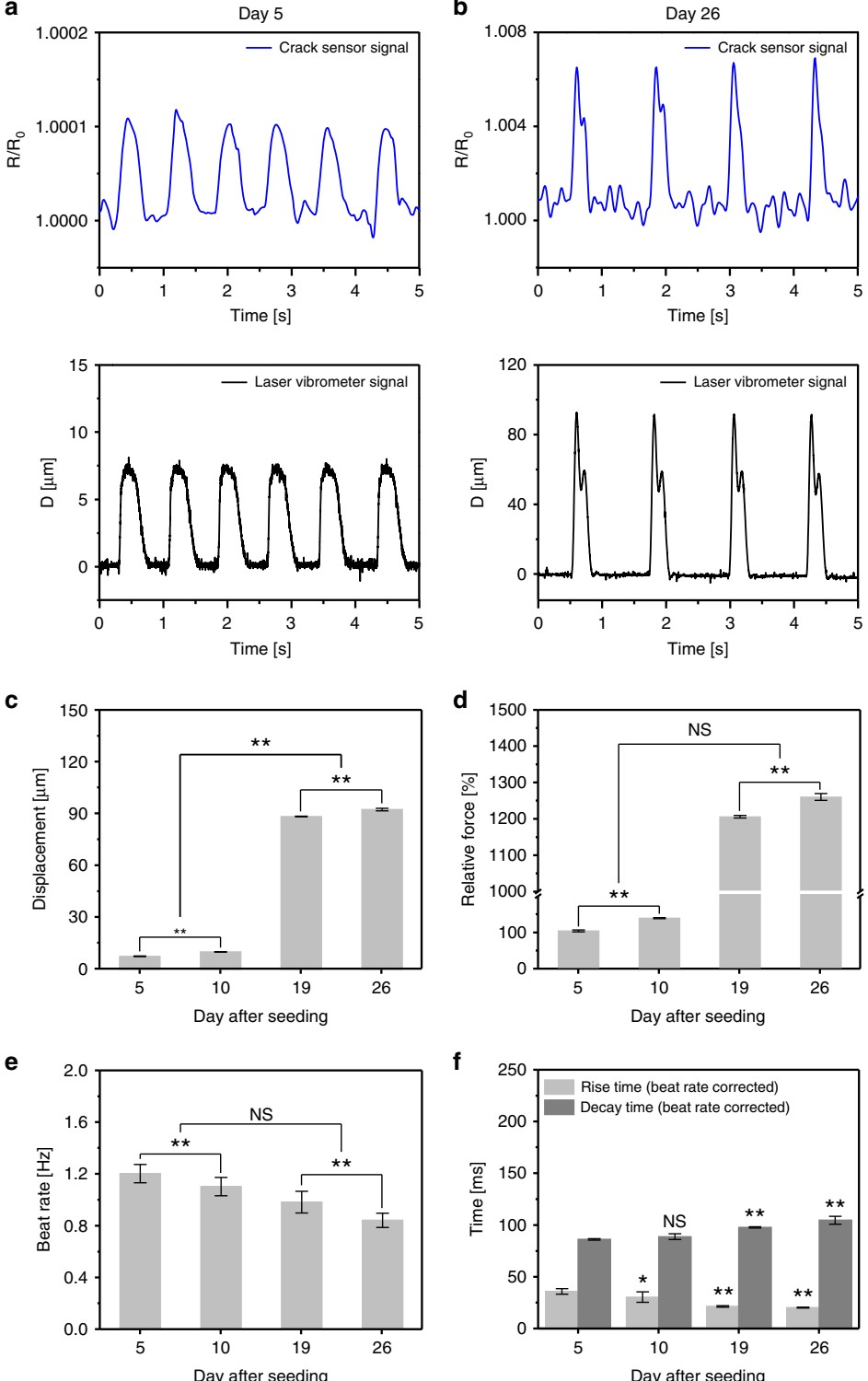

**Fig. 4 Long-term stability of the PDMS-encapsulated crack sensor integrated with silicone rubber cantilever. a, b** Representative traces of real-time change in resistance ratio and displacement of the cardiomyocytes cultured PDMS-encapsulated crack sensor at day 5 and 26 of the culture days. **c** Displacement of the cardiomyocytes seeded PDMS-encapsulated crack sensor integrated with silicone rubber cantilever at different culture periods. **d, e** Relative contraction force, and a beat rate of the cardiomyocytes at different culture periods. **f** Rise time and decay time of cultured cardiomyocytes measured over 26 days. **c–e** (**p < 0.01 measures by two-way ANOVA followed by Tukey's honest significant difference test). The rise time and decay time were analyzed, and beat rate corrected with Fridericia's formula (Rise time corrected (Rc) = R/interspike interval$^{1/3}$) and (Decay time corrected (Dc) = D/interspike interval$^{1/3}$), respectively. (*p < 0.05, **p < 0.01 measures by one-way ANOVA followed by Tukey's honest significant difference test). Error bars are mean ± s.d. (n = 5 biologically independent samples). NS and D represent displacement and nonsignificant, respectively.

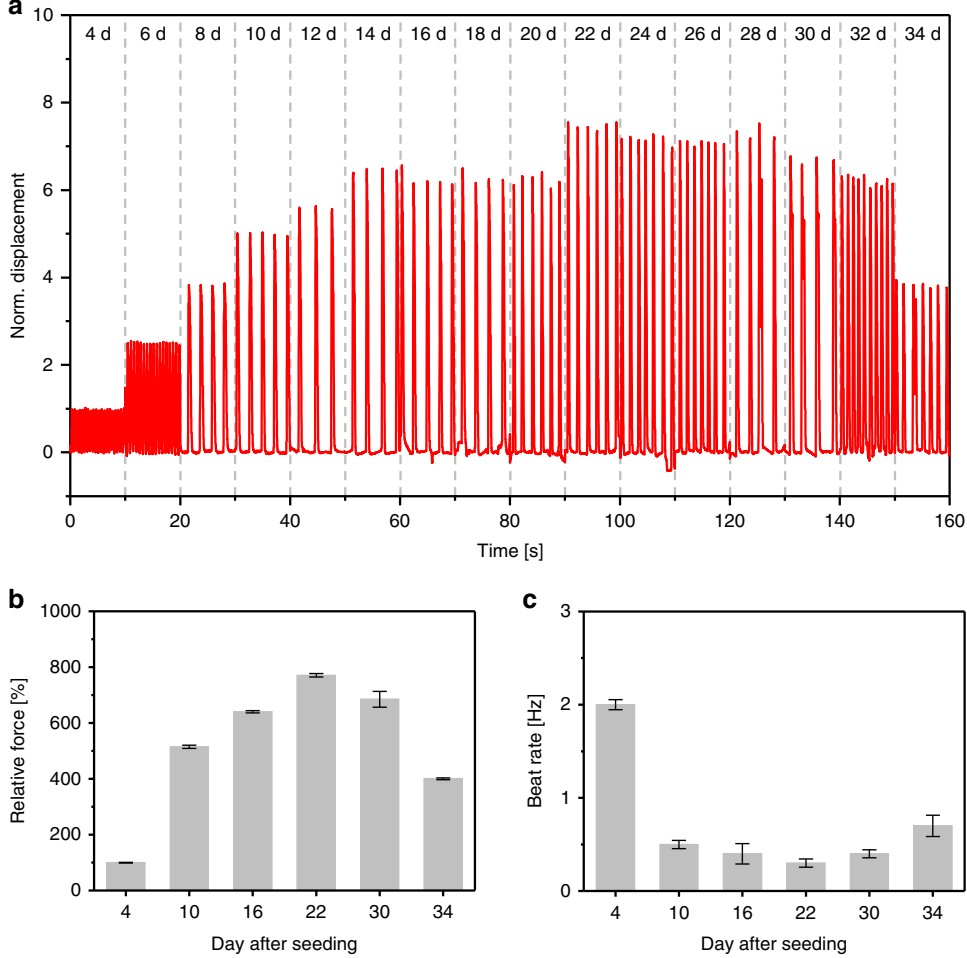

**Fig. 5 Real-time measurement shows the change in contractility of the cultured cardiomyocytes. a** Normalized cantilever displacement owing to the contraction and relaxation of the cultured cardiomyocytes. **b**, **c** Relative contraction force, and beating rate of the cardiomyocytes at different maturation days. Error bars are mean ± s.d., $n = 5$ biologically independent samples.

**Force-frequency analysis of the cultured cardiomyocytes.** Force-frequency relationship (FFR) is one of the significant factors for heart contraction analysis[28]. Hence, FFR of the cardiomyocytes cultured on the PDMS-encapsulated crack sensor was investigated at different external electrical stimulation pacing from 0.5 to 3 Hz. The spontaneous beating of cardiomyocytes was synchronized by the external electrical stimulation. Two carbon electrodes were placed parallel to the longitudinal direction of both sides of the silicone rubber cantilever. External electrical stimulation (square-wave pulses, 2 ms, 3 V/cm) was applied to the cardiomyocytes seeded on the cantilever. The cardiomyocytes were paced at 0.5, 1.0, 1.5, 2, 2.5, and 3 Hz, and the contractility of cardiomyocytes at each frequency was allowed to stabilize for 1 min. The experimental results show that the beating rate of cardiomyocytes can be precisely synchronized with increasing the electrical stimulation (Supplementary Fig. 19a). The contractile force of cardiomyocytes was compared with those at the control state to more closely examine the change in contraction force at various stimulation pacing. The contractility of cardiomyocytes was found to be similar to the control state at 0.5 Hz with no significant difference and then decreased with further increasing the external electrical pacing. The contractility was approximately twofold decreased at 3 Hz compared with the control state (Supplementary Fig. 19b). The decrease in contractile force or negative FFR of culture cardiomyocytes could be attributed to several factors, such as oxygen limitation in the cardiomyocytes,

potential overload due to altered $Ca^{2+}$, and changes in intracellular pH[29,30].

Figure 5a shows the real-time traces of the cantilever displacement measured over 34 days of the culture period. The cultured cardiomyocytes showed the measurable beat rate at an early stage of culture (4th) period. The displacement of the cantilever was increased with increasing culture period and reached highest on day 22 (Fig. 5a). The contraction force of cardiomyocytes was gradually increased with increasing cell culture day and reached maximum value on day 22, then slowly decreased and reached the lowest value on day 34 (Fig. 5b). The decrease in cantilever displacement can be explained based on the following lines. The part of the cultured cardiomyocytes on the silicone rubber cantilever was detached and aggregated with an increasing incubation period, therefore, not transmitting the resultant contractile force efficiently to the sensor. Similarly, the beating rate of the cultured cardiomyocytes was decreased with an increasing culture period, indicative of increased maturity of cultured cardiomyocytes (Fig. 5c).

**Drug-induced cardiac toxicity screening.** After verifying that the cantilever device integrated with the PDMS-encapsulated crack sensor could operate in culture media for a long time, further studies on drugs that affect contractility and beating rate in vitro were conducted based on these preliminary experiments. First,

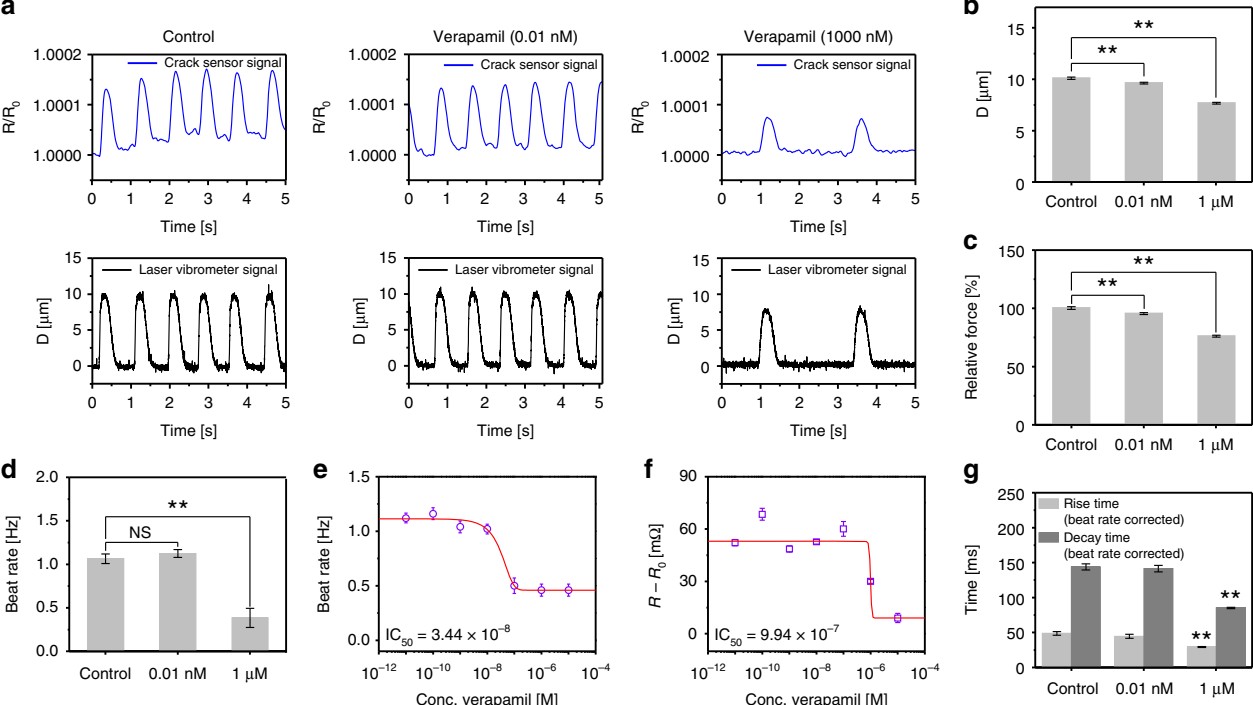

**Fig. 6 Change in contractility of the verapamil treated cultured cardiomyocytes. a** Representative real-time traces of change in sensor resistance ratio and cantilever displacement owing to the contraction and relaxation of different concentration of verapamil treated cardiomyocytes. **b**, **c** Bar plot depicting the cantilever displacement and relative contraction force generated by cardiomyocytes (**p < 0.01 measures by one-way ANOVA followed by Tukey's honest significant difference test.). **d** Beat rate of cardiomyocytes at verapamil concentrations of 0.01 nM and 1μM. **e** Beat rate of verapamil treated cardiomyocytes at different concentrations (0.01 nM–10 μM). **f** Change in resistance of the cardiomyocytes seeded cantilever at different verapamil concentrations (0.01 nM–10 μM). **g** Rise time and decay time of cardiomyocytes at verapamil concentrations of 0.01 nM and 1μM. The rise time and decay time were analyzed and beat rate corrected with Fridericia's formula (Rise time corrected (Rc) = R/interspike interval$^{1/3}$) and (Decay time corrected (Dc) = D/interspike interval$^{1/3}$), respectively. (**p < 0.01 measures by one-way ANOVA followed by Tukey's honest significant difference test.). Error bars are mean ± s.d. ($n$ = 5 biologically independent samples). NS and D represent displacement and nonsignificant, respectively.

cardiac contractility was measured using the integrated crack sensor at different verapamil concentrations. A sudden change in contraction force was observed at a drug concentration of ~1 μM (Fig. 6a and Supplementary Fig. 20). The displacement, relative contraction force generated by cardiomyocytes, and beating rate of cultured cardiomyocytes were gradually decreased with increasing verapamil concentration (Fig. 6b–d). Figure 6e shows the beating rate of verapamil treated cardiomyocytes at different concentrations ranging from 0.01 nM to 1 μM. The cardiomyocytes showed a negative inotropic response to verapamil, with an IC$_{50}$ of $3.44 \times 10^{-8}$ M. Increasing doses of verapamil caused a negative chronotropic effect in cardiomyocytes with an IC$_{50}$ of $9.94 \times 10^{-7}$ M (Fig. 6f), resembling studies based on neonatal rat ventricular myocytes (NRVM) tissue[18]. Furthermore, the increasing dose of verapamil yield a significant increase in the rise time of cardiomyocytes, indicative of the decrease in force generation of cardiomyocytes at higher verapamil concentration (Fig. 6g).

The adverse effects of quinidine on the cultured cardiomyocytes were investigated using an integrated crack sensor (Fig. 7a and Supplementary Fig. 21). The cardiomyocytes contraction force was decreased by increasing the quinidine concentration. furthermore, the early after depolarizations (EAD) in cardiomyocytes was observed when treating the cardiomyocytes at 10 μM quinidine. The relative fraction of EAD is defined as the ratio of the number of EAD beats to the regular beats in one minute. The quantified relative fraction of EAD in cardiomyocytes induced by quinidine (10 μM) was found to be ~0.79 ± 0.11. A decrease in cantilever

displacement with increasing quinidine concentration could be attributed to the decline in relative contraction force produced by cardiomyocytes (Fig. 7b, c). The beating rate of the cardiomyocytes was rapidly decreased at higher concentration (100 μM) of quinidine (Fig. 7d). Furthermore, the quinidine treated cardiomyocytes showed a negative inotropic effect with an IC$_{50}$ of $9.77 \times 10^{-6}$ M (Fig. 7e). Increasing the concentration of quinidine caused the Sigmoidal decrease in the change in resistance of the cantilever with an IC$_{50}$ of $1.42 \times 10^{-5}$ M, signifying negative chronotropic response of cardiomyocytes to quinidine (Fig. 7f). Treating the cardiomyocytes with 1 nM–10 μM of quinidine displayed apparent changes in the rise time of cardiomyocytes. Whereas, the decay time of cardiomyocytes was prolonged at higher quinidine concentration (10 μM) owing to EAD in cardiomyocytes (Fig. 7g).

Isoproterenol is a β-1 agonist of adrenergic receptor and produces a positive cardiac inotropic effect. Supplementary Fig. 22 shows the representative real-time traces of change in resistance ratio and cantilever displacement owing to the contraction and relaxation of different concentrations of isoproterenol treated cardiomyocytes. Upon increasing the concentration of isoproterenol to 1 μM, the beating pattern of cardiomyocytes became irregular, and the tachycardia sequence was observed (Supplementary Fig. 22a). After isoproterenol treatment, contractile force cardiomyocytes increased by 10%, arrhythmia and a side effect, was seen at drug concentrations exceeding 240 nM, which is known as the EC$_{50}$ value. Further, we found an increase in cantilever displacement, relative contraction force generated by cardiomyocytes, and the beating rate of cardiomyocytes in

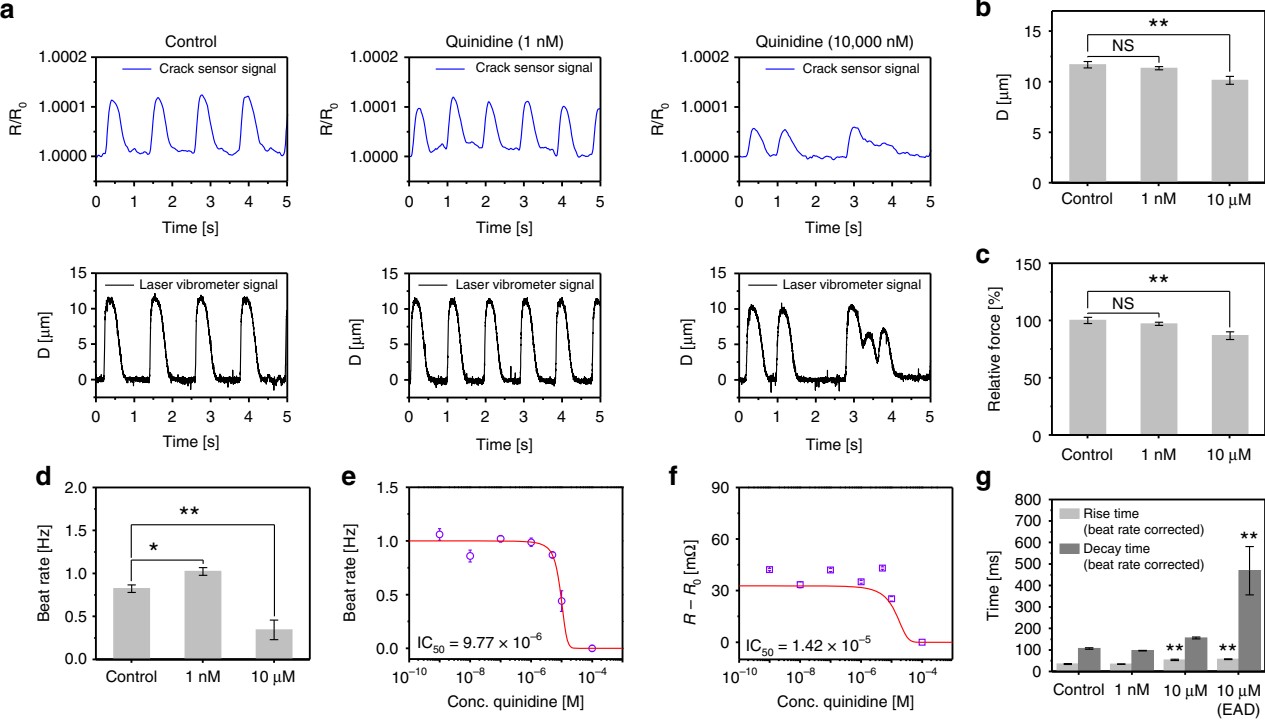

**Fig. 7 Change in contractility of the quinidine treated cultured cardiomyocytes. a** Representative real-time traces of change in sensor resistance ratio and cantilever displacement owing to the contraction and relaxation of different concentration of quinidine treated cardiomyocytes. **b**, **c** Bar plot depicting the cantilever displacement and relative contraction force generated by cardiomyocytes (**p < 0.01 measures by one-way ANOVA followed by Tukey's honest significant difference test.). **d** Beat rate of cardiomyocytes at quinidine concentrations of 1 nM and 10 μM. **e** Beat rate of quinidine treated cardiomyocytes at different concentrations (1 nM–100 μM). **f** Change in resistance of the cardiomyocytes seeded cantilever at different quinidine concentrations (1 nM–100 μM). **g** Rise time and decay time of cardiomyocytes at quinidine concentrations of 1 nM and 10 μM. The rise time and decay time were analyzed and beat rate corrected with Fridericia's formula (Rise time corrected (Rc) = R/interspike interval$^{1/3}$) and (Decay time corrected (Dc) = D/interspike interval$^{1/3}$), respectively. (*p < 0.05, **p < 0.01 measures by one-way ANOVA followed by Tukey's honest significant difference test.). Error bars are mean ± s.d. (n = 5 biologically independent samples). D is representsdisplacement.

1 nM–1 μM range (Supplementary Fig. 22b–d). The cardiomyocytes showed positive inotropic response to isoproterenol with an apparent $EC_{50}$ of $5.52 \times 10^{-7}$ M (Supplementary Fig. 22e). The change in resistance of the isoproterenol treated cardiomyocytes cultured cantilever was showed sigmoidal increase with increasing isoproterenol concentration, with an $EC_{50}$ value of $6.63 \times 10^{-7}$ M (Supplementary Fig. 22f). The rise time of the isoproterenol treated cardiomyocyte was decreased with increasing the concentration of isoproterenol indicative of increase in contraction force generation of cardiomyocytes (Supplementary Fig. 22g). The rise and decay time of the contractility of the isoproterenol treated cardiomyocytes is beat rate corrected with Fridericia's formula (Supplementary Fig. 23 and Supplementary Note 13).

In brief, a proposed sensor is well studied to identify the gradual changes in the contractile kinetics of cultured cardiomyocytes that occurred over the culture period. During this culture period, the contractile force of cardiomyocytes increased from day 4 to 22. Finally, we successfully demonstrate that the proposed crack sensor can detect changes in the contractile behavior of the cardiomyocytes due to the adverse effects of cardiac drugs such as verapamil ($Ca^{2+}$ blocker), quinidine ($Na^+$ blocker), and isoproterenol (β-1 agonist). The cultured cardiomyocytes were showed a negative inotropic and chronotropic response to verapamil with an $IC_{50}$ of $9.94 \times 10^{-7}$ M and $3.44 \times 10^{-8}$ M, respectively. Similarly, the cardiac tissues exhibit a negative inotropic and chronotropic response to quinidine with an $IC_{50}$ of $1.42 \times 10^{-5}$ M and $9.77 \times 10^{-6}$ M, respectively.

Furthermore, the device also detects EAD in quinidine treated cardiomyocytes (≥10 μM). The potential extension of the QT interval can be verified by mechanical contraction force measurement (Fig. 7). The cardiomyocytes showed a positive inotropic ($6.63 \times 10^{-7}$ M) and chronotropic response ($5.52 \times 10^{-7}$ M) to isoproterenol. Quantitative assessment of contractility measurement allows for physiological analysis of inotropic and chronotropic effects in drug-induced cardiomyocytes. The drug toxicity screening ability of the crack-based cantilever sensor makes it possible to solve the drawbacks of existing optical-based measurement systems and enables accurate data collection through the simple measurement platform. Also, by arranging multiple cantilevers integrated with crack sensors in parallel, we expect to be able to simultaneously analyze various drugs with high efficiency.

The real-time drug screening ability of the proposed silicone rubber cantilever integrated with a PDMS-encapsulated crack sensor was analyzed to detect rare events such as changes in the regularity of contraction force of drug-treated cardiomyocytes. Figure 8a shows the real-time traces of cell contractility and beat rate of cardiomyocytes treated with 1 μM verapamil. The cantilever displacement was measured at a regular interval of time after treating cultured cardiomyocytes with a concentration of 1 μM verapamil (Fig. 8b). After 40 min of drug treatment, the displacement of the cantilever was decreased ~45.63%, from $2.98 \pm 0.105$ μm (control state) to $1.62 \pm 0.054$ μm. The drug-treated cardiomyocytes were washed out again and measured the displacement of the cantilever. The displacement of the cantilever

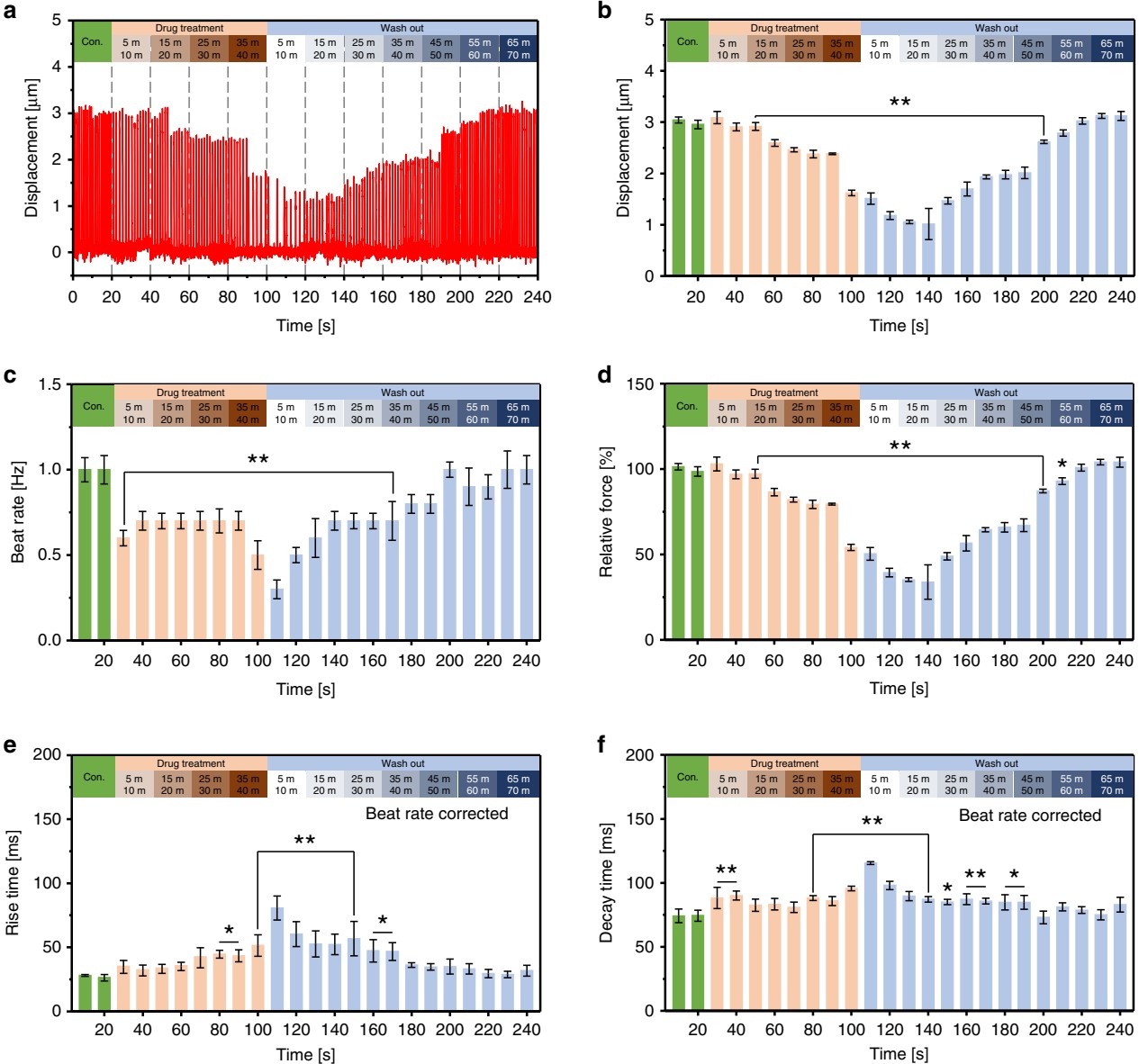

**Fig. 8 Real-time traces show the contractility and beat rate of 1 μM verapamil treated cardiomyocytes. a** Displacement of the cantilever owing to the contraction and relaxation of the 1 μM verapamil treated cardiomyocytes at a different time before and after drug washed out. **b**, **c** Displacement of the cantilever after treating the cardiomyocytes with 1 μM verapamil. **d** Relative contraction force generated by 1 μM verapamil treated cardiomyocytes at different drug treatment time. **e**, **f** Rise time, and decay time of the 1 μM verapamil treated cardiomyocytes. The rise time and decay time were analyzed and beat rate corrected with Fridericia's formula (Rise time corrected (Rc) = R/interspike interval$^{1/3}$) and (Decay time corrected (Dc) = D/interspike interval$^{1/3}$), respectively. (*$p < 0.05$, **$p < 0.01$ measures by one-way ANOVA followed by Tukey's honest significant difference test. Error bars are mean ± s.d.). ($n = 5$ biologically independent samples).

was gradually increased as cardiomyocytes slowly retain its inherent properties. The displacement of the cantilever after 50 and 55 min of drug washed out time was 2.61 ± 0.032 μm and 2.78 ± 0.059 μm, respectively. After 70 min the cantilever displacement was close to the control state as the cardiomyocytes completely refreshed.

The effect of verapamil on the beating rate of cardiomyocytes was investigated at a regular time interval (Fig. 8c). The results showed a decrease in the beating rate with increasing drug treatment time. The beating rate of 1 μM verapamil treated cardiomyocytes after 40 min of drug treatment time decreased ~50%, from 1 to 0.5 Hz. After 70 min of drug washed out, the cardiomyocytes were restored its original beating rate. The relative contraction force of the cardiomyocytes after 50 min of

drug treatment time decreased ~46% from 107 ± 2.056 nN (control state) to 57.1 ± 1.931 nN (Fig. 8d). The relative contraction force of the cardiomyocytes after 50 min of drug washed out was reached 93% of its original value (98.2 ± 2.102 nN). After 70 min of drug washed out, the relative contraction force of cardiomyocytes was nearly similar to the control state.

The gradual decrease in the contraction force of cardiomyocytes could be due to the slow diffusion of verapamil into the cell wall. The decline in the cardiomyocyte's contraction force could be explained based on the following lines. Verapamil is an L-type Ca$^{2+}$ channel blocker and prevents calcium influence to prolong atrioventricular node effective refractory period, thus reducing the contraction force of cardiomyocytes. The beat rate corrected rise and decay time of the verapamil treated cardiomyocytes is

shown in Supplementary Fig. 24. Furthermore, increasing drug treatment time yielded a significant increase in the rise time of cardiomyocytes indicative of a decrease in contractile force generation of cardiomyocytes (Fig. 8e). However, no substantial change in the decay time of cardiomyocytes was observed before and after the drug washed out (Fig. 8f). The long-term measurement analysis demonstrates that the proposed crack-based sensor not only measures the displacement of the cantilever but also detects various unusual contractility of cardiomyocytes which was attributed to the adverse effects of verapamil.

### Effects of a topographic pattern on cardiomyocytes growth.

The cardiomyocytes cultured on the nano-grooved substrates showed a high degree of alignment and a characteristically anisotropic cell structure. The maturation of cardiomyocytes is dependent on the topographic size of the cultured substrate and has been reported to be the most efficient when the groove size is ~800 nm[31]. The sarcomere length of the cardiomyocytes grown on the nano-patterned cantilever ($2.07 \pm 0.102$ μm) was higher than that of cardiomyocytes cultured on a flat cantilever ($1.94 \pm 0.077$ μm) (Supplementary Fig. 25a–e). These results indicate that morphological changes in the culture medium enhanced the organization of the cytoskeleton. The improved organization of the cytoskeleton increased cantilever displacement by a factor of ~2.2, which could result in an additional increase in sensitivity in the proposed crack-based cantilever sensor (Supplementary Fig. 25f, g). The improved performance of the proposed sensing platform was demonstrated using the traditional gene expression and protein localization analysis (Supplementary Fig. 26). The proposed PDMS-encapsulated crack sensor was also applied for flexible electronics applications, as shown in Supplementary Fig. 27.

### Discussion

The capabilities of the crack-based cantilever sensor proposed were experimentally verified; it can be used for a long time in a solution owing to the PDMS protection layer while also maintaining a high sensitivity. In particular, the chemical bonding of the proposed encapsulation layer, performed to maintain long-term stability in the same environment as the culture medium, showed that it is a very stable encapsulation method that does not affect sensitivity after bonding. Characteristic evaluation of the PDMS-encapsulated sensor in various environments (humidity, temperature, and culture medium) showed the advantage of this crack sensor. However, the PDMS-encapsulated crack sensor is not completely impervious to the environmental change. Further research is needed to improve the durability of the PDMS-encapsulated crack sensor-integrated silicone rubber cantilever. The proposed sensor can continuously monitor the FFR over the time-course of culture, which would be a useful way to assess future approaches of maturing cardiomyocytes in vitro. It was possible to stably measure the contractile forces of cardiomyocytes for ~4 weeks in vitro. During the cultivation period, a part of the cardiomyocytes was detached from the on the silicone rubber cantilever and not transmitting the resultant contractile force efficiently to the sensor. Therefore, the stable and long-term measurement of contractile force should be improved by reforming the cantilever design. The dose-response studies for verapamil, quinidine and isoproterenol indicate that the toxicity of drugs can be screened in real-time using the proposed crack-based cantilever sensor with a PDMS-encapsulation layer. The alignment of the cardiac tissue on the cantilever's longitudinal direction can induce a more significant displacement through the concentration of its contraction force. This can be expected to increase further the GF of the proposed crack sensor, which exhibits exponential behaviors in its resistance changes.

## Methods

**NRVM isolation and cell culture.** All animal experiments were performed following protocols approved by the Animal Ethics Committee at Chonnam National University with the Principles of Laboratory Animal Care of national laws (license number: CNU IACIC-YB-R-2015-1). The neonatal rat ventricular myocytes was isolated from the heart from Sprague-Dawley rats within days 1–3. The separated ventricles were washed by using 1 × ADS buffer solution (NaCl 120 mM, HEPES 20 mM, NaH₂PO₄ 8 mM, D-glucose 6 mM, KCl 5 mM, MgSO₄ 0.8 mM, DI water 1 L, pH 7.35). Single cardiomyocytes were acquired through enzyme solution (collagenase 0.5 mg/ml, pancreatin 0.6 mg/ml, 1 × ADS buffer solution 50 ml) and pre-plating. To effectively coat fibronectin (Corning®), the fabricated cantilever integrated with the PDMS-encapsulated crack sensor was exposed to an oxygen plasma system (FEMTO SCIENCE, 80 W, 30 s) for surface treatment. The acquired cardiomyocytes were then seeded on the crack-based cantilever sensor with a density of 1000 cells/mm². Finally, the cardiomyocytes seeded on the cantilever were cultivated at 37 °C in a 5% CO₂ incubator, and the culture medium was replaced every 72 h.

**Immunocytochemical staining.** The cardiomyocytes cultured on the cantilever were fixed in 3.7% formalin solution for 10 min at room temperature and washed three times using phosphate-buffered saline (PBS Takara). Next, permeabilization was accomplished with 0.1% Triton-X (Sigma-Aldrich) in PBS for 10 min at RT. To prevent the nonspecific binding of the antibodies, the sample was treated at room temperature for 30 min by using 3% bovine serum albumin (3% BSA, Sigma-Aldrich). The primary antibodies, monoclonal α-actinin (Abcam, #ab137346) and vinculin (Sigma-Aldrich, #V9131), were diluted 1:100 in 1% BSA and incubated at RT for 1.5 h. The secondary antibodies were (Alexa-Flour 488 goat anti-mouse IgG conjugate and Alexa-Flour 568 goat anti-rabbit IgG conjugate) diluted 1:200 in the same blocking solution and incubated for 2 h at RT. Finally, DAPI (4′,6-Diamidino-2-phenylindole) (Thermo Fisher Scientific, P36931) for nuclei staining was conducted at 37 °C for 15 min. Immunocytochemical staining analysis was quantitatively performed using ImageJ software with a normalization from the level of the entire protein[10].

**Fabrication process of PDMS-encapsulated crack sensor.** Supplementary Fig. 28 shows the thermal expansion coefficient of silicone rubber and PDMS. Supplementary Fig. 29 shows a schematic of the silicone rubber cantilever integrated with a PDMS-encapsulated crack sensor. The fabrication process was divided into two parts, one for the polymer part and one for the glass part. Silicone rubber with a relatively high Young's modulus (compared with PDMS) was used for the fabrication of the crack sensor because the Pt patterns on the silicone rubber were more stable. As shown in Supplementary Fig. 29a-1, 2 g of silicone rubber compounds (KEG-2000-80A/B, Shin-Etus) were placed on a PUA mold with nano-grooves. A 120-μm-thick feeler gauge (NIKO, Feeler 0.12) was placed on the edge of the PUA substrate, and a polyimide film was applied and cured by applying a pressure of 4 MPa at 130 °C for 30 min using a thermal press. Then, a shadow mask was placed on the backside of the silicone rubber film patterned with nano-grooves, and a Pt thin film with a thickness of 20 nm was deposited using a sputter (Supplementary Fig. 29a-2). Next, irregular cracks were generated in the Pt thin film by stretching the silicone rubber film by ~2% using a laboratory-made stretcher (Supplementary Fig. 30). To increase the bonding strength of the PDMS used as the encapsulation layer, an adhesion layer (Cr/SiO₂: 2 nm/2 nm) was deposited on the cracked Pt layer using a thermal evaporator (Supplementary Fig. 29a-3). The deposited SiO₂ layer was chemically bonded to the encapsulation layer (PDMS) via an oxygen-based atmospheric plasma treatment (CUTE-1MPR, Femto Science Inc.), as shown in Supplementary Fig. 29a-4. The shape of the encapsulated crack sensor was precisely defined using a roll-to-plate (TSUKATANI, BFX) and pinnacle die, as shown in Supplementary Fig. 29a-5. The main other process consisted of the fabrication of a glass body in which Au electrodes were formed to enhance the electrical reliability and stability of the crack sensor (Supplementary Fig. 29(b)). First, a photoresist was patterned onto a glass wafer and Cr/Au (3 nm/30 nm) was deposited using a thermal evaporator. After deposition, electrodes were formed on the glass wafer by removing the photoresist using acetone, as shown in Supplementary Fig. 29b-1. The glass wafer with Au electrodes was diced into a 9 mm × 12 mm shape using a dicing saw (AM Technology, NDS200), as shown in Supplementary Fig. 29b-2. Finally, the silicone rubber cantilever integrated with the crack sensor and the glass body were chemically bonded via an oxygen-based atmospheric plasma treatment (CUTE-1MPR, Femto Science Inc.), as shown in Supplementary Fig. 29c.

**Materials evaluations.** PDMS (Sylgard 184) and silicone rubber (KEG-2000-80) are often used as sensor substrates or structure materials in biotechnology owing to their low Young's modulus, biocompatibility, and easy processability. In this research, to maximize the sensor yield in the metal process and to stably cultivate the cells, the mechanical behavior of the two materials was compared, and the substrate material of the crack sensor was then selected. Their Young's modulus

was measured by taking the average slope of the stress-strain curves obtained using a universal tensile tester (Shimadzu, EZ-L) for the stress measurements of each material. The measured Young's moduli were ~0.6 MPa for PDMS and 4.5 MPa for silicone rubber (Supplementary Fig. 31). Owing to the low hardness (45 A) and the large thermal expansion coefficient (TEC, $9 \times 10^{-6}$) of PDMS, thermal expansion often occurs on the surface of PDMS during metal deposition, which makes stable metal deposition difficult. In particular, it is difficult to form uniform cracks because of delamination.

On the other hand, silicone rubber has a relatively high hardness (80 A) and low TCE ($4.2 \times 10^{-6}$) compared with PDMS, so a relatively stable metal deposition can be performed, thereby forming cracks without wrinkles. The designed silicone rubber cantilever had a width of 2 mm, a length of 6 mm, and a thickness of 100 µm. It had a spring constant of 35.2 mN/m, which is almost seven times higher than that of PDMS cantilevers with the same dimensions. Cantilevers with a low spring constant are easily deformed by the stress differences within the cardiomyocytes in the fluid, making it difficult to measure the displacement accurately. The cantilever made of silicone rubber had a higher spring constant than that made of PDMS, and it could easily measure the contraction force of cardiomyocytes owing to the high sensitivity of the PDMS-encapsulated crack sensor.

A previous report indicated that cell maturity is affected by the degree of the surface energy of the material used for cell culture and that there are differences in maturity of up to 30% depending on the material characteristics[32]. Surface energy can be quantified simply through hydrophobic and hydrophilic experiments, which indicate the wettability of water. PDMS and silicone rubber contains a methyl group and exhibits hydrophobic behavior. In particular, surface modification after $O_2$ plasma treatment is particularly important because it affects cell attachment and ECM (fibronectin) coating very sensitively. Supplementary Fig. 32a shows the water contact angle results before and after the plasma treatment. Both materials show hydrophobicity with a contact angle of 100–110° before the $O_2$ plasma treatment. However, after the $O_2$ plasma treatment, the contact angle decreased to approximately ≤11°, indicating that the surface had been temporally modified (Supplementary Fig. 32b). Supplementary Fig. 32c shows the normalized surface energy of PDMS and silicone rubber. Silicone rubber has a 50% higher surface energy than PDMS and is expected to have benefits for cell adhesion.

After culture cardiomyocytes on the surfaces of PDMS and silicone rubber, maturation and adhesion tests were performed according to the characteristics of the materials. Firstly, immunostaining processes for nuclei, α-actinin, and vinculin were performed to visualize the internal structure of the cardiomyocytes. Immunostaining images were then analyzed using ImageJ (NIH, Bethesda, MD, USA) and quantitative data were expressed as mean ± standard deviation (SD). In the immunocytochemistry staining images of a 250 µm × 250 µm area recorded using a confocal microscope (Leica), more nuclei could be observed in the silicone rubber substrates than the PDMS ones, which reveals the excellent biocompatibility of silicone rubber as cell culture substrates (Supplementary Fig. 33a, b). Cell adhesion was improved by ~300%, as shown in Supplementary Fig. 33c. Sarcomere length is an important variable for assessing the contractility and maturity of cardiomyocytes and is known to be ~1.8–2.0 µm in the adult myocardium[31]. The sarcomere length of the cells incubated on the silicone rubber cantilever was around 1.94 µm (±0.077), which is higher than the sarcomere length (1.79 ± 0.011 µm) of the cells cultured on the PDMS cantilever. These results indicate that silicone rubber promotes the structural organization of the cytoskeleton and elongates muscle fiber tissue and sarcomere length (Supplementary Fig. 33d–f). Within cardiomyocytes, vinculin is known to bind to the cytoskeleton as an adapter protein for integrin and to affect the contraction and relaxation of cardiomyocytes[33–35]. In the obtained immunostaining images of vinculin, the expression of this protein on the silicone rubber cantilever was 60% higher than that on PDMS (Supplementary Fig. 33g–i).

**Reporting summary**. Further information on research design is available in the Nature Research Reporting Summary linked to this article.

## Data availability

All relevant data supporting the findings of this study are available herein and in the Supplementary Information files, or from the corresponding author upon reasonable request. The data underlying Figs. 2c, f, i, 3c–e, 4c–f, 5, 6b–g, 7b–g, and 8, as well as Supplementary Figs. 2b, 5a, 8, 9, 11, 16, 19b, 22b–g, 25e, 26b–f, 28, 32b, c, and 33c, f, i are provided as a Source Data file.

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

## Acknowledgements

This study was supported by the National Research Foundation of Korea (NRF) grant funded by the Korea government (MSIT) (No. 2017R1E1A1A01074550).

## Author contributions

D.S.K designed the research, discussed the results with A.S., E.S.K., M.C. and D.W.L., and contributed to writing the manuscript with D.W.L. and A.S., Y.W.C. prepared and characterized the PDMS-encapsulated crack sensor, N.E.O. participated in cardiomyocytes experiment, Y.J.J. and J.P. assisted the cantilever fabrication.

## Competing interests

The authors declare no competing interests.
