## [Peer Review File · Nature Communications]

Reviewers' Comments:

Reviewer #1:

Remarks to the Author:

This paper proposed the encapsulated crack sensor integrated with Si rubber cantilever for measuring cardiac contractility. By encapsulating the crack sensor, highly sensitive long-term measurement with the minimized environmental effect was achieved. Authors claimed that this encapsulated crack sensor has the potential for high throughput measurement using electrical signals compared to optical measurement.

In my opinion, even though some verified results including in vitro cell experiment, the current form is not suitable for publication due to the lack of ground that supports the authors' claim, and insufficient results. However, if the authors clarify the following issues, it can be reconsidered.

1. The main issue of this study is the performance of the encapsulated crack sensor. First of all, it shows the strong nonlinearity in the crack sensor as shown in figure 2. In the range of 0 to 0.3% strain, the change of resistance ratio ($\Delta R/R$) is 1.7 but it reaches almost 90,000 suddenly in the range of 0 to 1% range. It means the proposed sensor is only sensitive for the large strain. In figure 2(e), I think the hysteresis is a minor problem, but due to the strong nonlinearity, there is a small variation in the initial phase (0 to 10 sec) and then large variation after 10 sec. This large nonlinearity is the critical drawback in the sensor performance. Authors should explain how to resolve this nonlinear performance for measurement of cardiac contractility.
2. By encapsulating the authors could reduce the environmental effect such as humidity. However, there is still a problem for thermal effect. I think this thermal issue comes from the different thermal coefficient between Si rubber and PDMS. Why the cantilevers was not made of the same polymer materials? The similar fabrication process may be applied for the same materials.
3. The authors claimed that the advantage of the proposed crack sensor over the optical measurement is the potential for high throughput measurement. However, I can not agree with it easily. The cracks were generated by elongation of Si rubber and they happen arbitrarily as you know (in page 17, irregular cracks were generated in the Pt thin film by stretching the Si rubber film by approximately 2% using a laboratory-made stretcher, as shown in Fig. 7). So, there might be a large device-to-device variation. Also, the authors showed only a single cantilever case. In my opinion, there may be variations among multiple cantilevers, if a microdevice with multiple cantilevers is designed. I strongly recommend that the authors should carry out similar in vitro cell experiments with a microdevice with multiple cantilevers.
4. Another issue is the noise level of the sensor. It seems to be that there is an unstable baseline signal without mechanical stimulus as shown in figure 5 and figure 6. Moreover, the trend of measurement signal rises even though the optical measurement is stable in figure 5(a) and figure 6(a). The authors should evaluate these noise level and instability whether they are acceptable for the contractility measurement.
5. Authors used the change of resistance (ΔR) and resistance ratio ($\Delta R/R$) arbitrary in the text, so it induces the confusion for reading. I recommend the authors should unify them because they convey a similar meaning. Also, the authors missed the unit for ΔR for some figures (i.e. figure 1, figure S6)
6. In the process flow, description of figure 7 in the main text is not consistent with the figure 7: The text includes figure 7(a-1) to (a-7) but figures 7(a-1) to (a-5) exist.

Reviewer #2:

Remarks to the Author:

Review of NCOMMS-19-01725

Highly durable crack sensor integrated with silicone rubber cantilever for measuring cardiac contractility in culture media"

The authors claim to have developed a method to encapsulate a crack-based pressure/force sensor

with PDMS, thereby allowing the sensor to be used for long-term cell culture to study the biology and pharmacology of contractile cells. The authors have convincingly addressed this claim, and the work is interesting. However, in its current state, the manuscript is more suited for a field-specific journal such as Lab-On-A-Chip, than for a general-interest journal like Nature Communications. In particular: articles involving new technology that are typically suited for this journal rise above development of interesting new devices and go on to use these devices to learn new biology.

By the authors' own admission, crack-based pressure sensors are not inherently novel, although their encapsulation strategy appears to be. However, the encapsulation is not ground-breaking enough on its own. The ability to do non-optical monitoring of force is an important advance, because optically-based data acquisition is time consuming, not amenable to parallelization (all samples have to be done one-at-a-time in a typical setup) and produces massive quantities of data (high speed video). However, development of non-optical force monitoring is also being done by others (see Mannhardt et al. *J Tissue Eng Regen Med* 2019, Lind et al. *Nat Mater* 2017, Bielawski et al. *Tissue Eng. Part C Methods* 2016), so the level of novelty that would be appropriate for a general-interest journal would be in applying such a system to ask new questions, rather than simply developing the system.

As the authors have developed a new way to perform long-term, non-optical monitoring of cardiac tissue contractile forces, they have the opportunity to do some rather interesting studies here – these would merit publication in Nature Communications. For example, several labs have shown recently that various physical manipulations (for example, continuous electrical pacing – see Richardson-Bouchard et al. *Nature* 2018, Lee et al. *Cell* 2019), chemical changes (for example, changes in ATP substrate – see Mills et al. *PNAS* 2017) and growth factors (for example IGF1 and NRG1, see Rupert and Coulombe, *Stem Cells International* 2017) can enhance maturation of pluripotent stem cell derived cardiomyocytes. Similar data have been shown for NRVM (see Godier-Furnemont et al. *Biomaterials* 2015). In all of these studies, however, we get a snapshot of “before” and “after” maturation. Part of why this happens is because the current optical techniques used to measure force and physiology are inherently data intensive, and sometimes are terminal (for example, tissues or micro-tissues that are not grown on posts, e.g. Zimmerman et al. *Biotechnol Bioeng.* 2002, Huebsch et al. *Sci Rep* 2016, require mounting on force-apparatus to get contractile force measurements and perform Frank-Starling analysis). The current technology developed by Kim et al. could give us a continuous-readout of force development. In turn, the time-scales over which maturation occurs (e.g. over which twitch force increases – or better, the time-course over which tissues begin to exhibit a positive Force-Frequency Relationship) could give some clues as to biologic mechanism (e.g. epigenetic changes are likely to involve very long time-scales, whereas changes in protein localization would take hours or less). This would be interesting and novel. Similarly, the ability to monitor continuously to detect rare events (for example, changes in the regularity of contraction that are only occurring over a few minutes every hour) in drug-treated or mutation-harboring cells (in particular, disease-prone iPSC-cardiomyocytes) would also be interesting and novel and merit publication in Nature Communications.

Another general point is that since the novelty appears to be the ability to use the device long-term, for applications in cell characterization, more robust characterization of the cells themselves is needed. For example, more detailed analysis of gene expression and protein localization after the extended culture.

Stylistically, the article needs to be much more comprehensive in referencing the literature related to cardiac tissue engineering. The authors appear to be using a “cells on a cantilever” system similar to the system developed by the lab of Kevin Parker, described in Feinberg et al. *Science* 2007, but have not referenced this. They also discuss maturation of tissue over time, but do not cite any of the literature on this rather extensively studied topic as well.

I would also recommend putting the device design into Figure 1, so that it incorporates more actual data on top of the conceptual material in that figure.

From a stylistic standpoint I would recommend deleting "novel" from the abstract and manuscript.

Finally, in most instances, when the authors are referring to previous work, they should cite the last name of the first author rather than the first name (for example, "Ribeiro et al." rather than "Alexandre et al." on page 2, line 40).

Reviewer #3:

Remarks to the Author:

It is a good manuscript. I like the authors to modify the manuscript based on the following comments.

1. Please check the English of the manuscript by a technical writer.
2. It may be good to use the references in a more better way. For example, "Typical high-sensitivity sensors in the field of microelectromechanical system (MEMS) include silicon nanowires (SiNWs), graphene or carbon nanotube (CNT) composites, crack sensors, etc. [10-21]." The references [10 - 21] is too many.
3. Provide the dimension details of Figure 1 a to make it more clear. It is not clear why the Si layer need to be so thick.
4. Page 6: " The sensing mechanisms of crack sensors have been reported several times as shown in Fig. S1 [13-22]." It is not clear what the authors mean by Fig. S1?
5. It is reported that the resistance of the crack sensor changes with strain. I am curious to know whether the authors have attempted to see whether there is any change of capacitance too?
6. "The electric resistance of the crack sensor increased 1.7 times for a 0.3% strain and increased approximately 9,000 times for a 1% strain". The change of resistance is too large compared to 0.3% to 1%. What is the reason behind this such a large change? Is the sensor still come back to their original structure/condition after the removal of strain? Figure 2 shows the hysteresis characteristics but it is not fully visible.
7. Figure 3 is not clearly visible. The effect of temperature is not much due to using a polymer with a low thermal conductivity coefficient. This means that the change takes place very slowly. It will be good to see the effect of temperature with a long time of operation. This is important as the effect of toxicity is also not expected to act very quickly.
8. It will be good to see more clear pictures for figures 5 and 6.
9. It is not clear why the frequency of opening/closing for the crack sensors initially decreased and then increased again.
10. The effect of change of environment should be noted and some experimental results to be provided.

Manuscript ID: NCOMMS-19-01725

Title: Highly durable crack sensor integrated with silicone rubber cantilever for measuring cardiac contractility in culture media

Author(s): Dong-Su Kim, Yong Whan Choi, Yun-Jin Jeong, Jongsung Park, Nomin-Erdene Oyunbaatar, Eung-Sam Kim, Mansoo Choi and Dong-Weon Lee.

Point-by-Point Response to the Reviewer's Comments

Reviewer #1

Summary Recommendation: This paper proposed the encapsulated crack sensor integrated with Si rubber cantilever for measuring cardiac contractility. By encapsulating the crack sensor, highly sensitive long-term measurement with the minimized environmental effect was achieved. Authors claimed that this encapsulated crack sensor has the potential for high throughput measurement using electrical signals compared to optical measurement. In my opinion, even though some verified results including in vitro cell experiment, the current form is not suitable for publication due to the lack of ground that supports the authors' claim, and insufficient results. However, if the authors clarify the following issues, it can be reconsidered.

Response: We sincerely thank the reviewer for his valuable suggestions and a very positive feedback. We are grateful to the reviewer for his valuable time and energy on our behalf to improve the quality of the manuscript. As recommended by the reviewer, we have carried out the additional experimental analysis to address the concerns raised by the reviewer. In the following section, we addressed all the points and suggestions made by the reviewer.

Comment (1). The main issue of this study is the performance of the encapsulated crack sensor. First of all, it shows the strong nonlinearity in the crack sensor as shown in Fig 2. In the range of 0 to 0.3% strain, the change of resistance ratio ($\Delta R/R$) is 1.7 but it reaches almost 90,000 suddenly in the range of 0 to 1% range. It means the proposed sensor is only sensitive for the large strain. In Fig. 2(e), I think the hysteresis is a minor problem, but due to the strong nonlinearity, there is a small variation in the initial phase (0 to 10 sec) and then large variation after 10 sec. This large nonlinearity is the critical drawback in the sensor performance. Authors should explain how to resolve this nonlinear performance for measurement of cardiac contractility.

Response: As rightly mentioned by the reviewer, the proposed crack sensor exhibits the high sensitivity and non-linear characteristics. Owing to the non-linear characteristics, the crack sensor exhibits the large resistance ratio variation (R / R_0) $\sim 90,000$ at higher strain range (0-1%) and relatively small variation of ~ 1.7 at the lower strain range (0-0.3%). As the reviewer well known, the strain produced by the cultured cardiomyocytes on the cantilever is very small. In this present study, the maximum strain produced by the cultured cardiomyocytes on the cantilever is $\sim 0.03\%$ which causes $\sim 100 \mu\text{m}$ cantilever displacement. In addition, the gauge factor of the crack sensor is ~ 156 times higher than the commercial piezoresistive sensor at 0.03% strain (Figure. S7). Therefore, the crack sensor is highly sensitive enough to detect even the smaller variation in the displacement of the cantilever caused by the contractile force of the cultured cardiomyocytes. Further, we are more focused on the development of a novel crack sensor integrated silicone rubber cantilever to analyze the mechanical properties (contraction force, heart beat) of cardiomyocytes in a culture media for a long time.

Figure. S7. Bar plot representing the gauge factor of the piezo resistive sensor and the crack sensor at 0.03% of applied strain.

Figure. S2. (a) Hysteresis of the PDMS-encapsulated crack sensor integrated silicone rubber cantilever in the tensile range of 0-0.5%, (b) change in resistance of the PDMS-encapsulated crack sensor integrated silicone rubber cantilever as a function of displacement.

As for as the hysteresis concerns, the proposed sensor exhibits the hysteresis only at the higher strain range. At the lower tensile in the range of 0-0.5% the crack sensor shows no hysteresis (Figure. S2). In this present investigation, the maximum strain produced by the cultured cardiomyocytes is $\sim 0.03\%$ which causes $\sim 100 \mu\text{m}$ cantilever displacement. Figure. S2b shows the change in resistance of the crack sensor as a function of displacement. Different displacement with the ramp of $10 \mu\text{m}$ was applied to the crack sensor integrated silicone rubber cantilever using a motorized stage and measured the resistance variation. The resistance of the sensor increases almost linearly with increasing the displacement of the sensor. The obtained experimental data demonstrated that the hysteresis does not appear at low tensile range where the crack sensor operated to measure cell contraction force of the cardiomyocytes.

In addition, the response time of the crack sensor is very similar to that of the conventional optical (laser displacement measurement) method. Therefore, it is easy to analyze the contraction force of the cardiomyocytes changing in real-time. A $100 \mu\text{m}$ displacement was applied to the integrated silicone rubber cantilever using a motorized stage and measured the displacement using laser vibrometer and crack sensor. The laser vibrometer and crack sensor showed the response time of 121 and 123 ms and remained constant output after the displacement was applied (Figure. S8). It is confirmed that this method can be used to analyze cardiomyocyte contractility and heart rate.

Figure. S8. A comparison of the response of a laser vibrometer and a crack sensor.

Comment (2). By encapsulating the authors could reduce the environmental effect such as humidity. However, there is still a problem for thermal effect. I think this thermal issue comes from the different thermal coefficient between Si robber and PDMS. Why the cantilevers was not made of the same polymer materials? The similar fabrication process may be applied for the same materials.

Response: We do agree with the reviewer that the existence of thermal effect due to different thermal coefficient of the polymer materials. As rightly mentioned by the reviewer, we encapsulated the crack sensor with a thin layer of silicone rubber during the initial stage of this proposed work. We have tried to control the thickness of encapsulating silicone rubber layer through various feeler gauges. However, when the applied external pressure exceeding more than 4 MPa during the thin film formation, the PUA-glass plate employed as a groove mold was damaged and the PI film was also deformed. When the applied pressure lower than 4 MPa, the formation of a uniform thin layer was extremely difficult. As a result, we could able to form the 120 μm thick encapsulation layer. The higher thickness encapsulation layer increases the spring constant and decreased the cantilever displacement and sensitivity. Therefore, an ultra-thin encapsulation layer with the thickness $\leq 1 \mu\text{m}$ is highly desired to ensure the contractile force of the cultured cell is sufficiently transferred to the sensor. Further, producing the several μm thick silicone rubber encapsulation layer through a spin coating method is impossible due to high viscosity of silicone rubber (1,120 Pa \cdot s). Consequently, we chose PDMS which has a thermal coefficient (310) comparably similar to the silicone rubber (~ 250). In addition, encapsulation layer with the thickness less than $\leq 1 \mu\text{m}$ is also possible with other polymer materials such as PI or PET. However, compared to PDMS, PI and PET show a

large difference in thermal expansion coefficient compared to silicone rubber (Figure. S20).

Figure. S20. Bar plot represents the thermal expansion coefficient of the silicone rubber, PDMS, PI, PET.

Figure. S3. (a, b) Thermal-expansion analysis of cantilever made by the silicone rubber and PDMS +

silicone rubber at different temperatures.

Further, we have carried out the additional simulation and experimental analysis to show the effect of thermal coefficient on the cantilever displacement. The temperature dependent deformation of the cantilever composed of the silicon rubber cantilever, silicon rubber and PDMS (Silicone + PDMS) was analyzed using the commercial finite element analysis program (ANSYS). As a result of the thermal expansion analysis, the deflection of the cantilever increases as the temperature increases (Figure. S3). The temperature dependent displacement of the cantilever composed of silicon rubber and PDMS was measured using a laser displacement sensor under a home-made stage-top incubator environment.

The bar plot (Figure S4a) shows the cantilevers displacement as a function of temperature. The deformation of the cantilever was found to be increased with increasing the incubator temperature. At 40 °C the simulation studies show the displacement of $\sim 106 \mu\text{m}$, whereas, the experimental analysis shows the displacement of $\sim 66 \mu\text{m}$. However, the cantilever showed no further deformation once the temperature stabilized inside the incubator. Figure. S4b shows the PDMS encapsulated Si rubber cantilever deformation at 40 °C as a function of time. The time dependent studies showed almost stable cantilever deformation and remains constant indicating the reliability of the proposed device.

Figure. S4. (a) the displacement of the cantilever with temperature change, (b) the displacement of the cantilever in a constant temperature environment

Comment (3). The authors claimed that the advantage of the proposed crack sensor over the optical measurement is the potential for high throughput measurement. However, I cannot agree with it easily. The cracks were generated by elongation of Si rubber and they happen arbitrarily as you know (in page 17, irregular cracks were generated in the Pt thin film by stretching the Si rubber film by approximately 2% using a laboratory-made stretcher, as shown in Fig. 7). So, there might be a large device-to-device variation. Also, the authors showed only a single cantilever case. In my opinion, there may be variations among multiple cantilevers, if a microdevice with multiple cantilevers is designed. I strongly recommend that the authors should carry out similar in vitro cell experiments with a microdevice with multiple cantilevers.

Response: We do agree with the reviewer’s opinion that the “large device-to-device performance variation”. The device would show the different performance when the cracks are formed in macro level and random interval during the fabrication process. But in this proposed work the cracks are formed in the nanoscale level and regular interval in a very dense Pt mesh. Further, we have always employed a home-made linear tensioner with an accuracy of $\pm 1 \mu\text{m}$ (Figure. S21) to keep the density of the crack’s constant during the similar fabrication conditions. Therefore, we strongly believe that the sensor should exhibits constant performance when fabricated under the similar fabrication conditions.

Figure S21. Tensile testing machine for crack sensor fabrication

Further, the reproducibility of the crack sensor has been verified by fabricating the five crack sensors under similar manufacturing conditions. Figure S9 shows the change in resistance of the fabricated crack sensors at different applied strain ranging from 0 to 0.7 %. As shown in figure the

sensors fabricated under similar manufacturing conditions showed comparatively uniform characteristics with the maximum change in resistance ratio of 44.17 ± 3.67 , indicating the high-performance reliability of the proposed crack sensors.

Figure S9. Change in resistance and standard deviation of the five different crack sensors at different applied strain in the range of 0 – 0.7%.

As recommended by the reviewer, we have fabricated the crack sensor arrays by integrating three cantilevers in a glass body and analyzed the characteristics of each cantilever by applying 0.03% strain. The initial resistance of the fabricated cantilever array was found to be $\sim 276 \pm 8.2 \Omega$, and the resistance increased to $\sim 288 \pm 8.1 \Omega$ after applying 0.3% of strain. As shown in Figure S10. Fabricated sensors showed almost similar change in resistance trend with the maximum resistance ratio (R/R_0) of 1.0437 ± 0.002 , indicating the high reliable performance of the proposed crack sensors.

Figure S10. (a, b) Optical image of the fabricated multi-cantilever arrays and (b) the change in resistance ratio of the multi-cantilever arrays as a function of different applied strain in the range of 0 to 0.7%,

Figure S16. (a) Photograph of a multi-cantilever device consisting of three cantilevers and (b) change in resistance ratio of the cantilever due to contractile force of cardiomyocytes.

In addition, the performance of the multi-cantilevers array was evaluated after seeding the cultured cardiomyocytes on the crack sensor integrated PDMS encapsulated silicone cantilever. As shown in Figure S16 the resistance ratio of the sensors was found almost similar. The smaller variation in the resistance variation trend between the three sensors could be attributed to the different behavior seeded cardiomyocytes under the cell culture medium. The obtained results demonstrating the high reliability and repeatability of the proposed bio-sensing devices.

Comment (4). Another issue is the noise level of the sensor. It seems to be that there is an unstable basement signal without mechanical stimulus as shown in Figure 5 and Figure 6. Moreover, the trend of measurement signal rises even though the optical measurement is stable in Figure 5(a) and Figure 6(a). The authors should evaluate these noise level and instability whether they are acceptable for the contractility measurement.

Response: The noise level of the crack sensor shows a high noise compared to the optical measuring method using the commercial laser displacement meter. However, as mentioned on page

3 of the text, the optical measurement method using the laser displacement sensor requires a motorized stage for the movement of the laser displacement sensor or multiple laser displacement sensors for the high-throughput screening, it is difficult to analyze various drugs in real time. High-throughput drug screening is possible by employing the PDMS encapsulated crack sensor integrated silicone rubber cantilever. Also, the proposed crack sensor has a 583 times higher signal-to-noise ratio (SNR) than the conventional piezo-resistive sensor, allowing accurate analysis of changes in the cardiomyocyte contractile force (Figure S6). In addition, the proposed crack sensor has a phenomenon that the base line increases or decreases according to the fluctuation of the internal temperature of the stage-top incubator due to temperature-dependent characteristics. This problem can be kept constant by using the active-dummy method used in commercial strain gauges.

Figure S6. Signal to noise (SNR) of resistance change of crack sensor and piezo-resistive sensor by cell contraction force

Comment (5). Authors used the change of resistance (ΔR) and resistance ratio ($\Delta R/R$) arbitrary in the text, so it induces the confusion for reading. I recommend the authors should unify them because they convey a similar meaning. Also, the authors missed the unit for ΔR for some figures (i.e. Figure 1, Figure 6).

Response: We sincerely regret our oversights and it has been rectified in the revised version of the manuscript.

Figure 1. Revised version of the manuscript

Figure 1. (a) Dimensions of the different layer of the proposed crack sensor. (b) Schematic of a Si rubber cantilever composed of various layers and operation principle of the cantilever sensor integrated with a PDMS-encapsulated crack sensor to measure the contractile force of cardiac cells in liquids. (c) Nano-patterns used to align the cardiac cells on the cantilever surface. (d) Circuit diagrams of two different crack sensors operating in culture media.

Figure 4. Revised version of the manuscript

Figure 4. Optical and SEM images of (a) the PDMS-encapsulated crack sensor and (b) an Au strain sensor integrated on a Si rubber cantilever. Resistance changes of (c) the proposed crack sensor and (d) the Au strain sensor as a function of displacement. (e) Gauge factor of the crack sensors depending on strain values (D: Displacement).

Figure 5. Revised version of the manuscript

Figure 5. Resistance changes and corresponding displacements of the PDMS-encapsulated crack sensor measured over 26 days (D: Displacement).

Figure 6. Revised version of the manuscript

Figure 6. Changes in sensor resistance and cantilever displacement as a function of drug concentration for (a) Verapamil and (b) Quinidine (D: Displacement).

Comment (6). In the process flow, description of Figure 7 in the main text is not consistent with the Fig7: The text includes Fig7(a-1) to (a-7) but figure 7(a-1) to (a-5) exist.

Response: We sincerely regret our oversights. Figure 7 and its explanation are consistent in the revised version of the manuscript. We have also improved the quality of the figure 7.

Figure 7. Process flow for the fabrication of the proposed cantilever sensor integrated with a PDMS-encapsulated crack sensor.

Page no 19. Revised version of the manuscript.

Process flow of the PDMS-encapsulated crack sensor. Figure 7 shows a schematic of the silicon rubber cantilever integrated with a PDMS-encapsulated crack sensor. The fabrication process was divided into two parts, one for the polymer part and one for the glass part. A silicon rubber with a relatively high Young's modulus (compared with PDMS) was used for the fabrication of the crack sensor because the Pt patterns on the silicon rubber were more stable. As shown in Figure. 7 (a-1), 2 g of silicon rubber compounds (KEG-2000-80A/B, Shin-Etus) were placed on a PUA mold with nano-grooves. A 120- μm -thick feeler gauge (NIKO, Feeler 0.12) was placed on the edge of the PUA substrate, and a polyimide (PI) film was applied and cured by applying a pressure of 4 MPa at 130 °C

for 30 min using a thermal press. Then, a shadow mask was placed on the backside of the Si rubber film patterned with nano-grooves and a Pt thin film with a thickness of 20 nm was deposited using a sputter (Figure. 7 (a-2)). Next, irregular cracks were generated in the Pt thin film by stretching the Si rubber film by approximately 2% using a laboratory-made stretcher, as shown in Figure 7 (a-2). To increase the bonding strength of the PDMS used as the encapsulation layer, an adhesion layer (Cr/SiO₂: 2 nm/2 nm) was deposited on the cracked Pt layer using a thermal evaporator (Fig. 7 (a-3)). The deposited SiO₂ layer was chemically bonded to the encapsulation layer (PDMS) via an oxygen-based atmospheric plasma treatment (CUTE-1MPR, Femto Science Inc.), as shown in Figure. 7 (a-4). The shape of the encapsulated crack sensor was precisely defined using a roll-to-plate (TSUKATANI, BFX) and pinnacle die, as shown in Figure. 7 (a-5). The main other process consisted in the fabrication of a glass body in which Au electrodes were formed to enhance the electrical reliability and stability of the crack sensor (Fig. 7 (b)). First, a photoresist was patterned onto a glass wafer and Cr/Au (3 nm/30 nm) was deposited using a thermal evaporator. After deposition, electrodes were formed on the glass wafer by removing the photoresist using acetone, as shown in Figure. 7 (b-1). The glass wafer with Au electrodes was diced into a 9 mm × 12 mm shape using a dicing saw (AM Technology, NDS200), as shown in Figure. 7 (b-2). Finally, the Si rubber cantilever integrated with the crack sensor and the glass body were chemically bonded via an oxygen-based atmospheric plasma treatment (CUTE-1MPR, Femto Science Inc.), as shown in Figure. 7 (c).

Reviewer #2

Summary Recommendation: Highly durable crack sensor integrated with silicone rubber cantilever for measuring cardiac contractility in culture media. The authors claim to have developed a method to encapsulate a crack-based pressure/force sensor with PDMS, thereby allowing the sensor to be used for long-term cell culture to study the biology and pharmacology of contractile cells. The authors have convincingly addressed this claim, and the work is interesting. However, in its current state, the manuscript is more suited for a field-specific journal such as Lab-On-A-Chip, than for a general-interest journal like Nature Communications. In particular: articles involving new technology that are typically suited for this journal rise above development of interesting new devices and go on to use these devices to learn new biology.

Response: We thank the reviewer for his valuable time, appreciation and a very positive feedback on our work. We acknowledge his valuable suggestions toward improving the quality of our manuscript. We made our best efforts to address all the valuable comments raised by the reviewer. We have carried out the additional experimental analysis including measuring maturation of the cardiomyocytes over 40 days of the culture period, force-frequency relationship and contraction force of the drug treated cardiomyocytes over a period of time to evaluate the rare events such as changes in the regularity of contraction force of the cardiomyocytes and immunofluorescence of the mature and immature cardiomyocytes. In addition, based on the suggested references, we have also modified the introduction part of the manuscript to clearly describe the state-of-the-art and novelty of the manuscript. We sincerely, believe that the additional experimental data and the corresponding modifications would justify our conclusions and get back reviewer's confidence and provide us an opportunity to publish our manuscript in this esteemed journal "Nature Communication"

Comment (1). By the authors' own admission, crack-based pressure sensors are not inherently novel, although their encapsulation strategy appears to be. However, the encapsulation is not ground-breaking enough on its own. The ability to do non-optical monitoring of force is an important advance, because optically-based data acquisition is time consuming, not amenable to parallelization (all samples have to be done one-at-a-time in a typical setup) and produces massive quantities of data (high speed video). However, development of non-optical force monitoring is also being done by others (see Mannhardt et al. J Tissue Eng Regen Med 2019, Lind et al. Nat Mater 2017, Bielawski et al. Tissue Eng. Part C Methods 2016), so the level of novelty that would be

appropriate for a general-interest general would be in applying such a system to ask new questions, rather than simply developing the system.

Response: We really appreciate the kind suggestion made by the reviewer. We are all also thankful to the reviewer for providing us the useful references. Based on the reviewer recommendation and suggested references we have modified the introduction of the revised manuscript. The novelty of the present work and its importance are compared to the above-mentioned reported papers in the revised version of the manuscript.

Page no 3-4: Revised version of the manuscript.

Several non-optical methods also have been developed to measure the mechanophysiology of the cardiomyocytes. For instance, Bielawski et al. proposed a magnet-integrated post and giant magnetoresistive (GMR) sensor array to measure the contraction force of the cardiomyocytes [15]. The GMR sensor can measure even large displacement of the post owing to the contraction force of the cardiomyocytes. However, the use of GMR in the high-throughput analysis is limited due to the interference of the adjacent magnetic field. Mannhardt et al. proposed the piezo actuator-based measurement system for high-throughput real-time analysis of the heart rate, contraction and relaxation rate of the engineered heart muscle (EHM) [16]. The strain caused by the mechanical deformation of a metal thin film can be converted into an electrical signal, such as resistance, and analyzed using integrated sensors. Lind et al. established a new class of instrumented cardiac microphysiological devices via multi material three-dimensional (3D) printing [17]. They fabricated the cantilever structure, strain sensor, groove and culture well all at once. The cantilever integrated with a carbon black strain sensor can measure the contraction/relaxation of cardiomyocytes in real time. However, the sensor exhibits low sensitivity due to the low gauge factor (GF) of the integrated carbon black strain sensor (2.51) which is similar to that of the commercial strain sensors. Our previous study also demonstrated the piezo-resistive sensor integrated PDMS cantilever for measuring the contraction force of the cardiomyocytes [18]. However, proposed measurement systems showed less sensitivity due to a low GF (< 3) of the metal strain sensors. Therefore, developing the highly sensitive sensor to continuously measure the contraction force and detect rare events such as changes in the regularity of the contraction force of drug treated cardiomyocytes are imperative for the next generation high throughput drug screening platform.

Comment-2: As the authors have developed a new way to perform long-term, non-optical monitoring of cardiac tissue contractile forces, they have the opportunity to do some rather interesting studies here – these would merit publication in Nature Communications. For example, several labs have shown recently that various physical manipulations (for example, continuous electrical pacing – see Richardson-Bouchard et al. Nature 2018, Lee et al. Cell 2019), chemical changes (for example, changes in ATP substrate – see Mills et al. PNAS 2017) and growth factors (for example IGF1 and NRG1, see Rupert and Coulombe, Stem Cells International 2017) can enhance maturation of pluripotent stem cell derived cardiomyocytes. Similar data have been shown for NRVM (see Godier-Furnemont et al. Biomaterials 2015). In all of these studies, however, we get a snapshot of “before” and “after” maturation. Part of why this happens is because the current optical techniques used to measure force and physiology are inherently data intensive, and sometimes are terminal (for example, tissues or micro-tissues that are not grown on posts, e.g. Zimmerman et al. Biotechnol Bioeng. 2002, Huebsch et al. Sci Rep 2016, require mounting on force-apparatus to get contractile force measurements and perform Frank-Starling analysis). The current technology developed by Kim et al. could give us a continuous-readout of force development. In turn, the time-scales over which maturation occurs (e.g. over which twitch force increases – or better, the time-course over which tissues begin to exhibit a positive Force-Frequency Relationship) could give some clues as to biologic mechanism (e.g. epigenetic changes are likely to involve very long time-scales, whereas changes in protein localization would take hours or less). This would be interesting and novel. Similarly, the ability to monitor continuously to detect rare events (for example, changes in the regularity of contraction that are only occurring over a few minutes every hour) in drug-treated or mutation-harboring cells (in particular, disease-prone iPSC-cardiomyocytes) would also be interesting and novel and merit publication in Nature Communications.

Response: We are thankful to the reviewer for his valuable suggestion, as recommended by the reviewer, we have studied the maturation of the cardiomyocytes over 40 days of the culture period, force-frequency relationship and measured the contraction force of the drug treated cardiomyocytes over a period of time to evaluate the rare events such as changes in the regularity of contraction force of the cardiomyocytes.

Figure S13. Real-time measurement shows the contractile force and changes in contractility due to maturation of cardiomyocytes.

The heartrate and cantilever displacement were continuously measured over 36 days of the culture period (Figure S13). The cultured cardiomyocytes showed measurable heart rate at the early stage of culture (4th) period. The contraction force of the cultured cardiomyocytes found to be increase with increasing the culture period and reached highest on day 22 of the culture period and then started to decrease due to the finite lifetime of the cultured cardiomyocytes. The gradual increase in the displacement of the cantilever and beating rate of the cardiomyocytes at the early stage of the culture period indicates that the cells are matured.

Page no 16-17: Revised version of the manuscript.

The real-time drug screening ability of the proposed PDMS encapsulated crack sensor integrated silicone rubber cantilever was analyzed to detect rare events such as changes in the regularity of the contraction force of the drug treated cardiomyocytes. Figure 7 shows the real-time changes in the cell contractility and heart rate of the cardiomyocytes treated with 1 μM verapamil. The cantilever displacement was measured at regular interval of time after treating the cultured cardiomyocytes with a concentration of 1 μM Verapamil. At control state the displacement of the cantilever was found to be 1 and then found to be decreasing with increasing time after drug treatment. The

gradual decrease in the contraction force of the cardiomyocytes could be due to the slow diffusion of Verapamil into the cell wall. The decrease in contraction force of the cardiomyocytes could be explained based on the following lines. Verapamil is an L-type Ca^{2+} channel blocker and prevents the calcium influence to prolong atrioventricular node effective refractory period thus reducing the contraction force. After wash-out, the cultured cardiomyocytes retain slowly its inherent properties. After 70 minutes from wash-out drug treated cardiomyocytes completely refreshed to values that are close to control. As shown in Figure 7, the proposed sensor detected various unusual contractility of cardiomyocytes which could be attributed to the adverse effects of Verapamil.

Figure 7. Real-time changes in the cell contractility and heart rate of the cardiomyocytes treated with 1 μM verapamil.

As kindly advised by the reviewer, authors also agree that our current platform can be utilized in exploring the epigenetic changes such as DNA methylation, histone modification, and microRNA-based gene regulation during the postnatal maturation of cardiomyocytes (CMs). It is known that epigenetic profiles of the heart tissue rely on its cell types and their culture conditions (i.e. culture time, flexibility of cell-adhered substrate, or treatment of chemicals). We have isolated the CMs from the neonatal rat heart and cultured them on the flexible substrate to induce its mechanical deflection due to their cyclic contraction, which may mimic the *in vivo* ventricular beating. Since our platform

allows a long-term culture of CMs, it is expected that we can profile epigenetic dynamics as a function of time for least on month, which will uncover the molecular mechanism associated with the postnatal maturation or differentiation of CMs. As one of co-author groups in Department of Biological Sciences, Prof. Kim, E.-S.'s group has a plan to analyze the epigenetic pattern of GpC methylation of CMs that will be harvested from the current platform, with an aim to expanding to analysis histone acetylation/deacetylation or methylation.

Comment-3: Another general point is that since the novelty appears to be the ability to use the device long-term, for applications in cell characterization, more robust characterization of the cells themselves is needed. For example, more detailed analysis of gene expression and protein localization after the extended culture.

Figure 8. Representative immunofluorescence of cardiomyocytes cultured on PDMS encapsulated crack sensor integrated silicone rubber cantilever (a) matured and immature cardiomyocytes. Green: α -actinin, Red: Vinculin. (b) Sarcomere length of the matured and immature cardiomyocytes. (c) normalized displacement of the cantilever owing to the matured and immature cardiomyocytes. (d-f) RT-qPCR results (MHC 6, α -actinin), (e) changes in cTnT expression of the matured and immature

cardiomyocytes before and after drug treatment. (f) changes in β -actin expression of the matured and immature cardiomyocytes before and after drug treatment. (Immature and matured cardiomyocytes are obtained at day 10 and 22 of the culture period, respectively)

Page no 18-19: Revised version of the manuscript.

To demonstrate the improved performance of the proposed platform, we have performed the traditional gene expression and protein localization analysis. Figure 8 shows the immunofluorescence of the immature and matured cardiomyocytes obtained from the PDMS encapsulated crack sensor integrated silicone rubber cantilever at day 10 and 22 of the culture period. The immature cardiomyocytes had a randomly oriented α -actinin. Meanwhile the mature cardiomyocytes showed elongated in the direction of the grooves and formed an aligned cell tissue like that seen in vivo (Figure 8a). Further, the matured cardiomyocytes showed higher sarcomere length and cantilever displacement compared to the immature cardiomyocytes (Figure 8 (b, c)). RT-qPCR of the four genes analysis confirms the improved contractile force (MHC 6, cTnI) higher maturation (α -actinin, β -actin), and improved contractility of the matured cardiomyocytes compared to the immature cardiomyocytes (Figure 8(d-f)).

Comment-4: Stylistically, the article needs to be much more comprehensive in referencing the literature related to cardiac tissue engineering. The authors appear to be using a “cells on a cantilever” system similar to the system developed by the lab of Kevin Parker, described in Feinberg et al. Science 2007, but have not referenced this. They also discuss maturation of tissue over time, but do not cite any of the literature on this rather extensively studied topic as well.

Response: We sincerely regret our oversights. We have included the suggested references and other relevant references in the revised version of the manuscript.

Page no 25. Revised version of the manuscript.

Ref no: 14: Feinberg, A. W., Feigel, A., Shevkoplyas, S. S., Sheehy, S., Whitesides, G. M., & Parker, K. K. (2007). Muscular thin films for building actuators and powering devices. *Science*, 317(5843), 1366-1370.

Comment-5. I would also recommend putting the device design into Figure 1, so that it incorporates more actual data on top of the conceptual material in that figure.

Response: We really appreciate the kind suggestion made by the reviewer. As recommended by the reviewer, we have incorporated the device design in the Fig1 of the revised version of the manuscript.

Figure 1. (a) Dimensions of the different layer of the proposed crack sensor. (b) Schematic of a Si rubber cantilever composed of various layers and operation principle of the cantilever sensor integrated with a PDMS-encapsulated crack sensor to measure the contractile force of cardiac cells in liquids. (c) Nano-patterns used to align the cardiac cells on the cantilever surface. (d) Circuit diagrams of two different crack sensors operating in culture media.

Page no 6-7. Revised version of the manuscript.

Concept of highly durable crack sensor working culture media. The proposed cantilever integrated with the PDMS encapsulated crack sensor consists of a silicone rubber cantilever, a PDMS thin film and a glass body. Figure 1 (a) shows dimensions of the various layer and of the proposed crack sensor. Figure 1(b) shows schematic of a PDMS encapsulated crack sensor integrated silicone rubber cantilever. The highly sensitive strain sensor based on metal cracks was formed on the silicone rubber cantilever, allowing us to precisely monitor the strain changes caused by the mechanical contraction of cultured cardiomyocytes on the cantilever. In addition, Au patterns formed on the glass substrate were electrically connected to other Au patterns formed on the silicone rubber via plasma bonding. The use of the glass body with the Au patterns greatly improved the electrical reliability of the fabricated cantilever sensor. Figure 1 (c) shows a schematic of nano-patterns formed on the cantilever surface to align cardiomyocytes along the length direction of the cantilever and the principle of the cantilever sensor used to directly measure the contractile force of cardiomyocytes. Figure 1 (d) shows equivalent circuit diagrams of two different crack sensors operating in culture media. The electrical pathway of conventional crack sensors consists of a parallel resistor with cracks and liquid, whereas the current flow in the proposed crack sensor only occurs through cracks generated on the cantilever. Crack sensors exposed to culture media behave very unstably like variable resistors. Furthermore, as for the previous PDMS-coating method proposed by Hong et al., the protection layer is physically combined with the crack sensor via spin-coating, which also affects the long-term durability of the crack sensor [24]. However, the proposed crack sensor was chemically bonded to a very thin PDMS film via plasma bonding and can be used very reliably in various ionic liquids. The use of an intermediate Cr/SiO₂ layer on the sensing layer greatly improved the adhesion force between Pt and PDMS.

Comment-6: From a stylistic standpoint I would recommend deleting “novel” from the abstract and manuscript.

Response: As recommended by the reviewer, we have removed the word “novel” in the revised version of the manuscript. However, we have mentioned the first-time demonstration of the crack sensor to measure the contraction force of the cardiomyocytes.

With this research background and considering the advantages of the crack sensor, herein, we present for the first time the demonstration of the highly sensitive crack sensor integrated silicone rubber cantilever to analyze the drug-induced changes in the cardiac contractility in real time. The durability of the crack sensor in the cell culture medium greatly improved by the PDMS encapsulation layer. The mechanical durability of the crack sensor was amazingly improved due to the chemical bonding between the crack sensor and protection layer. The crack-based sensor made of platinum (Pt) was chemically bonded with a PDMS thin layer by depositing an adhesion layer (SiO₂: 2 nm) on the Pt. The plasma bonding process performed at a low vacuum and room temperature not only does not affect the function of the conductive layer but also avoids contamination of the cantilever surface. The fabricated crack sensor exhibited a high GF of 9×10^6 at a strain of 1% even after the formation of the encapsulation layer. The durability (26 days, >5 million heartbeats) of the sensor was also confirmed in various solutions, such as DI water, culture media. After various basic experiments, the changes in the contractile force of cardiac cells induced by two drugs, namely Verapamil and Quinidine, were evaluated using silicone rubber cantilever integrated with the PDMS-encapsulated crack sensor. The experimental findings of the crack sensor were compared with the data obtained from the laser vibrometer and the results are consistent with each other. The proposed crack-based sensor integrated silicone rubber cantilever arrays are expected to be applicable in various fields, such as cardiac toxicity tests in the initial stage of the development of new drugs, owing to its excellent sensitivity, reversibility, reproducibility and stability over long periods in a culture medium.

Comment-7: Finally, in most instances, when the authors are referring to previous work, they should cite the last name of the first author rather than the first name (for example, “Ribeiro et al.” rather than “Alexandre et al.” on page 2, line 40).

Response: We sincerely regret our oversights and the references are in now in the correct format in the revised version of the manuscript.

Page no 24. Revised version of the manuscript.

Ribeiro, A. J., Zaleta-Rivera, K., Ashley, E. A., & Pruitt, B. L. Stable, covalent attachment of laminin to microposts improves the contractility of mouse neonatal cardiomyocytes. *ACS Appl. Mater. Interfaces*. **6**, 15516-15526 (2014).

Reviewer #3

Summary Recommendation: It is a good manuscript. I like the authors to modify the manuscript based on the following comments.

Response: We thank the reviewer for his valuable time, critical appreciation and a very positive feedback on our work. His words of encouragement and an unequivocally strong support surely boosts us to continue our research efforts with a lot of confidence. We acknowledge his valuable suggestions towards improving the quality of our manuscript. We have carried out the additional experimental analysis to address all the concerns raised by the reviewer.

Comment (1). Please check the English of the manuscript by a technical writer.

Response: As recommended by the reviewer, the revised manuscript has been carefully checked the modified by the English language editorial service. The certificate for English editing has been included with this response letter for the reviewer's kind consideration.

Comment (2). It may be good to use the references in a more-better way. For example, "Typical high-sensitivity sensors in the field of microelectromechanical system (MEMS) include silicon nanowires (SiNWs), graphene or carbon nanotube (CNT) composites, crack based sensors, etc. [10-21]. The references [10 - 21] is too many.

Response: We sincerely regret our oversights. As recommended by the reviewer, the references are properly utilized in the revised version of the manuscript.

Comment (3). Provide the dimension details of Figure 1a to make it more clear. It is not clear why the Si layer need to be so thick.

Response: As recommended by the reviewer, we have provided the dimension of each layer of the silicone rubber cantilever in the revised version of the manuscript. In this work, we followed heat-pressing method to fabricate the silicon rubber cantilever. We have controlled the thickness of silicone rubber layer through the feeler gauge. When we applied the external pressure more than 4 MPa during the silicone rubber thin film formation, the PUA-glass plate used as groove mold was damaged and the PI film deformed. Whereas, when the applied pressure lower than 4 MPa the formation of a uniform silicone layer was extremely difficult. Therefore, we applied the optimized external pressure of 4 MPa during the fabrication of silicone rubber layer which produced $\sim 120 \mu\text{m}$ thick silicone rubber cantilever.

Page no 6. Revised version of the manuscript

Concept of highly durable crack sensor working culture media. The proposed cantilever integrated with the PDMS encapsulated crack sensor consists of a silicone rubber cantilever, a PDMS thin film layer and a glass body. Fig. 1 shows the schematic illustration and operating principal of the proposed cantilever. The highly sensitive strain sensor based on metal cracks was formed on the silicone rubber cantilever, allowing us to precisely monitor the strain changes caused by the mechanical contraction of cultured cardiac cells on the cantilever. In addition, Au patterns formed on the glass substrate were electrically connected to other Au patterns formed on the silicone rubber via plasma bonding. The use of the glass body with the Au patterns greatly improved the electrical reliability of the fabricated cantilever sensor. Fig. 1 (a) shows the various layer and dimensions of the proposed crack sensor. Fig. 1(b) shows the principle of the cantilever sensor used to measure the contractile force of cardiac cells. Fig. 1 (c) shows nano-patterns formed on the

cantilever surface to align cardiac cells along the length direction of the cantilever. Fig. 1 (d) illustrates equivalent circuit diagrams of two different crack sensors operating in culture media. The electrical pathway of conventional crack sensors consists of a parallel resistor with cracks and liquid, whereas the current flow in the proposed crack sensor only occurs through cracks generated on the cantilever. Crack sensors exposed to culture media behave very unstably like variable resistors. Furthermore, as for the previous PDMS-coating method proposed by Hong et al., the protection layer is physically combined with the crack sensor via spin-coating, which also affects the long-term durability of the crack sensor [16]. However, the proposed crack sensor was chemically bonded to a very thin PDMS film via plasma bonding and can be used very reliably in various ionic liquids. The use of an intermediate Cr/SiO₂ layer on the sensing layer greatly improved the adhesion force between Pt and PDMS.

Figure 2. (a) Dimensions of the different layer of the proposed crack sensor. (b) Schematic of a Si rubber cantilever composed of various layers and operation principle of the cantilever sensor

integrated with a PDMS-encapsulated crack sensor to measure the contractile force of cardiac cells in liquids. (c) Nano-patterns used to align the cardiac cells on the cantilever surface. (d) Circuit diagrams of two different crack sensors operating in culture media.

Comment (4). Page 6: "The sensing mechanisms of crack sensors have been reported several times as shown in Fig. S1 [13-22]." It is not clear what the authors mean by Fig. S1?

Response: As recommended by the reviewer, we have elaborated in detail the sensing mechanism of the crack sensor in the revised version of the manuscript. The detailed explanation of the Fig. S1 are now are provided as the supporting information of the revised manuscript.

Figure S1. Conceptual diagram related to the operating principle of the crack sensor.

Page no 3-4. Supporting information of the revised version of the manuscript.

Fig. S1 describes the mechanism of crack sensor that are distinctly different from the commercial strain gage sensor. The cured silicone rubber cantilever integrated with the proposed crack sensor was chemically bonded to PDMS by plasma treatment in an oxygen atmosphere for long-term use in conductive culture medium. Since the metal layer can-not chemically bond with PDMS encapsulation layer even after the plasma treatment, an adhesion layer of Cr/SiO₂ was deposited on the Pt layer for appropriate adhesion between the Pt metal layer and the PDMS layer. The proposed crack sensor retains its non-linear characteristics same as the conventional crack sensor even after embedding the Cr, SiO₂ and PDMS.

The physical model in Fig. S1 illustrates the mechanism of the proposed crack sensor. At the initial state, all the lips of the crack are in a closed state. When strain is applied to the crack sensor (State 1), all the lips are instantly opened, and the resistance increases as the electrical pathways are blocked.

Then the lips are partially closed due to the compressive force of the Poisson's ratio (State 2) and partially open the electrical pathway. Thus, through the repetition of states 1 and 2, the crack sensor exhibits an exponential resistance increase rather than momentarily increase in resistance. Finally, when the applied strain approaching or exceeding 1% all the lips are opened, and the resistance reaches to infinity (state 5). After withdrawing the applied tension, the gap between the cracks closed and the initial resistance of the crack sensor is completely restored.

Detailed theoretical modeling and the non-linear characteristics of the crack sensor is explained in detail in the supporting information according to our previous study (Scientific reports, 7, 40116).

For that sensor due to a technology of producing large unidirectional strain the free cracks cut the sensor strip through so that the normalized conductance \mathcal{S} of the sensor vs strain ε

$$\mathcal{S} = \int_{\varepsilon}^{\infty} \mathbf{P}(x) dx \quad (\text{S1})$$

was determined by the probability distribution function (pdf) $\mathbf{P}(x)$ of the steps on a crack lip¹⁵ making contacts between the lips. For a free crack we found an equation for $\mathbf{P}(x)$ with the only “size” parameter – the strain ε_0 that corresponds to the crack gap width $k \varepsilon_0$ being about the grain size $x_0 = k \varepsilon_0$

$$\mathbf{P}(x) = \mathbf{P}(1/x)/x^2, \quad (\text{S2})$$

where $x = \frac{\varepsilon}{\varepsilon_0}$ and k is the proportionality factor to be defined by relating the crack gap width to the strain¹⁵. k can be different for different material realization of parallel crack systems and should be obtained from experiment. Physically, Eq. (S2) states that tiny steps of the crack asperity made by the shifts of grains are distributed the same as the large steps made by grain piling, because the substrate elastic field being scale-less and thus having no characteristic length may not distinguish between tiny and large meandering asperity. Among solutions of Eq. (S2) one may choose either the *log-normal* pdf

$$\mathbf{P}(\varepsilon) = \frac{1}{\varepsilon \mu \sqrt{\pi}} \exp\left(-\frac{(\ln(\varepsilon/\varepsilon_0))^2}{\mu^2}\right) \quad (\text{S3})$$

or a nearly identical *log-logistic* pdf

$$\mathbf{P}(\varepsilon) = \frac{B}{\varepsilon_0} \frac{(\varepsilon/\varepsilon_0)^{B-1}}{(1+(\varepsilon/\varepsilon_0)^B)^2} \quad (\text{S4}),$$

where μ and B are parameters of the pdf.

Both distributions of Eqs. (S3) and (S4) belong to the class of so-called skew distributions with long tails. As we discussed in Ref. 1, the non-zero probability of large but rare contacts between crack lips lies in the essence of the mechanism of the conduction through the crack and is therefore in concordance with the tailed distributions. With Eq. (S3), Eq. (S1) gives for the resistance $R = 1/S$ as a function of strain the following:

$$R = 2 / \left(1 - \operatorname{erf} \left(\frac{\ln \left(\frac{\varepsilon}{\varepsilon_0} \right)}{\mu} \right) \right), \quad (\text{S5})$$

$\operatorname{erf}(x)$ is the error function. Eq. (S5) renders the normalized resistance that remarkably fits the experiment¹⁵ for the strains up to 2%. At the same time, one can show that the log-logistic pdf of Eq. (S4) together with Eq. (S1) leads to

$$R = 1 + (\varepsilon/\varepsilon_0)^B \quad (\text{S6})$$

that fits the experiment¹⁵ with fitting parameters $\varepsilon_0 = 0.39$ and $B = 2.39$ (see Fig S12) with the same accuracy as the log-normal pdf of Eq. (S5). Yet, the *power-law* function of Eq. (S6) is much simpler than the error function in Eq. (S5). We may suggest this universal power law for data fitting by experimentalists who study free parallel cracks.

Comment (5). It is reported that the resistance of the crack sensor changes with strain. I am curious to know whether the authors have attempted to see whether there is any change of capacitance too?

Response: We do agree with the reviewer’s opinion that the external strain causes change in resistance also capacitance of the crack sensor. The applied external strain opened the lips and reduces the electrical paths of the crack sensor. The resistance component generated at the contact point of the cracks and the gap between the cracks behave like resistance and capacitance parallel circuit (Figure S1b).

Figure S1b. Equivalent circuit modeling of the proposed PDMS encapsulated crack sensor integrated silicone rubber cantilever.

The impedance component of the crack sensor shown in the equivalent circuit model (Fig. S1b) of the PDMS encapsulated crack sensor integrated silicone rubber cantilever can be represented as Eq. (S7). However, we have carried out all the sensing performance measurement of the crack sensor through DC power, hence, the change in capacitance (acts as open circuits in a DC condition) due to the applied tensile strain is not considered in this present work. Therefore, the equivalent circuit of the crack sensor can be expressed as shown in Fig. S1c.

$$Z = \frac{R}{1+j\omega RC} , \quad \omega = 2\pi f , \quad (S7)$$

Figure S1c. Equivalent circuit modeling of the PDMS encapsulated crack sensor integrated silicone rubber cantilever.

Comment (6). “The electric resistance of the crack sensor increased 1.7 times for a 0.3% strain and increased approximately 9,000 times for a 1% strain”. The change of resistance is too large compared to 0.3% to 1%. What is the reason behind this such a large change? Is the sensor still come back to their original structure/condition after the removal of strain? Figure 2 shows the hysteresis characteristics but it is not fully visible.

In this study, we proposed the crack sensor integrated silicone rubber cantilever to analyze the mechanical properties (contraction force, heartbeat) of cardiac cells for a long time. The proposed crack sensor exhibits the high sensitivity and non-linear characteristics. The nonlinear performance is the main characteristics of the crack sensor as reported by the various studies (NATURE, 2014, 516, 222.). The non-linear characteristics of the crack sensor can be explained based on the

following lines. At the initial state, all the lips of the crack are in a closed state. When the less tensile strain is applied to the crack sensor, the lips of the cracks are instantly opened, and the resistance increases as the electrical pathways are blocked (state 1). Then the lips are partially closed due to the compressive force of the Poisson's ratio of the silicone rubber and partially open the electrical pathways (state 2). Thus, through the repetition of states 1 and 2, the crack sensor exhibits a nonlinear increase in resistance rather than momentarily increase. Finally, when the applied strain approaching or exceeding more than 1% all the lips are opened, and the resistance reaches to infinity (state 5). After withdrawing the applied tension, the gap between the cracks closed and the initial resistance of the crack sensor is completely restored. Owing to these non-linear characteristics, the crack sensor exhibits the extremely large resistance ratio variation (R / R_0) $\sim 90,000$ at a strain range of 0-1% and relatively smaller variation ~ 1.7 at a strain range of 0-0.3%.

However, the non-linear characteristics and small resistance ratio variation at low tensile range are not the disadvantages of the crack sensor. As the reviewer well known, the change in resistance and strain produced by the cultured cardiomyocytes are relatively very small. In this present study, the maximum strain produced by the cultured cardiomyocytes on the cantilever is $\sim 0.03\%$ which causes only $\sim 100 \mu\text{m}$ cantilever deformation. In addition, the gauge factor of the crack sensor is ~ 156 times higher than the commercial piezo resistive sensor at 0.03% strain (Figure S6). Therefore, the proposed crack sensor is highly sensitive enough to detect even the smaller variation in the resistance of the cantilever caused by the contractile force of the cultured cardiomyocytes.

Figure S6. Bar plot representing the gauge factor of the piezo resistive sensor and the crack sensor

at 0.03% of applied strain.

As for as the hysteresis concerns, the proposed sensor exhibits the hysteresis only at higher strain (>0.5%) range. At the lower tensile in the range of 0-0.5% the crack sensor shows no hysteresis (Figure S2). In this present investigation, the maximum strain produced by the cultured cardiomyocytes is $\sim 0.03\%$ which causes $\sim 100\ \mu\text{m}$ cantilever displacement. Figure S2b shows the change in resistance of the crack sensor as a function of displacement. Different displacement with the ramp of $10\ \mu\text{m}$ was applied to the crack sensor integrated silicone rubber cantilever using a motorized stage and measured the resistance variation. The resistance of the sensor increases almost linearly with increasing the displacement of the sensor. The obtained experimental data demonstrated that the hysteresis does not appear at low tensile range where the crack sensor operated to measure the contraction force of the cardiomyocytes.

Figure S2. (a) Hysteresis of the PDMS-encapsulated crack sensor integrated silicone rubber cantilever in the tensile range of 0-0.5%, (b) change in resistance of the PDMS-encapsulated crack sensor integrated silicone rubber cantilever as a function of displacement.

In addition, the response time of the crack sensor is very similar to that of the conventional optical (laser displacement measurement) method. Therefore, it is easy to analyze the contraction force of the cardiomyocytes changing in real-time. A $100\ \mu\text{m}$ displacement was applied to the integrated silicone rubber cantilever using a motorized stage and measured the displacement using laser vibrometer and crack sensor. The laser vibrometer and crack sensor showed the response time of 121 and 123 ms and remained constant after the displacement was applied (Figure S8). It is confirmed that this method can be used to analyze cardiomyocyte contractility and heart rate.

Figure S8. A comparison of the response of a laser vibrometer and a crack sensor.

Comment (7). Figure 3 is not clearly visible. The effect of temperature is not much due to using a polymer with a low thermal conductivity coefficient. This means that the change takes place very slowly. It will be good to see the effect of temperature with a long time of operation. This is important as the effect of toxicity is also not expected to act very quickly.

Response: As recommended by the reviewer, the Figure 3 has been modified in the revised version of the manuscript.

Figure 3. Changes of resistance under different operating conditions for two different cantilevers

with and without PDMS encapsulation. Performances of the crack sensors as a function of (a, d) humidity changes (45–95 %) and (b, e) temperature changes. (c, f) Stability of the PDMS-encapsulated crack sensor in a stage-top bioreactor at 37 °C.

As mentioned by the reviewer, the effect of temperature on the deformation of the cantilever is not significant owing to the use of low thermal conductivity polymer materials. Figure S5 (a) shows the change in resistance of the PDMS encapsulated crack sensor integrated silicone rubber cantilever as a function of time at different incubator set temperature. As observed the resistance of the sensor increases with increasing temperature and stabilized approximately in 10 min and maintained stable resistance with further increasing time. Figure S5(b) shows the displacement of the PDMS encapsulated crack sensor integrated silicone rubber cantilever sensor at 40 °C. The sensor showed almost stable displacement over 20 min indicating the negligible thermal effect on the proposed sensor.

Figure S5. (a) Change in resistance of the PDMS encapsulated crack sensor integrated silicone rubber cantilever as a function of time at different temperature. (b) Displacement of the PDMS encapsulated crack sensor integrated silicone rubber cantilever as a function of time under the home-made stage top incubator at 40 °C.

The cardiomyocytes seeded proposed cantilever is placed in an incubator at constant temperature and humidity. The displacement of the cantilever owing to the contraction force of the cantilever and the resistance change of the crack sensor were monitored on 4th day of the cell culture. In addition, to improve the accuracy of the measurement, we have carried out the drug screening

process after 9 days of cell contraction. Thus, it is expected that there will be no deformation due to the use of a polymer having a low thermal conductivity coefficient. Since, all the experiments were carried out at a controlled temperature and humidity the displacement of the cantilever was kept constant in the incubator during the drug toxicity test.

Comment (8): It will be good to see more clear pictures for figures 5 and 6.

Response: As recommended by the reviewer, the Figure 5 and Figure 6 are modified in the revised version of the manuscript.

Figure 5. Resistance changes and corresponding displacements of the PDMS-encapsulated crack sensor measured over 26 days (D: displacement).

Figure 6. Changes in sensor resistance and cantilever displacement as a function of drug concentration for (a) Verapamil and (b) Quinidine (D : displacement).

Comment (9): It is not clear why the frequency of opening/closing for the crack sensors initially decreased and then increased again.

Response: In Figure 5, the displacement of the cantilever caused by the contraction / relaxation of the cardiomyocytes was measured using a crack sensor, and it was verified with the data obtained using a laser vibrometer. Figure 5 (a) and (b) show the resistance changes and corresponding displacements of two PDMS-encapsulated crack sensors with different densities cultured cardiomyocyte. We have systematically calculated the beating frequency of the cultured cardiomyocytes at periodic time interval of the cell culture period. Figure S14 also shows the changes in resistance and beating frequency of the proposed crack-based cantilever sensors over 26 days after culturing high density cardiac cells. As observed the cultured cardiomyocytes showed small variation in the beating frequency over the cell culture period. Once the culture media changed

the cultured cardiomyocytes gets refreshed and the beating frequency increased. In general, the spontaneously beating of cardiomyocytes is not constant at all the time so that the crack sensor integrated on the silicone rubber cantilever is not opened and closed at a constant frequency.

This issue has been rectified by applying the external electrical stimulation. The spontaneously beating of cardiomyocytes can be synchronized by applying the electrical stimulation. Two carbon electrodes were placed parallel to the longitudinal direction of the both sides of the silicone rubber cantilever. Then the external electrical stimulation (square-wave pulses, ms, and the electric field was 3 V / cm) was applied to the cardiomyocytes seeded silicone rubber cantilever. The experimental results show that the beating rate of cardiomyocytes can be precisely synchronized to 30 beats/min, 60 beats/min, 90 beats/min, 120 beats/min, 150 beats/min and 180 beats/min as the electrical stimulation frequency increases from 0.5Hz to 3Hz (Figure R1). When the beating frequency was more than 1Hz, cardiomyocytes did not completely relax and contracted. In this way, the maximum displacement of the cantilever was kept constant while the beating frequency due to the electrical stimulation was increased. In addition, the displacement and beat shape of spontaneously beating cardiomyocytes and synchronized beating cardiomyocytes before and after electrical stimulation were analyzed (Figure R2).

Figure R1. Contraction behavior of cultured cardiomyocytes under the external electrical stimulation (only for review).

Figure R2. Comparison of contraction force behaviors before and after electrical stimulation of cardiomyocytes (a) Without electrical stimulation, (b) With electrical stimulation, (c) Comparison of beat shape before and after electrical stimulation (only for review)

Comment (10): The effect of change of environment should be noted and some experimental results to be provided.

Response: We do agree with the reviewer's opinion that the change of environment such as temperature and humidity would greatly influence the sensor performance. To avoid the undesirable interference from the environmental effect, we have investigated in detail all the sensor characteristics under the home-made stage top incubator (Figure R3). Mechanophysiology of the cardiomyocytes using a PDMS encapsulated crack sensor integrated silicone rubber cantilever was carried out at a controlled cell culture environment. The temperature, humidity and CO₂ concentration were maintained constant using a home-made stage-top incubator. The cardiomyocytes seeded PDMS encapsulated crack sensor integrated silicone rubber cantilever was placed in a stage-top incubator and then positioned under a microscope to measure the displacement of the cantilever using the external laser displacement sensor. The change in resistance of the crack sensor was measured using an external electrical circuit. The sensor was constantly protected from the environmental effects during the investigation of sensing characteristics of the crack sensor.

Figure R3. Optical image of stage-top incubator used for the investigation of the sensing characteristics of the crack sensor (only for review).

As mentioned by the reviewer, we have explained in detail the effect of environmental conditions on the sensing characteristics of crack sensor. Figure 3 shows the sensing characteristics of the non-encapsulated and PDMS encapsulated crack sensor according to humidity, temperature and electrolytic solution. The PDMS encapsulated crack sensor exhibits much improved performance under different conditions compared to the non-encapsulated crack sensor owing to the low thermal conductive ($0.27 \text{ W/m } ^\circ\text{K}$) PDMS encapsulation layer.

Page no. 11. Revised version of the manuscript.

Experiments were conducted in environments with varying humidity to monitor the changes in resistance of the crack sensor due to changes in the dielectric constant of the medium between the cracks. The PDMS-encapsulated crack sensor showed very stable behavior in the humidity range of 45–95% compared with the non-encapsulated crack sensor. This is probably due to the encapsulation layer preventing vaporized water molecules from penetrating into cracks. On the other hand, the crack sensor without the encapsulation layer allows water molecules to permeate through its cracks, which may cause irregular crack opening and closing. This results in unstable resistance changes of $\pm 4\%$, as shown in Fig. 3 (a, d). The temperature characteristics of the crack sensor according to encapsulation are shown in Fig. 3 (b,e). For the non-encapsulated crack sensor, the resistance change rate was 170% at a maximum temperature of $35 \text{ }^\circ\text{C}$. However, the PDMS-encapsulated crack sensor exhibited relatively stable changes in resistance with a change of approximately 120% at a maximum temperature of $35 \text{ }^\circ\text{C}$ owing to the protection of the

encapsulation layer. This can be explained by the fact that the crack sensor was protected from sudden temperature changes by using a polymer with a low thermal conductivity coefficient. In addition, because the proposed crack-based cantilever sensor used for measuring the contractility of cardiac cells is to be used in an incubator environment in which temperature and humidity are maintained, changes in the initial resistance can be neglected as shown in Fig. 3 (c,f). For the non-encapsulated crack sensor, its initial resistance value increased up to approximately four times as the temperature increased from room temperature to 38 °C, as required by the stage-top incubator. Exposing this crack sensor to electrolyte solutions had a significant impact on its reliability because its resistance value changed rapidly even in the culture media with a constant temperature. In contrast, the PDMS-encapsulated crack sensor saturated with a resistance change similar to that of simply increasing the temperature in air. The hydrophobic characteristics and mechanical sealing effect of the encapsulation layer mentioned above seemed to prevent the penetration of the culture media into the cracks even after the operating temperature was increased. Therefore, the crack-based cantilever sensor with the chemically bonded encapsulation layer exhibited improved durability in a variety of environments in terms of humidity, temperature, and culture media.

Figure 3. Changes of resistance under different operating conditions for two different cantilevers with and without PDMS encapsulation. Performances of the crack sensors as a function of (a, d) humidity changes (45–95 %) and (b, e) temperature changes. (c, f) Stability of the PDMS-

encapsulated crack sensor in a stage-top bioreactor at 37 °C.

Reviewers' Comments:

Reviewer #1:

Remarks to the Author:

In overall, authors had addressed the comments and questions from reviewers well in a point-by-point manner. Especially, the authors had carried out additional long-term culture experiments over 40 days and had fabricated the multi-cantilevers to show the potential for high-throughput measurement and the reproducibility (low device-to-device variation). In my opinion, technical issues for the proposed system are well addressed and its novelty is suitable for acceptance. However, the description for the merit of the proposed system which enables to reveal the biological mechanism, such as continuous and multiple measurements, is still weak compared to the technical novelty.

The most critical point is the force-frequency relationship was not studied, even though the authors claimed they had done. Of course, the revised figure 7 and figure S13 are very attractive due to its long-term measurement to study the effect of drug treatment. However, they are not a description of the force-frequency relationship. Authors should perform frequency-domain analysis for the long-term measurement, but the results from this work are only time course data. Please refer the following and carry out the additional analysis in the force-frequency relationship in the measured data.

[1] Masao Endoh, Force-frequency relationship in intact mammalian ventricular myocardium: physiological and pathophysiological relevance, *European Journal of Pharmacology* 500 (2004) 73–86.

Reviewer #2:

Remarks to the Author:

Review of Nature Comm 195741-1

The authors have done a great job in responding to the reviewer critiques, adding many new experiments. The manuscript will now be suitable for publication in *Nature Communications*, with minor revisions that include the addition of new analysis of their existing data, and additional pharmacology experiments.

Minor experimental criticisms to be addressed:

1) More quantitative data and statistical analysis is needed. Both for analysis of the device itself (Figure 3 and 4) and the biologic characterization (Figure 5-8), the authors need to provide more experimental data to test the reproducibility of their measurements. For example, in Figure 3f, there is a small change in R/R_0 with temperature. This relative change needs to be quantified, with statistical analysis (standard deviation of the response across multiple devices). Same to be done for gauge factor in Figure 4e and relationship between resistance and average displacement in 4c-d.

2) For additional statistical analysis of mechanical contractility data: displacement (in μm), relative force generated by the cell layer, beat-rate, and contraction kinetics (time constants related to time required to go from baseline to maximum force, and to decay from maximum to resting force) should all be provided, with statistical analysis, in Figures 5 and 6. Multiple biologic replicates should be used to produce error bars describing the IC_{50} values calculated for each drug.

Kinetics of force development and relaxation can be corrected to beat-rate using appropriate means (for example Fridericia's method). Quinidine is causing early after depolarizations (EAD) in figure 6; the relative fraction of EADs should be quantified (either in the figure or the text).

Importantly, these different metrics of contractility are likely to provide different IC50 values, because they describe different biologic processes that are being inhibited by the drugs in the cells. Having provided these data, the authors can describe which processes are most likely to be inhibited, and discuss the ability to distinguish this as an advantage of their system.

3) One or two positive inotropes (isoproterenol, omecamtiv mecarbil, digoxin) should be included in Figure 6 to demonstrate the ability to sense increase in force.

4) Figure 7 – the same statistics described in #2 above should be provided. Also, the in vitro maturation data is probably the most important piece of data in the paper, and should be in the main figure set, not the supplement. The authors should discuss why they think the force decays after day 22, as one would expect the force either continually increases, or plateaus over time. Are the cells becoming detached from the silicone rubber (and therefore not transmitting force to it efficiently) at these later time-points?

5) The included data on force-frequency behavior should either be included or at least described, i.e. “external pacing at different frequencies suggested these cells display a negative force-frequency relationship.”

6) On all figures where multiple biological replicates (cell experiments) or devices are analyzed, the authors need to clearly state in figure captions whether error bars reflect SD or SEM, and what the n is. Appropriate statistical tests (two way t-test to directly compare things, for example, MHC6 expression from day 10 to day 22 in figure 8d, 1-way ANOVA to do multiple comparisons, for example, effects of time on contractility parameters).

Writing changes to be made

1) The concept of “gauge factor” is very important to this publication and it is repeatedly referred to as “GF.” Considering Nature Communications is a general interest journal, the authors need to define GF explicitly before they use the abbreviation.

2) The authors don’t appear to refer to Figures 9 and 10 in the main manuscript – these should be moved to supplemental data. However, even then, the authors need to report what the error bars mean and perform appropriate statistical analysis (Figure 10).

3) There are still quite a few grammatical errors. I have noted as many as I can but the authors do need to have this read over carefully (see below).

Minor changes:

The authors aren’t wrong that non-optical techniques are ideal. However, it is not completely true that specialized equipment is needed for optical techniques (in fact, most labs have microscopes with camera capable of going up to 30 frames/second and therefore can do at least rudimentary contractility analysis). I would instead emphasize that with imaging, it is very hard to parallelize measurements – in 96-well plate formats, for example, it is often necessary to image each well one at a time, so that imaging an entire plate can take half an hour or longer.

The authors keep using the term “mechanophysiology,” which I am not sure is a real word? Why not just use the more standard term “physiology?”

Page 5: I would note that, in addition to limiting crack sensor success, if the metal layer were exposed to culture medium, leached metal ions would likely be very toxic to cells.

Page 19: the authors mention that the immature cells had randomly oriented α -actinin, but their data (Figure 8a) suggests that the cells are pretty well-aligned in both cases.

Minor grammatical errors:

Abstract: line 24 – change “directly measures the cardiac contractility” to “directly measures cardiac contractility.”

Page 3

Line 46: change “imaging analysis” to “image analysis”

Line 51: change “we have proposed the various” to “we have proposed various”

Line 60: change “sensor can measure even large” to “sensor can measure large”

Page 4

Line 80: change “among them, crack sensors” to “among them, crack-based sensors”

Line 82: change “the authors have developed” to “the authors developed”

Page 5

Line 92: change “of the highly sensitive crack” to “of a highly sensitive crack”

Line 93: change “rubber cantilever to...in real time” to “rubber cantilever for real-time analysis of drug and maturation induced changes in cardiac contractility.”

Line 95: delete “amazingly”. Consider replacing with a less sensationalistic word, like “substantially” or “markedly”

Page 7

Line 135: change “behave very unstably like” to “behave very unstably, like”

Reviewer #3:

Remarks to the Author:

The queries have been properly addressed.

I am happy that the current version of the manuscript is suitable for publication.

Manuscript ID: NCOMMS-19-01725A

Title: Highly durable crack sensor integrated with silicone rubber cantilever for measuring cardiac contractility in culture media

Author(s): Dong-Su Kim, Yong Whan Choi, Arunkumar Shanmugasundaram, Yun-Jin Jeong, Jongsung Park, Nomin-Erdene Oyunbaatar, Eung-Sam Kim, Mansoo Choi and Dong-Weon Lee

Point-by-Point Response to the Reviewer's Comments

Reviewer #1

Summary Recommendation: In overall, authors had addressed the comments and questions from reviewers well in a point-by-point manner. Especially, the authors had carried out additional long-term culture experiments over 40 days and had fabricated the multi-cantilevers to show the potential for high-throughput measurement and the reproducibility (low device-to-device variation). In my opinion, technical issues for the proposed system are well addressed and its novelty is suitable for acceptance. However, the description for the merit of the proposed system which enables to reveal the biological mechanism, such as continuous and multiple measurements, is still weak compared to the technical novelty.

The most critical point is the force-frequency relationship was not studied, even though the authors claimed they had done. Of course, the revised figure 7 and figure S13 are very attractive due to its long-term measurement to study the effect of drug treatment. However, they are not a description of the force-frequency relationship. Authors should perform frequency-domain analysis for the long-term measurement, but the results from this work are only time course data. Please refer the following and carry out the additional analysis in the force-frequency relationship in the measured data.

Response: We sincerely thank the reviewer for his valuable suggestions and a very positive feedback on the quality of the revised manuscript. We are also sincerely thanking the reviewer for recommending our manuscript for publication in this esteemed journal "Nature communication". The reviewer's comments were very useful and informative and greatly helped us to improve the quality of the manuscript. The reviewer recommendations are properly addressed in the revised version of the manuscript. The details of the changes made in revised version of the manuscript are

appended in the response letter for the reviewer’s kind consideration. As recommended by the reviewer, we have carried out the additional force-frequency analysis of the long-term measurement and changes in the contraction force of cardiomyocytes as a function of applied electrical stimulation. In addition, we have also included contractile kinetics of cardiomyocytes (relative force, beat rate and rise time and decay time) with statistical analysis for Figures 5, 6, 7, 8 and 9 in the revised version of the manuscript. All these studies are included in the revised version of the manuscript. Once again, we thank the reviewer for providing his valuable time and energy to help us improve quality of the manuscript. We sincerely believe that the modified manuscript justifies our conclusions and get back reviewer’s confidence and opportunity to publish our work in this esteemed journal “Nature Communications”

Figure S19. **a**, Force-frequency relationships of cultured cardiomyocytes as a function of time. **b**, Relative contraction force generated by cardiomyocytes at different applied electrical stimulation frequency. Error bars are mean \pm s.d., $n = 5$; ** $p < 0.01$ measures by one-way ANOVA followed by Tukey’s honest significant difference test. (C and NS are representing control and non-significant)

Page no: 14. Revised version of the manuscript.

Force - Frequency Relationship (FFR) is one of the significant factors for heart contraction analysis [28]. Hence, FFR of the cardiomyocytes cultured on PDMS-encapsulated crack sensor was investigated at different external electrical stimulation pacing from 0.5 Hz to 3 Hz (Figure S19). The spontaneous beating of cardiomyocytes was synchronized by the external electrical stimulation. Two carbon electrodes were placed parallel to the longitudinal direction of both sides of silicone rubber cantilever. External electrical stimulation (square-wave pulses, 2 ms, and the electric field was 3 V /

cm) was applied to the cardiomyocytes seeded on the cantilever. The cardiomyocytes were paced at 0.5, 1.0, 1.5, 2, 2.5 and 3 Hz and the contractility of cardiomyocytes at each frequency was allowed to stabilize for 1 min. The experimental results show that the beating rate of cardiomyocytes can be precisely synchronized with increasing the electrical stimulation (Figure S19a). The contractile force of cardiomyocytes was compared to those at control state to more closely examine the change in contraction force at various stimulation pacing. The contractility of cardiomyocytes was found to be similar to the control state at 0.5 Hz with no significant difference, and then decreased with further increasing the external electrical pacing. The contractility was ~2-fold decreased at 3 Hz compared to the control state (Figure S19b). The decrease in contractile force or negative FFR of culture cardiomyocytes could be attributed to several factors, such as oxygen limitation in the cardiomyocytes, potential overload due to altered Ca^{2+} and changes in intracellular pH [29, 30].

References

- [28] Endoh, M. Force–frequency relationship in intact mammalian ventricular myocardium: physiological and pathophysiological relevance. *Eur. J. Pharmacol.* **500**, 73-86 (2004).
- [29] Antoons, G., Mubagwa, K., Nevelsteen, I., Sipido, K. R. Mechanisms underlying the frequency dependence of contraction and $[\text{Ca}^{2+}]_i$ transients in mouse ventricular myocytes. *The Journal of physiology* **543**, 889-898 (2002).
- [30] Maier, L. S., Bers, D. M., Pieske, B. Differences in Ca^{2+} -handling and sarcoplasmic reticulum Ca^{2+} -content in isolated rat and rabbit myocardium. *J. Mol. Cell. Cardiol.* **32**, 2249-2258 (2000).

Figure 8. **a**, Real-time traces show the cell contractility and beat rate of cardiomyocytes treated with 1 μM verapamil. **b,c**, Displacement of the cantilever after treating the cardiomyocytes with 1 μM verapamil. Error bars are representing mean ± s.d., n=5, * $P < 0.05$, ** $P < 0.01$ measures by one-way ANOVA followed by Tukey's honest significant difference test. **d**, Relative contraction force generated by 1 μM verapamil treated cardiomyocytes at different drug treatment time. Error bars are representing mean ± s.d., n=5, * $P < 0.05$, ** $P < 0.01$ measures by one-way ANOVA followed by Tukey's honest significant difference test. **e,f**, Rise time and decay time of the 1 μM verapamil

treated cardiomyocytes. Error bars are representing mean \pm s.d., $n=5$, $** P < 0.01$ measures by one-way ANOVA followed by Tukey's honest significant difference test.

Page no: 22. Revised version of the manuscript.

The real-time drug screening ability of the proposed silicone rubber cantilever integrated with a PDMS-encapsulated crack sensor was analyzed to detect rare events such as changes in the regularity of contraction force of drug-treated cardiomyocytes. Figure 8a shows the real-time traces of cell contractility and beat rate of cardiomyocytes treated with 1 μ M verapamil. The cantilever displacement was measured at regular interval of time after treating cultured cardiomyocytes with a concentration of 1 μ M Verapamil (Figure 8b). After 40 minutes of drug treatment, the displacement of the cantilever was decreased by 45.63%, from $2.98 \pm 0.105 \mu\text{m}$ (control state) to $1.62 \pm 0.054 \mu\text{m}$. The drug-treated cardiomyocytes were washed out again and measured the displacement of the cantilever. The displacement of the cantilever was gradually increased as cardiomyocytes slowly retain its inherent properties. The displacement of the cantilever after 50 and 55 minutes of drug washed out time were $2.61 \pm 0.032 \mu\text{m}$ and $2.78 \pm 0.059 \mu\text{m}$, respectively. After 70 minutes the cantilever displacement was close to control state as the cardiomyocytes completely refreshed.

The effect of Verapamil on the beating rate of cardiomyocytes was investigated at a regular time interval (Figure 8c). The results showed a decrease in beating rate with increasing drug treatment time. The beating rate of 1 μ M Verapamil treated cardiomyocytes after 40 minutes of drug treatment time decreased $\sim 50\%$, from 1 Hz to 0.5 Hz. After 70 minutes of drug washed out, the cardiomyocytes were restored its original beating rate. The relative contraction force of the cardiomyocytes after 50 minutes of drug treatment time decreased 46% from $107 \pm 2.056 \text{ nN}$ (control state) to $57.1 \pm 1.931 \text{ nN}$ (Figure 8d). The relative contraction force of the cardiomyocytes after 50 minutes of drug washed out was reached 93% of its original value ($98.2 \pm 2.102 \text{ nN}$). After 70 minutes of drug washed out the relative contraction force of cardiomyocytes was nearly similar to the control state.

The gradual decrease in contraction force of cardiomyocytes could be due to the slow diffusion of Verapamil into the cell wall. The decrease in contraction force of cardiomyocytes could be explained based on the following lines. Verapamil is an L-type Ca^{2+} channel blocker and prevents calcium

influence to prolong atrioventricular node effective refractory period thus reducing the contraction force of cardiomyocytes. Furthermore, increasing drug treatment time yielded a significant increase in the rise time of cardiomyocytes indicative of a decrease in contractile force generation of cardiomyocytes (Figure 8e). However, no significant change in the decay time of cardiomyocytes was observed before and after the drug washed out (Figure 8f). The long-term measurement analysis demonstrates that the proposed crack-based sensor not only measure the displacement of the cantilever but also detect various unusual contractility of cardiomyocytes which was attributed to the adverse effects of Verapamil.

Page no: 15. (Figure 5) Revised version of the manuscript (Figure S13 in the previous version of the manuscript).

Figure 5. a, Real-time measurement shows the contractile force and changes in contractility due to the maturation of cardiomyocytes. **b,c**, Relative contraction force and beating rate of the cardiomyocytes at different maturation day. Error bars are mean \pm s.d., n=5.

Page no: 15. Revised version of the manuscript.

Figure 5a shows the real-time traces of the cantilever displacement measured over 34 days of the culture period. The cultured cardiomyocytes showed measurable beat rate at an early stage of culture (4th) period. The displacement of the cantilever was increased with increasing culture period and reached highest on day 22 (Figure 5b). The contraction force of cardiomyocytes was gradually increased with increasing cell culture day and reaches maximum value on day 22, then, slowly decreased and reaches lowest value on day 34 (Figure 5b). The decrease in cantilever displacement can be explained based on the following lines. The part of the cultured cardiomyocytes on the silicone rubber cantilever was detached and aggregated with increasing incubation period and therefore, not transmitting the resultant contractile force efficiently to the sensor. Similarly, the beating rate of the cultured cardiomyocytes was decreased with increasing culture period, indicative of increased maturity of cultured cardiomyocytes.

Page no: 13. (Figure 4) Revised version of the manuscript (Figure 5 in the previous version of the manuscript).

Figure 4. a,b, Representative traces of real-time change in resistance ratio and displacement of the cardiomyocytes cultured PDMS-encapsulated crack sensor at different culture day. (D is representing displacement)

Figure S15. **a**, Displacement of the cardiomyocytes seeded PDMS encapsulated crack sensor at different culture period. **b,c**, Relative contraction force and a beat rate of the cardiomyocytes at different culture period. **d**, Rise time and decay time of cultured cardiomyocytes measured over 26 days. Error bars are mean \pm s.d. (n=5).

Figure 6. **a**, Representative real-time traces of change in sensor resistance ratio and cantilever displacement owing to the contraction and relaxation of different concentration of Verapamil treated cardiomyocytes. **b, c** Bar plot depicting the cantilever displacement and relative contraction force generated by cardiomyocytes ($n=5$, ** $p < 0.01$ measures by one-way ANOVA followed by Tukey's honest significant difference test, error bars are mean \pm s.d.). **d**, Beat rate of cardiomyocytes at Verapamil concentrations of 0.01 nM and 1 μM ($n=5$). Error bars are mean \pm s.d. **e**, Beat rate of Verapamil treated cardiomyocytes at different concentrations (0.01 nM to 10 μM) ($n=5$). Error bars are mean \pm s.d. **f**, Change in resistance of the cardiomyocytes seeded cantilever at different Verapamil concentrations (0.01 nM to 10 μM) ($n=5$). Error bars are mean \pm s.d. **g**, Rise time and decay time of cardiomyocytes at Verapamil concentrations of 0.01 nM and 1 μM ($n=5$, ** $p < 0.01$ measures by one-way ANOVA followed by Tukey's honest significant difference test, error bars are mean \pm s.d.). (D is representing displacement).

Figure 7. **a**, Representative real-time traces of change in sensor resistance ratio and cantilever displacement owing to the contraction and relaxation of different concentration of Quinidine treated cardiomyocytes. **b**, **c** Bar plot depicting the cantilever displacement and relative contraction force generated by cardiomyocytes ($n=5$, $** p < 0.01$ measures by one-way ANOVA followed by Tukey's honest significant difference test, error bars are mean \pm s.d.). **d**, Beat rate of cardiomyocytes at Quinidine concentrations of 1 nM and 10 μ M ($n=5$). Error bars are mean \pm s.d. **e**, Beat rate of Quinidine treated cardiomyocytes at different concentrations (1 nM to 100 μ M) ($n=5$). Error bars are mean \pm s.d. **f**, Change in resistance of the cardiomyocytes seeded cantilever at different Quinidine concentrations (1 nM to 100 μ M) ($n=5$). Error bars are mean \pm s.d. **g**, Rise time and decay time of cardiomyocytes at Quinidine concentrations of 1 nM and 10 μ M ($n=5$, $** p < 0.01$ measures by one-way ANOVA followed by Tukey's honest significant difference test, error bars are mean \pm s.d.). (D is representing displacement).

Figure S21. a, Representative real-time traces of change in sensor resistance ratio and cantilever displacement owing to the contraction and relaxation of different concentration of Isoproterenol treated cardiomyocytes. b, c, Bar plot depicting the cantilever displacement and relative contraction force generated by cardiomyocytes ($n=5$, $** p < 0.01$ measures by one-way ANOVA followed by Tukey's honest significant difference test, error bars are mean \pm s.d.). d, Beat rate of cardiomyocytes at Isoproterenol concentrations of 1 nM and 1 μ M ($n=5$). Error bars are mean \pm s.d. e, Beat rate of Isoproterenol treated cardiomyocytes at different concentrations (1 nM to 2 μ M) ($n=5$). Error bars are mean \pm s.d. f, Change in resistance of the cardiomyocytes seeded cantilever at different Isoproterenol concentrations (1 nM to 2 μ M) ($n=5$). Error bars are mean \pm s.d. g, Rise time and decay time of cardiomyocytes at Isoproterenol concentrations of 1 nM and 1 μ M ($n=5$, $** p < 0.01$ measures by one-way ANOVA followed by Tukey's honest significant difference test, error bars are mean \pm s.d.). (D is representing displacement).

Reviewer #2

Summary Recommendation: The authors have done a great job in responding to the reviewer critiques, adding many new experiments. The manuscript will now be suitable for publication in Nature Communications, with minor revisions that include the addition of new analysis of their existing data, and additional pharmacology experiments. Minor experimental criticisms to be addressed:

Response: We sincerely thank the reviewer for his appreciation of our revised manuscript. The reviewer's words of encouragement and an unequivocally strong support surely boosts us to continue our research efforts with a lot of confidence. We also thank the reviewer for his thoughtful review of the manuscript. The reviewer raises important issues and his inputs are very helpful for further improving the quality of the manuscript. We have made our best efforts to address all the reviewer's comments in the revised version of the manuscript. The details of the changes made in revised version of the manuscript are appended in the response letter for the reviewer's kind consideration. We sincerely believe that the modified manuscript justifies our conclusions and get back reviewer's confidence and opportunity to publish our work in this esteemed journal "Nature Communications"

Comment (1). More quantitative data and statistical analysis is needed. Both for analysis of the device itself (Figure 3 and 4) and the biologic characterization (Figure 5-8), the authors need to provide more experimental data to test the reproducibility of their measurements. For example, in Figure 3f, there is a small change in R/R_0 with temperature. This relative change needs to be quantified, with statistical analysis (standard deviation of the response across multiple devices). Same to be done for gauge factor in Figure 4e and relationship between resistance and average displacement in 4c-d.

Response: As recommended by the reviewer, we have carried out the additional experimental analysis to demonstrate the reproducibility of the sensor and biologic characteristics. The change in resistance with respect to humidity, temperature and stability as a function of time were measured by using five different crack sensors (non-encapsulated and PDMS encapsulated) and all the obtained data are expressed as mean \pm s.d. for at least five independent experiments. The significance levels were set at * $p < 0.05$ and ** $p < 0.01$.

Figure 2. Changes of resistance under different operating conditions for two different cantilevers with and without PDMS encapsulation. **a-c**, Performances of the crack sensors as a function of humidity changes (45–95 %). **d-f**, Effect of temperature. **g-i**, Stability of the PDMS-encapsulated crack sensor in a stage-top bioreactor at 37 °C. **c, f and i** Error bars are mean \pm s.d. (n=5, * P < 0.05, ** P < 0.01).

Figure 3. a,b, Optical and SEM images of the PDMS-encapsulated crack sensor and Au strain sensor integrated on a silicone rubber cantilever. c,d, Change in sensor resistance ratio of the proposed crack sensor and Au strain sensor as a function of displacement with the ramp of 10 μm. e, Averaged gauge factor of the crack sensor at different applied strain. c-e, Error bars are mean ± s.d. (n=5). (D is representing displacement).

Figure S7. a,b, Representative traces of real-time change in resistance of the proposed crack sensor and Au strain sensor at 100 μm displacement.

Figure 4. a,b, Representative traces of real-time change in resistance ratio and displacement of the cardiomyocytes cultured PDMS-encapsulated crack sensor at different culture day. (D is representing displacement)

Figure S15. **a**, Displacement of the cardiomyocytes seeded PDMS encapsulated crack sensor at different culture period. **b,c**, Relative contraction force and a beat rate of the cardiomyocytes at different culture period. **d**, Rise time and decay time of cultured cardiomyocytes measured over 26 days. Error bars are mean \pm s.d. (n=5).

Figure 5. a, Real-time measurement shows the contractile force and changes in contractility due to the maturation of cardiomyocytes. b,c, Relative contraction force and beating rate of the cardiomyocytes at different maturation day. Error bars are mean \pm s.d., n=5.

Figure 6. **a**, Representative real-time traces of change in sensor resistance ratio and cantilever displacement owing to the contraction and relaxation of different concentration of Verapamil treated cardiomyocytes. **b, c** Bar plot depicting the cantilever displacement and relative contraction force generated by cardiomyocytes (n=5, ** $p < 0.01$ measures by one-way ANOVA followed by Tukey's honest significant difference test, error bars are mean \pm s.d.). **d**, Beat rate of cardiomyocytes at Verapamil concentrations of 0.01 nM and 1 μM (n=5). Error bars are mean \pm s.d. **e**, Beat rate of Verapamil treated cardiomyocytes at different concentrations (0.01 nM to 10 μM) (n=5). Error bars are mean \pm s.d. **f**, Change in resistance of the cardiomyocytes seeded cantilever at different Verapamil concentrations (0.01 nM to 10 μM) (n=5). Error bars are mean \pm s.d. **g**, Rise time and decay time of cardiomyocytes at Verapamil concentrations of 0.01 nM and 1 μM (n=5, ** $p < 0.01$ measures by one-way ANOVA followed by Tukey's honest significant difference test, error bars are mean \pm s.d.). (D is representing displacement).

Figure 7. **a**, Representative real-time traces of change in sensor resistance ratio and cantilever displacement owing to the contraction and relaxation of different concentration of Quinidine treated cardiomyocytes. **b**, **c** Bar plot depicting the cantilever displacement and relative contraction force generated by cardiomyocytes ($n=5$, $** p < 0.01$ measures by one-way ANOVA followed by Tukey's honest significant difference test, error bars are mean \pm s.d.). **d**, Beat rate of cardiomyocytes at Quinidine concentrations of 1 nM and 10 μ M ($n=5$). Error bars are mean \pm s.d. **e**, Beat rate of Quinidine treated cardiomyocytes at different concentrations (1 nM to 100 μ M) ($n=5$). Error bars are mean \pm s.d. **f**, Change in resistance of the cardiomyocytes seeded cantilever at different Quinidine concentrations (1 nM to 100 μ M) ($n=5$). Error bars are mean \pm s.d. **g**, Rise time and decay time of cardiomyocytes at Quinidine concentrations of 1 nM and 10 μ M ($n=5$, $** p < 0.01$ measures by one-way ANOVA followed by Tukey's honest significant difference test, error bars are mean \pm s.d.). (D is representing displacement).

Figure S21. **a**, Representative real-time traces of change in sensor resistance ratio and cantilever displacement owing to the contraction and relaxation of different concentration of Isoproterenol treated cardiomyocytes. **b, c**, Bar plot depicting the cantilever displacement and relative contraction force generated by cardiomyocytes ($n=5$, $** p < 0.01$ measures by one-way ANOVA followed by Tukey's honest significant difference test, error bars are mean \pm s.d.). **d**, Beat rate of cardiomyocytes at Isoproterenol concentrations of 1 nM and 1 μ M ($n=5$). Error bars are mean \pm s.d. **e**, Beat rate of Isoproterenol treated cardiomyocytes at different concentrations (1 nM to 2 μ M) ($n=5$). Error bars are mean \pm s.d. **f**, Change in resistance of the cardiomyocytes seeded cantilever at different Isoproterenol concentrations (1 nM to 2 μ M) ($n=5$). Error bars are mean \pm s.d. **g**, Rise time and decay time of cardiomyocytes at Isoproterenol concentrations of 1 nM and 1 μ M ($n=5$, $** p < 0.01$ measures by one-way ANOVA followed by Tukey's honest significant difference test, error bars are mean \pm s.d.). (D is representing displacement).

Figure 8. **a**, Real-time traces show the cell contractility and beat rate of cardiomyocytes treated with 1 μM verapamil. **b,c**, Displacement of the cantilever after treating the cardiomyocytes with 1 μM verapamil. Error bars are representing mean \pm s.d., $n=5$, * $P < 0.05$, ** $P < 0.01$ measures by one-way ANOVA followed by Tukey's honest significant difference test. **d**, Relative contraction force generated by 1 μM verapamil treated cardiomyocytes at different drug treatment time. Error bars are representing mean \pm s.d., $n=5$, * $P < 0.05$, ** $P < 0.01$ measures by one-way ANOVA followed by Tukey's honest significant difference test. **e,f**, Rise time and decay time of the 1 μM verapamil treated cardiomyocytes. Error bars are representing mean \pm s.d., $n=5$, ** $P < 0.01$ measures by one-way ANOVA followed by Tukey's honest significant difference test.

Figure S23. Representative immunofluorescence of cardiomyocytes cultured on PDMS encapsulated crack sensor integrated silicone rubber cantilever. **a**, Maturation of the cardiomyocytes investigated on day 10 and 22 of the culture day. Green: α -actinin, Red: Vinculin. **b**, Sarcomere length of cardiomyocytes on day 10 and 22 of culture day. Error bars are mean \pm s.d., ($n = 10$), * $p < 0.05$. **c**, Normalized displacement of the cantilever owing to the contraction force of cardiomyocytes on day 10 and 22 of the culture day. Error bars are mean \pm s.d., ($n = 5$), ** $p < 0.01$. **d**, mRNA expression (MHC 6, α -actinin) in cardiomyocytes on day 10 and 22 of the culture period. Error bars are mean \pm s.d., ($n = 5$), ** $p < 0.01$. **e**, Change in cTnT expression of the cardiomyocytes before and after drug treatment on day 10 and 22 of the culture period. Error bars are mean \pm s.d., ($n = 5$), ** $p < 0.01$ measures by two-way ANOVA followed by Tukey's honest significant difference test. **f**, Change in β -actin expression of the cardiomyocytes on day 10 and 22 of the culture day before and after drug treatment. Error bars are mean \pm s.d., ($n = 5$), ** $p < 0.01$ measures by two-way ANOVA followed by Tukey's honest significant difference test.

Comment (2). For additional statistical analysis of mechanical contractility data: displacement (in μm), relative force generated by the cell layer, beat-rate, and contraction kinetics (time constants related to time required to go from baseline to maximum force, and to decay from maximum to resting force) should all be provided, with statistical analysis, in Figures 5 and 6. Multiple biologic replicates should be used to produce error bars describing the IC₅₀ values calculated for each drug. Kinetics of force development and relaxation can be corrected to beat-rate using appropriate means (for example Fridericia's method). Quinidine is causing early after depolarizations (EAD) in figure 6; the relative fraction of EADs should be quantified (either in the figure or the text). Importantly, these different metrics of contractility are likely to provide different IC₅₀ values, because they describe different biologic processes that are being inhibited by the drugs in the cells. Having provided these data, the authors can describe which processes are most likely to be inhibited, and discuss the ability to distinguish this as an advantage of their system.

Response: We sincerely thank the reviewer for his valuable suggestions. As recommended by the reviewer, we have provided the displacement, relative force, beat rate and rise time and decay time with statistical analysis for Figures 5, 6, 7, 8 and 9 in the revised version of the manuscript. The early after depolarizations (EAD) of cardiomyocytes owing to the Quinidine is now quantified in the revised version of the manuscript. We have also performed the multiple biologic replicates to produce error bars describing the IC₅₀ values calculated for Verapamil, Quinidine and Isoproterenol.

Figure 4. a,b, Representative traces of real-time change in resistance ratio and displacement of the cardiomyocytes cultured PDMS-encapsulated crack sensor at different culture day. (D is representing displacement)

The long-term durability of crack sensor is critical for in vitro assays. Figure 4 a, b shows the change in resistance ratio and corresponding displacements of two PDMS-encapsulated crack sensors with different cardiomyocytes densities. The optical images of the cultured cardiomyocytes and real time traces of the cantilever owing to the contractility of cardiomyocytes is shown in figure S13 and S14. As the density of cell increases, the displacement of cantilever increases, and change in resistance increases accordingly. As observed the measurement of cantilever displacement can be performed stably for 11 days in culture medium even at a displacement of 5 μm or less (Figure 4 (a) and Figure S15 (a)).

Figure 4b shows the real-time traces of contraction and relaxation characteristics of cardiomyocytes measured between day 17 and 26 using the cantilever with higher cell density. The cantilever displacement owing to the relative contraction force generated by cardiomyocytes increased with increasing culture period (Figure S15 (a, b)). Besides, the beating rate and rise time of

cardiomyocytes significantly decreased at higher culture period (Figure S15 (c, d)) indicative of more maturation of cardiomyocytes. Figure S16 shows the changes in resistance over 26 days after culturing the cardiomyocytes on the proposed crack-based cantilever sensors. The actual displacement of cantilevers owing to the contraction force of cardiomyocytes was further confirmed by using a laser vibrometer. The beating rate of cardiomyocytes was very fast (averaging 4 Hz) at an early stage and then stabilized after 5 days of incubation. Whenever the culture medium was changed every three days after cell seeding, temporary changes in temperature induced a fast-beating rate (5 Hz or more) and an unstable beating for a certain period. This abnormal beating of cardiomyocytes in contraction was normalized by stabilizing the temperature of the culture media. The fabricated crack sensor responded very quickly even at a rapid beating rate of ~ 6 Hz (systolic 82 ms and diastolic 92 ms). The crack-based cantilever sensor showed a significantly stable output for 26 days even for abnormal beating due to changes in the external environment (Figure S17). Most importantly, no change in GF due to fatigue fracture of the PDMS-encapsulated crack sensor was observed even after 5 million instances of repeated operation. An array of crack sensors was also fabricated and evaluated to validate the possibility of high-efficiency drug toxicity screening capability. The repeatability and reproducibility of crack sensor arrays under cell culture medium are shown in Figure S18 (a) and (b), respectively.

Page no: 13. Revised version of the manuscript.

Figure S15. **a**, Displacement of the cardiomyocytes seeded PDMS encapsulated crack sensor at different culture period. **b,c**, Relative contraction force and a beat rate of the cardiomyocytes at different culture period. **d**, Rise time and decay time of cultured cardiomyocytes measured over 26 days. Error bars are mean \pm s.d. (n=5).

Figure 6. **a**, Representative real-time traces of change in sensor resistance ratio and cantilever displacement owing to the contraction and relaxation of different concentration of Verapamil treated cardiomyocytes. **b, c** Bar plot depicting the cantilever displacement and relative contraction force generated by cardiomyocytes ($n=5$, $** p < 0.01$ measures by one-way ANOVA followed by Tukey's honest significant difference test, error bars are mean \pm s.d.). **d**, Beat rate of cardiomyocytes at Verapamil concentrations of 0.01 nM and 1 μ M ($n=5$). Error bars are mean \pm s.d. **e**, Beat rate of Verapamil treated cardiomyocytes at different concentrations (0.01 nM to 10 μ M) ($n=5$). Error bars are mean \pm s.d. **f**, Change in resistance of the cardiomyocytes seeded cantilever at different Verapamil concentrations (0.01 nM to 10 μ M) ($n=5$). Error bars are mean \pm s.d. **g**, Rise time and decay time of cardiomyocytes at Verapamil concentrations of 0.01 nM and 1 μ M ($n=5$, $** p < 0.01$ measures by one-way ANOVA followed by Tukey's honest significant difference test, error bars are mean \pm s.d.). (D is representing displacement).

After verifying that the cantilever device integrated with the PDMS-encapsulated crack sensor could operate in culture media for a long time, further studies on drugs that affect contractility and beating rate in vitro were conducted based on these preliminary experiments. First, cardiac contractility was measured using the integrated crack sensor as a function of various concentrations of Verapamil. A sudden change in contraction force was observed at a drug concentration of $\sim 1 \mu$ M (Figure 6 and Figure S20 (a)). The displacement, relative contraction force generated by cardiomyocytes and

beating rate of cultured cardiomyocytes were gradually decreased with increasing Verapamil concentration (Figure 6 (b-d)). Figure 6e shows the beating rate of Verapamil treated cardiomyocytes at a different concentration ranging from 0.01 nM to 1 μ M. The cardiomyocytes showed a negative inotropic response to Verapamil, with an IC_{50} of 3.44×10^{-8} M. Increasing doses of Verapamil caused a negative chronotropic effect in cardiomyocytes with an IC_{50} of 9.94×10^{-7} M (Figure 6f), resembling studies based on NRVM tissue [18]. Furthermore, increasing doses of Verapamil yield a significant increase in the rise time of cardiomyocytes, indicative of the decrease in force generation of cardiomyocytes at higher Verapamil concentration (Figure 6g).

- [18] Kim, D. S., Jeong, Y. J., Lee, B. K., Shanmugasundaram, A., Lee, D. W. Piezoresistive sensor-integrated PDMS cantilever: A new class of device for measuring the drug-induced changes in the mechanical activity of cardiomyocytes. *Sens. Actuators, B.* **240**, 566-572 (2017).

Figure 7. **a**, Representative real-time traces of change in sensor resistance ratio and cantilever displacement owing to the contraction and relaxation of different concentration of Quinidine treated cardiomyocytes. **b**, **c** Bar plot depicting the cantilever displacement and relative contraction force generated by cardiomyocytes ($n=5$, $** p < 0.01$ measures by one-way ANOVA followed by Tukey's honest significant difference test, error bars are mean \pm s.d.). **d**, Beat rate of cardiomyocytes at Quinidine concentrations of 1 nM and 10 μ M ($n=5$). Error bars are mean \pm s.d. **e**, Beat rate of Quinidine treated cardiomyocytes at different concentrations (1 nM to 100 μ M) ($n=5$). Error bars are mean \pm s.d. **f**, Change in resistance of the cardiomyocytes seeded cantilever at different Quinidine concentrations (1 nM to 100 μ M) ($n=5$). Error bars are mean \pm s.d. **g**, Rise time and decay time of cardiomyocytes at Quinidine concentrations of 1 nM and 10 μ M ($n=5$, $** p < 0.01$ measures by one-way ANOVA followed by Tukey's honest significant difference test, error bars are mean \pm s.d.). (D is representing displacement).

The adverse effects of Quinidine on the cultured cardiomyocytes was investigated using integrated crack sensor (Figure 7 and S20b). As observed the contraction force of cardiomyocytes decreased with increasing the concentration of Quinidine. Furthermore, the early afterdepolarizations (EAD)

in cardiomyocytes was observed when treating the cardiomyocytes at 10 μ M Quinidine. The relative fraction of EAD is defined as the ratio of the number of EAD beats to the regular beats in one minute. The quantified relative fraction of EAD in cardiomyocytes induced by Quinidine (10 μ M) was found to be $\sim 0.79 \pm 0.11$. A decrease in cantilever displacement with increasing Quinidine concentration could be attributed to the decrease in relative contraction force produced by cardiomyocytes (Figure 7b, c). The beating rate of the cardiomyocytes was rapidly decreased at higher concentration (100 μ M) of Quinidine. Furthermore, the Quinidine treated cardiomyocytes showed a negative inotropic effect with an IC_{50} of 9.77×10^{-6} M. Increasing the concentration of Quinidine caused the Sigmoidal decrease in the change in resistance of the cantilever with an IC_{50} of 1.42×10^{-5} M, signifying negative chronotropic response of cardiomyocytes to Quinidine. Treating the cardiomyocytes with 1 nM to 10 μ M of Quinidine displayed apparent changes in the rise time of cardiomyocytes. Whereas, the decay time of cardiomyocytes was prolonged at higher Quinidine concentration (10 μ M) owing to EAD in cardiomyocytes (Figure 7g).

Figure S21. a, Representative real-time traces of change in sensor resistance ratio and cantilever displacement owing to the contraction and relaxation of different concentration of Isoproterenol treated cardiomyocytes. b, c, Bar plot depicting the cantilever displacement and relative contraction force generated by cardiomyocytes ($n=5$, $** p < 0.01$ measures by one-way ANOVA followed by Tukey's honest significant difference test, error bars are mean \pm s.d.). d, Beat rate of cardiomyocytes at Isoproterenol concentrations of 1 nM and 1 μ M ($n=5$). Error bars are mean \pm s.d. e, Beat rate of Isoproterenol treated cardiomyocytes at different concentrations (1 nM to 2 μ M) ($n=5$). Error bars are mean \pm s.d. f, Change in resistance of the cardiomyocytes seeded cantilever at different Isoproterenol concentrations (1 nM to 2 μ M) ($n=5$). Error bars are mean \pm s.d. g, Rise time and decay time of cardiomyocytes at Isoproterenol concentrations of 1 nM and 1 μ M ($n=5$, $** p < 0.01$ measures by one-way ANOVA followed by Tukey's honest significant difference test, error bars are mean \pm s.d.). (D is representing displacement).

Isoproterenol is a β -1 agonist of adrenergic receptor and produces a positive cardiac inotropic effect. Figure S21a shows the representative real-time traces of change in sensor resistance ratio and cantilever displacement owing to the contraction and relaxation of different concentration of isoproterenol treated cardiomyocytes. Upon increasing the concentration of Isoproterenol to 1 μ M,

the beating pattern of cardiomyocytes became irregular and tachycardia sequence was observed. After isoproterenol treatment, contractile force cardiomyocytes increased by 10%, arrhythmia and a side effect, was observed at drug concentrations exceeding 240 nM, which is known as the EC_{50} value. Further, we observed an increase in cantilever displacement, relative contraction force generated by cardiomyocytes and beating rate of cardiomyocytes in 1 nM to 1 μ M range (Figure S21 (b-d)). In addition, the cardiomyocytes showed positive inotropic response to isoproterenol with an apparent EC_{50} of 5.52×10^{-7} M. The change in resistance of the Isoproterenol treated cardiomyocytes cultured cantilever was showed sigmoidal increase with increasing Isoproterenol concentration, with an EC_{50} value of 6.63×10^{-7} M. The rise time of the Isoproterenol treated cardiomyocyte was decreased with increasing the concentration of Isoproterenol indicative of increase in contraction force generation of cardiomyocytes (Figure S21f).

In brief, a proposed sensor is well studied to identify the gradual changes in the contractile kinetics of cultured cardiomyocytes occurred over the culture period. During this culture period, the contractile force of engineered cardiac tissues increased from day 4 to 22. Finally, we successfully demonstrate that the proposed crack sensor can detect changes in a contractile force of the cardiomyocytes due to adverse effects of cardiac drugs such as Verapamil (Ca^{2+} blocker), Quinidine (Na^+ blocker) and Isoproterenol (β -1 agonist). The cultured cardiomyocytes were showed a negative inotropic and chronotropic response to Verapamil with an IC_{50} of 9.94×10^{-7} M and 3.44×10^{-8} M, respectively (Figure 6). Similarly, the cardiac tissues exhibit a negative inotropic and chronotropic response to Quinidine with an IC_{50} of 1.42×10^{-5} M and 9.77×10^{-6} M, respectively. Furthermore, the device also detects EAD in Quinidine treated cardiomyocytes ($\geq 10 \mu$ M). The potential extension of the QT interval can be verified by mechanical contraction force measurement (Figure 7). The cardiomyocytes showed a positive inotropic (6.63×10^{-7} M) and chronotropic response (5.52×10^{-7} M) to Isoproterenol (Figure S21). Quantitative assessment of contractility measurement allows for physiological analysis of inotropic and chronotropic effect in drug-induced cardiomyocytes. The drug toxicity screening ability of the crack-based cantilever sensor makes it possible to solve the drawbacks of existing optical-based measurement systems and enables accurate data collection through the simple measurement platform. Also, by arranging multiple cantilevers integrated with crack sensors in parallel, we expect to be able to simultaneously analyze various drugs with high efficiency.

Comment (3). One or two positive inotropes (isoproterenol, omecamtiv mecarbil, digoxin) should be included in Figure 6 to demonstrate the ability to sense increase in force.

Response: As recommended by the reviewer, we have measured the effect of isoproterenol on the cultured cardiomyocytes using the proposed crack sensor. The suggested study has been included in the revised version of the manuscript.

Page no: 19. Revised version of the manuscript.

Figure S21. **a**, Representative real-time traces of change in sensor resistance ratio and cantilever displacement owing to the contraction and relaxation of different concentration of Isoproterenol treated cardiomyocytes. **b c**, Bar plot depicting the cantilever displacement and relative contraction force generated by cardiomyocytes ($n=5$, $** p < 0.01$ measures by one-way ANOVA followed by Tukey's honest significant difference test, error bars are mean \pm s.d.). **d**, Beat rate of cardiomyocytes at Isoproterenol concentrations of 1 nM and 1 μ M ($n=5$). Error bars are mean \pm s.d. **e**, Beat rate of Isoproterenol treated cardiomyocytes at different concentrations (1 nM to 2 μ M) ($n=5$). Error bars are mean \pm s.d. **f**, Change in resistance of the cardiomyocytes seeded cantilever at different Isoproterenol concentrations (1 nM to 2 μ M) ($n=5$). Error bars are mean \pm s.d. **g**, Rise time and decay time of cardiomyocytes at Isoproterenol concentrations of 1 nM and 1 μ M ($n=5$, $** p < 0.01$

measures by one-way ANOVA followed by Tukey's honest significant difference test, error bars are mean \pm s.d.). (D is representing displacement).

Isoproterenol is a β -1 agonist of adrenergic receptor and produces a positive cardiac inotropic effect. Figure S21a shows the representative real-time traces of change in sensor resistance ratio and cantilever displacement owing to the contraction and relaxation of different concentration of isoproterenol treated cardiomyocytes. Upon increasing the concentration of Isoproterenol to 1 μ M, the beating pattern of cardiomyocytes became irregular and tachycardia sequence was observed. After isoproterenol treatment, contractile force cardiomyocytes increased by 10%, arrhythmia and a side effect, was observed at drug concentrations exceeding 240 nM, which is known as the EC_{50} value. Further, we observed an increase in cantilever displacement, relative contraction force generated by cardiomyocytes and beating rate of cardiomyocytes in 1 nM to 1 μ M range (Figure S21 (b-d)). In addition, the cardiomyocytes showed positive inotropic response to isoproterenol with an apparent EC_{50} of 5.52×10^{-7} M. The change in resistance of the Isoproterenol treated cardiomyocytes cultured cantilever was showed sigmoidal increase with increasing Isoproterenol concentration, with an EC_{50} value of 6.63×10^{-7} M. The rise time of the Isoproterenol treated cardiomyocyte was decreased with increasing the concentration of Isoproterenol indicative of increase in contraction force generation of cardiomyocytes (Figure S21f).

In brief, a proposed sensor is well studied to identify the gradual changes in the contractile kinetics of cultured cardiomyocytes occurred over the culture period. During this culture period, the contractile force of engineered cardiac tissues increased from day 4 to 22. Finally, we successfully demonstrate that the proposed crack sensor can detect changes in a contractile force of the cardiomyocytes due to adverse effects of cardiac drugs such as Verapamil (Ca^{2+} blocker), Quinidine (Na^+ blocker) and Isoproterenol (β -1 agonist). The cultured cardiomyocytes were showed a negative inotropic and chronotropic response to Verapamil with an IC_{50} of 9.94×10^{-7} M and 3.44×10^{-8} M, respectively (Figure 6). Similarly, the cardiac tissues exhibit a negative inotropic and chronotropic response to Quinidine with an IC_{50} of 1.42×10^{-5} M and 9.77×10^{-6} M, respectively. Furthermore, the device also detects EAD in Quinidine treated cardiomyocytes ($\geq 10 \mu$ M). The potential extension of the QT interval can be verified by mechanical contraction force measurement (Figure 7). The cardiomyocytes showed a positive inotropic (6.63×10^{-7} M) and chronotropic response (5.52×10^{-7} M) to Isoproterenol (Figure S21). Quantitative assessment of contractility measurement

allows for physiological analysis of inotropic and chronotropic effect in drug-induced cardiomyocytes. The drug toxicity screening ability of the crack-based cantilever sensor makes it possible to solve the drawbacks of existing optical-based measurement systems and enables accurate data collection through the simple measurement platform. Also, by arranging multiple cantilevers integrated with crack sensors in parallel, we expect to be able to simultaneously analyze various drugs with high efficiency.

Comment (4). Figure 7 – the same statistics described in #2 above should be provided. Also, the in vitro maturation data is probably the most important piece of data in the paper, and should be in the main figure set, not the supplement. The authors should discuss why they think the force decays after day 22, as one would expect the force either continually increases, or plateaus over time. Are the cells becoming detached from the silicone rubber (and therefore not transmitting force to it efficiently) at these later time-points?

Response: As recommended by the reviewer, we have performed the statistical analysis and accordingly modified the figure 9 (Previously Figure 7) in the revised version of the manuscript. As suggested, the real-time measurement of cardiomyocytes maturation over the culture period moved from supporting information to the main manuscript. We have also briefly discussed the contractile force decay of cardiomyocytes after day 22 of the cultured period in the revised version of the manuscript.

Figure 8. **a**, Real-time traces show the cell contractility and beat rate of cardiomyocytes treated with 1 μM verapamil. **b,c**, Displacement of the cantilever after treating the cardiomyocytes with 1 μM verapamil. Error bars are representing mean ± s.d., n=5, * $P < 0.05$, ** $P < 0.01$ measures by one-way ANOVA followed by Tukey's honest significant difference test. **d**, Relative contraction force generated by 1 μM verapamil treated cardiomyocytes at different drug treatment time. Error bars are representing mean ± s.d., n=5, * $P < 0.05$, ** $P < 0.01$ measures by one-way ANOVA followed by Tukey's honest significant difference test. **e,f**, Rise time and decay time of the 1 μM verapamil treated cardiomyocytes. Error bars are representing mean ± s.d., n=5, ** $P < 0.01$ measures by one-way ANOVA followed by Tukey's honest significant difference test.

The real-time drug screening ability of the proposed silicone rubber cantilever integrated with a PDMS-encapsulated crack sensor was analyzed to detect rare events such as changes in the regularity of contraction force of drug-treated cardiomyocytes. Figure 8a shows the real-time traces of cell contractility and beat rate of cardiomyocytes treated with 1 μM verapamil. The cantilever displacement was measured at regular interval of time after treating cultured cardiomyocytes with a concentration of 1 μM Verapamil (Figure 8b). After 40 minutes of drug treatment, the displacement of the cantilever was decreased by 45.63%, from $2.98 \pm 0.105 \mu\text{m}$ (control state) to $1.62 \pm 0.054 \mu\text{m}$. The drug-treated cardiomyocytes were washed out again and measured the displacement of the cantilever. The displacement of the cantilever was gradually increased as cardiomyocytes slowly retain its inherent properties. The displacement of the cantilever after 50 and 55 minutes of drug washed out time were $2.61 \pm 0.032 \mu\text{m}$ and $2.78 \pm 0.059 \mu\text{m}$, respectively. After 70 minutes the cantilever displacement was close to control state as the cardiomyocytes completely refreshed.

The effect of Verapamil on the beating rate of cardiomyocytes was investigated at a regular time interval (Figure 8c). The results showed a decrease in beating rate with increasing drug treatment time. The beating rate of 1 μM Verapamil treated cardiomyocytes after 40 minutes of drug treatment time decreased $\sim 50\%$, from 1 Hz to 0.5 Hz. After 70 minutes of drug washed out, the cardiomyocytes were restored its original beating rate. The relative contraction force of the cardiomyocytes after 50 minutes of drug treatment time decreased 46% from $107 \pm 2.056 \text{ nN}$ (control state) to $57.1 \pm 1.931 \text{ nN}$ (Figure 8d). The relative contraction force of the cardiomyocytes after 50 minutes of drug washed out was reached 93% of its original value ($98.2 \pm 2.102 \text{ nN}$). After 70 minutes of drug washed out the relative contraction force of cardiomyocytes was nearly similar to the control state.

The gradual decrease in contraction force of cardiomyocytes could be due to the slow diffusion of Verapamil into the cell wall. The decrease in contraction force of cardiomyocytes could be explained based on the following lines. Verapamil is an L-type Ca^{2+} channel blocker and prevents calcium influence to prolong atrioventricular node effective refractory period thus reducing the contraction force of cardiomyocytes. Furthermore, increasing drug treatment time yielded a significant increase in the rise time of cardiomyocytes indicative of a decrease in contractile force generation of cardiomyocytes (Figure 8e). However, no significant change in the decay time of cardiomyocytes was observed before and after the drug washed out (Figure 8f). The long-term measurement analysis

demonstrates that the proposed crack-based sensor not only measure the displacement of the cantilever but also detect various unusual contractility of cardiomyocytes which was attributed to the adverse effects of Verapamil.

Page no: 15. (Figure 5) Revised version of the manuscript (Figure S13 in the previous version of the manuscript).

Figure 5. a, Real-time measurement shows the contractile force and changes in contractility due to the maturation of cardiomyocytes. b,c, Relative contraction force and beating rate of the cardiomyocytes at different maturation day. Error bars are mean \pm s.d., n=5.

Page no: 15. Revised version of the manuscript.

Figure 5a shows the real-time traces of the cantilever displacement measured over 34 days of the culture period. The cultured cardiomyocytes showed measurable beat rate at an early stage of culture (4th) period. The displacement of the cantilever was increased with increasing culture period and reached highest on day 22 (Figure 5b). The contraction force of cardiomyocytes was gradually increased with increasing cell culture day and reaches maximum value on day 22, then, slowly decreased and reaches lowest value on day 34 (Figure 5b). The decrease in cantilever displacement can be explained based on the following lines. The part of the cultured cardiomyocytes on the silicone rubber cantilever was detached and aggregated with increasing incubation period and therefore, not transmitting the resultant contractile force efficiently to the sensor. Similarly, the beating rate of the cultured cardiomyocytes was decreased with increasing culture period, indicative of increased maturity of cultured cardiomyocytes.

Comment-5. The included data on force-frequency behavior should either be included or at least described, i.e. “external pacing at different frequencies suggested these cells display a negative force-frequency relationship.”

Response: As recommended by the reviewer, we have also investigated the effect of external electrical pacing on the contraction force of the cultured cardiomyocytes. The suggested studies are included in the revised version of the manuscript.

Figure S19. a, Force-frequency relationships of cultured cardiomyocytes as a function of time. b, Relative contraction force generated by cardiomyocytes at different applied electrical stimulation frequency. Error bars are mean \pm s.d., n = 5; ** p < 0.01 measures by one-way ANOVA followed

by Tukey's honest significant difference test. (C and NS are representing control and non-significant)

Page no: 14. Revised version of the manuscript.

Force - Frequency Relationship (FFR) is one of the significant factors for heart contraction analysis [28]. Hence, FFR of the cardiomyocytes cultured on PDMS-encapsulated crack sensor was investigated at different external electrical stimulation pacing from 0.5 Hz to 3 Hz (Figure S19). The spontaneous beating of cardiomyocytes was synchronized by the external electrical stimulation. Two carbon electrodes were placed parallel to the longitudinal direction of both sides of silicone rubber cantilever. External electrical stimulation (square-wave pulses, 2 ms, and the electric field was 3 V / cm) was applied to the cardiomyocytes seeded on the cantilever. The cardiomyocytes were paced at 0.5, 1.0, 1.5, 2, 2.5 and 3 Hz and the contractility of cardiomyocytes at each frequency was allowed to stabilize for 1 min. The experimental results show that the beating rate of cardiomyocytes can be precisely synchronized with increasing the electrical stimulation (Figure S19a). The contractile force of cardiomyocytes was compared to those at control state to more closely examine the change in contraction force at various stimulation pacing. The contractility of cardiomyocytes was found to be similar to the control state at 0.5 Hz with no significant difference, and then decreased with further increasing the external electrical pacing. The contractility was ~2-fold decreased at 3 Hz compared to the control state (Figure S19b). The decrease in contractile force or negative FFR of culture cardiomyocytes could be attributed to several factors, such as oxygen limitation in the cardiomyocytes, potential overload due to altered Ca^{2+} and changes in intracellular pH [29, 30].

References

- [28] Endoh, M. Force–frequency relationship in intact mammalian ventricular myocardium: physiological and pathophysiological relevance. *Eur. J. Pharmacol.* **500**, 73-86 (2004).
- [29] Antoons, G., Mubagwa, K., Nevelsteen, I., Sipido, K. R. Mechanisms underlying the frequency dependence of contraction and $[Ca^{2+}]_i$ transients in mouse ventricular myocytes. *The Journal of physiology* **543**, 889-898 (2002).

[30] Maier, L. S., Bers, D. M., Pieske, B. Differences in Ca²⁺-handling and sarcoplasmic reticulum Ca²⁺-content in isolated rat and rabbit myocardium. *J. Mol. Cell. Cardiol.* **32**, 2249-2258 (2000).

Comment-6. On all figures where multiple biological replicates (cell experiments) or devices are analyzed, the authors need to clearly state in figure captions whether error bars reflect SD or SEM, and what the n is. Appropriate statistical tests (two-way t-test to directly compare things, for example, MHC6 expression from day 10 to day 22 in figure 8d, 1-way ANOVA to do multiple comparisons, for example, effects of time on contractility parameters).

Response: As recommended by the reviewer, we have modified all the obtained data and expressed as mean \pm standard deviation (s.d.) for at least five independent experiments. As suggested by the reviewer, all the figure captions are now clearly state the error bars in the revised version of the manuscript. We have also performed appropriate statistical tests and modified the figure 8 in the revised version of the manuscript.

Page no: 23. (Figure S23) Revised version of the manuscript (Figure 8 in the previous version of the manuscript).

To demonstrate the improved performance of the proposed sensing platform, we have performed the traditional gene expression and protein localization analysis. Figure S23 shows the immunofluorescence of cardiomyocytes obtained from the PDMS encapsulated crack sensor integrated silicone rubber cantilever on day 10 and 22 of the culture period. The cardiomyocytes obtained from day 22 are found to be more elongated in the direction of grooves and formed an aligned cell tissue like that seen in vivo compared to that of cardiomyocytes obtained from day 10 (Figure S23a). The sarcomere length of cardiomyocytes on day 22 was found to be $1.78 \pm 0.032 \mu\text{m}$, whereas, the sarcomere length of cardiomyocytes on day 10 was $1.69 \pm 0.048 \mu\text{m}$ (Figure S23b). The displacement of the cantilever on day 10 and 22 were $1.01 \pm 0.021 \mu\text{m}$ and $15.97 \pm 0.232 \mu\text{m}$, respectively (Figure S23c). RT-qPCR of the four genes analysis confirms the improved contractile force (MHC 6, cTnT), higher maturation (α -actinin, β -actin), and improved contractility of cardiomyocytes on day 22 compared to that of cardiomyocytes on day 10. The MHC 6 and α -actinin of cardiomyocytes on day 10 and 22 were 10 ± 0.277 and 206 ± 21.816 , respectively (Figure S23d). The cTnT expression of cardiomyocytes on day 10 was 39.4 ± 4.464 at control state and it decreased

to 29.7 ± 1.804 after treating cardiomyocytes with $1 \mu\text{M}$ Verapamil. Whereas, the cTnT expression of the cardiomyocytes on day 22 was found to be increased to 30.8 ± 0.865 after μM Verapamil treatment compared to 17.1 ± 1.545 at control state (Figure S23e). The β -actin of the cardiomyocytes on day 10 and day 22 of the culture period before and after $1 \mu\text{M}$ Verapamil were 197 ± 13.373 , 286.79 ± 20.169 and 38 ± 5.131 , 221.91 ± 17.581 , respectively (Figure S23e).

Figure S23. Representative immunofluorescence of cardiomyocytes cultured on PDMS encapsulated crack sensor integrated silicone rubber cantilever. **a**, Maturation of the cardiomyocytes investigated on day 10 and 22 of the culture day. Green: α -actinin, Red: Vinculin. **b**, Sarcomere length of cardiomyocytes on day 10 and 22 of culture day. Error bars are mean \pm s.d., ($n = 10$), * $p < 0.05$. **c**, Normalized displacement of the cantilever owing to the contraction force of cardiomyocytes on day 10 and 22 of the culture day. Error bars are mean \pm s.d., ($n = 5$), ** $p < 0.01$. **d**, mRNA expression (MHC 6, α -actinin) in cardiomyocytes on day 10 and 22 of the culture period. Error bars are mean \pm s.d., ($n = 5$), ** $p < 0.01$. **e**, Change in cTnT expression of the cardiomyocytes before and after drug treatment on day 10 and 22 of the culture period. Error bars are mean \pm s.d., ($n = 5$), ** $p < 0.01$ measures by two-way ANOVA followed by Tukey's honest

significant difference test. **f**, Change in β -actin expression of the cardiomyocytes on day 10 and 22 of the culture day before and after drug treatment. Error bars are mean \pm s.d., (n = 5), ** p < 0.01 measures by two-way ANOVA followed by Tukey's honest significant difference test.

Comment (7). Writing changes to be made

1) The concept of “gauge factor” is very important to this publication and it is repeatedly referred to as “GF.” Considering Nature Communications is a general interest journal, the authors need to define GF explicitly before they use the abbreviation.

Response: As recommended by the reviewer, we have defined “gauge factor” of the strain sensor in the revised version of the manuscript.

Page no. 4: Revised version of the manuscript.

Recently, strain sensors have been used to measure the contractile force of cardiomyocytes where sensitivity is determined by the gauge factor (GF) of the strain sensor [17, 18]. In general, the GF is defined as the ratio of relative change in electrical resistance to the mechanical strain ($GF=(\Delta R/R_0)/\epsilon$) of the strain sensor [17].

References

- [17] Lind, J. U. et al. Instrumented cardiac microphysiological devices via multimaterial three-dimensional printing. *Nat. Mater.* **16**, 303 (2017).
- [18] Kim, D. S., Jeong, Y. J., Lee, B. K., Shanmugasundaram, A., Lee, D. W. Piezoresistive sensor-integrated PDMS cantilever: A new class of device for measuring the drug-induced changes in the mechanical activity of cardiomyocytes. *Sens. Actuators, B.* **240**, 566-572 (2017).

2) The authors don't appear to refer to Figures 9 and 10 in the main manuscript – these should be moved to supplemental data. However, even then, the authors need to report what the error bars mean and perform appropriate statistical analysis (Figure 10).

Response: As suggested by the reviewer, we have moved figure 9 (Figure S26 in the revised version of the manuscript) and figure 10 (Figure S30 in the revised version of the manuscript) to the supporting information. We have also added error bars mean and perform appropriate statistical

analysis in figure 10 (Figure S30 in the revised version of the manuscript) and included in the revised version of the manuscript.

Figure S30. Immunocytochemistry staining images of cardiomyocytes cultured on PDMS and silicone rubber substrates. **a–c**, DAPI staining. **d–f**, α -actinin staining and **g–i**, vinculin staining. Error bars are mean \pm s.d., $n=10$, * $P < 0.05$, ** $P < 0.01$.

3) There are still quite a few grammatical errors. I have noted as many as I can, but the authors do need to have this read over carefully (see below).

Response: We sincerely thank the reviewer for his valuable suggestions. We have included all the reviewer's recommendation in the revised version of the manuscript.

Comment (7). Minor changes:

The authors aren't wrong that non-optical techniques are ideal. However, it is not completely true that specialized equipment is needed for optical techniques (in fact, most labs have microscopes with camera capable of going up to 30 frames/second and therefore can do at least rudimentary contractility analysis). I would instead emphasize that with imaging, it is very hard to parallelize measurements – in 96-well plate formats, for example, it is often necessary to image each well one at a time, so that imaging an entire plate can take half an hour or longer.

Response: We sincerely thank the reviewer for his valuable suggestion. As recommended by the reviewer and based on his valuable suggestion, we have modified the part of introduction section in the revised version of the manuscript.

Page no. 3: Revised version of the manuscript

Over the years, several in vitro methods have been proposed for assessing drug-induced cardiac toxicity by measuring the contractile force change of cardiomyocytes [1-7]. Investigating the physiology of the cultured cardiomyocytes by deformation of the sensing platform such as micro-post and/or cantilever [8-13] is considered as a promising method owing to its ease of use, low fabrication cost, high sensitivity and possible to employ for high throughput application. Feinberg et al. reported for the first time using the polydimethylsiloxane (PDMS) thin film for measuring the contraction force of the cardiomyocytes [14]. The deformation of the PDMS thin film owing to the contraction force of the cardiomyocytes was measured by image analysis. Ribeiro et al. quantitatively measured the contractility of a single cardiomyocytes using PDMS micro-post arrays [8]. Although PDMS micro-posts can measure the contractility of single cells, the deformation of the micro-posts caused by cardiac contractility is very small. Moreover, it is not easy to grow cells on top of micro-posts. To overcome the limitations of micro-posts cantilever-type sensor structures were proposed and utilized for the same purpose. Recently, we have proposed various types of cantilever device for investigating the drug-induced changes in the cardiomyocytes [11-13]. Although all these investigations have been intensively studied, the optical-based data acquisition techniques are intrinsically data intensive. Additionally, it is very hard to parallelize measurements in multi-well plate formats as imaging each well is a time-consuming process and it is not convenient to employ for rapidly analyzing drug-induced cardio toxicity effects.

The authors keep using the term “mechanophysiology,” which I am not sure is a real word? Why not just use the more standard term “physiology?”

Response: As recommended by the reviewer, we have changed the term “mechanophysiology” to “physiology” in the revised version of the manuscript.

Page 5: I would note that, in addition to limiting crack sensor success, if the metal layer were exposed to culture medium, leached metal ions would likely be very toxic to cells.

Response: We thank the reviewer for his valuable suggestion. Based on the reviewer’s recommendation we have modified part of the introduction section in the revised version of the manuscript.

Page no. 5: Revised version of the manuscript.

However, to date, there is no experimental demonstration of the crack sensor for measuring the physiology of cardiomyocytes. The limited success of the crack sensor in the biomedical drug screening application could be due to the continuous exposure of the metal layer to the conductive culture medium, which reduces the long-term durability of the sensor. In addition, the leached metal ions are harmful to the cardiomyocyte’s health [24].

[24] Ahn, S., et al. Mussel-inspired 3D fiber scaffolds for heart-on-a-chip toxicity studies of engineered nanomaterials. *Anal. Bioanal. Chem.* **410**, 6141-6154 (2018).

Page 19: the authors mention that the immature cells had randomly oriented α -actinin, but their data (Figure 8a) suggests that the cells are pretty well-aligned in both cases.

Response: We sincerely regret for our oversights. We have removed term mature and immature cells and appropriately modified the description of Figure S23 (previously Figure 8) in the revised version of the manuscript.

Page no: 23. (Figure S23) Revised version of the manuscript (Figure 8 in the previous version of the manuscript).

Figure S23. Representative immunofluorescence of cardiomyocytes cultured on PDMS encapsulated crack sensor integrated silicone rubber cantilever. **a**, Maturation of the cardiomyocytes investigated on day 10 and 22 of the culture day. Green: α -actinin, Red: Vinculin. **b**, Sarcomere length of cardiomyocytes on day 10 and 22 of culture day. Error bars are mean \pm s.d., ($n = 10$)., * $p < 0.05$. **c**, Normalized displacement of the cantilever owing to the contraction force of cardiomyocytes on day 10 and 22 of the culture day. Error bars are mean \pm s.d., ($n = 5$)., ** $p < 0.01$. **d**, mRNA expression (MHC 6, α -actinin) in cardiomyocytes on day 10 and 22 of the culture period. Error bars are mean \pm s.d., ($n = 5$)., ** $p < 0.01$. **e**, Change in cTnT expression of the cardiomyocytes before and after drug treatment on day 10 and 22 of the culture period. Error bars are mean \pm s.d., ($n = 5$)., ** $p < 0.01$ measures by two-way ANOVA followed by Tukey's honest significant difference test. **f**, Change in β -actin expression of the cardiomyocytes on day 10 and 22 of the culture day before and after drug treatment. Error bars are mean \pm s.d., ($n = 5$)., ** $p < 0.01$ measures by two-way ANOVA followed by Tukey's honest significant difference test.

To demonstrate the improved performance of the proposed sensing platform, we have performed

the traditional gene expression and protein localization analysis. Figure S23 shows the immunofluorescence of cardiomyocytes obtained from the PDMS encapsulated crack sensor integrated silicone rubber cantilever on day 10 and 22 of the culture period. The cardiomyocytes obtained from day 22 are found to be more elongated in the direction of grooves and formed an aligned cell tissue like that seen in vivo compared to that of cardiomyocytes obtained from day 10 (Figure S23a). The sarcomere length of cardiomyocytes on day 22 was found to be $1.78 \pm 0.032 \mu\text{m}$, whereas, the sarcomere length of cardiomyocytes on day 10 was $1.69 \pm 0.048 \mu\text{m}$ (Figure S23b). The displacement of the cantilever on day 10 and 22 were $1.01 \pm 0.021 \mu\text{m}$ and $15.97 \pm 0.232 \mu\text{m}$, respectively (Figure S23c). RT-qPCR of the four genes analysis confirms the improved contractile force (MHC 6, cTnT), higher maturation (α -actinin, β -actin), and improved contractility of cardiomyocytes on day 22 compared to that of cardiomyocytes on day 10. The MHC 6 and α -actinin of cardiomyocytes on day 10 and 22 were 10 ± 0.277 and 206 ± 21.816 , respectively (Figure S23d). The cTnT expression of cardiomyocytes on day 10 was 39.4 ± 4.464 at control state and it decreased to 29.7 ± 1.804 after treating cardiomyocytes with $1 \mu\text{M}$ Verapamil. Whereas, the cTnT expression of the cardiomyocytes on day 22 was found to be increased to 30.8 ± 0.865 after μM Verapamil treatment compared to 17.1 ± 1.545 at control state (Figure S23e). The β -actin of the cardiomyocytes on day 10 and day 22 of the culture period before and after $1 \mu\text{M}$ Verapamil were 197 ± 13.373 , 286.79 ± 20.169 and 38 ± 5.131 , 221.91 ± 17.581 , respectively (Figure S23e).

Comment (7). Minor grammatical errors:

Response: We sincerely regret our oversights. We sincerely thank the reviewer for his valuable suggestions. All the suggested modifications are included in the revised version of the manuscript.

Abstract: line 24 – change “directly measures the cardiac contractility” to “directly measures cardiac contractility.”

Response: Complied with the reviewer’s suggestion.

Page no 2. Revised version of the manuscript.

We propose a cantilever device integrated with a polydimethylsiloxane (PDMS)-encapsulated crack sensor that directly measures cardiac contractility.

Page 3, Line 46: change “imaging analysis” to “image analysis”

Response: Complied with the reviewer’s suggestion.

Page no 3. Revised version of the manuscript.

The deformation of the PDMS thin film owing to the contraction force of the cardiomyocytes was measured by image analysis.

Line 51: change “we have proposed the various” to “we have proposed various”

Response: Complied with the reviewer’s suggestion.

Page no 3. Revised version of the manuscript.

Recently, we have proposed various types of cantilever device for investigating the drug-induced changes in the cardiomyocytes [11-13].

Line 60: change “sensor can measure even large” to “sensor can measure large”

Response: Complied with the reviewer’s suggestion.

Page no 3. Revised version of the manuscript.

The GMR sensor can measure large displacement of the post owing to the contraction force of the cardiomyocytes. However, the use of GMR in the high-throughput analysis is limited due to the interference of the adjacent magnetic field.

Page 4, line 80: change “among them, crack sensors” to “among them, crack-based sensors”

Line 82: change “the authors have developed” to “the authors developed”

Response: Complied with the reviewer’s suggestion.

Page no 4. Revised version of the manuscript.

Among them, crack-based sensors have received considerable attention owing to its flexibility, durability, and ultrahigh mechanosensitivity.

The authors developed the crack-based sensor by depositing platinum (Pt) on a poly(urethane acrylate) (PUA) and applying 2% strain to the substrate.

Page 5, Line 92: change “of the highly sensitive crack” to “of a highly sensitive crack”

Line 93: change “rubber cantilever to...in real time” to “rubber cantilever for real-time analysis of drug and maturation induced changes in cardiac contractility.”

Response: Complied with the reviewer’s suggestion.

Page no 5. Revised version of the manuscript.

With this research background and considering the advantages of the crack sensor, herein, we present for the first time the demonstration of a highly sensitive crack sensor integrated silicone rubber cantilever for real-time analysis of drug and maturation induced changes in cardiac contractility.

Line 95: delete “amazingly”. Consider replacing with a less sensationalistic word, like “substantially” or “markedly”

Response: Complied with the reviewer’s suggestion.

Page no 5. Revised version of the manuscript.

The mechanical durability of the crack sensor was substantially improved due to the chemical bonding between the crack sensor and protection layer.

Page 7, Line 135: change “behave very unstably like” to “behave very unstably, like”

Response: Complied with the reviewer’s suggestion.

Page no 7. Revised version of the manuscript.

Crack sensors exposed to culture media behave very unstably, like variable resistors.

Reviewer #3

Summary Recommendation: The queries have been properly addressed.

I am happy that the current version of the manuscript is suitable for publication.

Response: We are thankful to the reviewer for his valuable efforts, time and suggestions to improve the quality of the manuscript. We also sincerely thank the reviewer for recommending our manuscript for publication in this esteemed journal “Nature communication”

Reviewers' Comments:

Reviewer #1:

Remarks to the Author:

Authors have done great work for revising the paper with many additional analyses. I think this manuscript is ready to be published in Nature Communication as it is.

Reviewer #2:

Remarks to the Author:

The authors have responded appropriately to the previous critiques, and the manuscript is ready for acceptance after some very minor revisions to the text. No new experiments are needed, and only two very minor analyses is warranted: 1) all rise and decay times for contractility analysis should be beat rate corrected using Fridericia's or other method; 1-way ANOVA should be done to verify that there is a time-dependent effect on force generation by these cardiomyocytes.

Minor changes noted below:

Figure 2 Caption: change "stage top bioreactor" to "stage top incubator"

Figure 2 – the data on R/R0 show excellent performance of the PDMS encapsulated crack sensor, relative to a non-encapsulated sensor. Nevertheless, they do appear to show that the PDMS crack sensor is not completely impervious to environmentally induced change (ie if the authors compare R to R0 for the sealed condition, it does appear that there would be a significant difference). This indicates that there is still room for future improvement of the sensor and/or the need to ensure that the sensors are properly calibrated under experimental operated conditions, which the authors can note.

Pg. 11, line 184: Change "the home-made stage top incubator" to "a custom stage top incubator"

Figure 4 – Currently, the most important data with this figure are buried in supplemental Figure 15. The authors should simply show the representative R/R0 curves at the first and last time points (day 5 and 25) and move all other representative R/R0 curves to the supplemental data. The data that are now in Supplemental Figure 15 (which do a much better job of reflecting the robustness of the measurement and the type of data that can be obtained) should be moved into Figure 4. 1-way ANOVA should be done on the data that is now Supplemental Figure 15 (to verify that there is a time-dependent change in force, frequency, etc.).

Page 15, lines 283 – 286 – the negative force-frequency relationship (FFR) these authors see is typical for neonatal rodent cells. The ability to measure the force-frequency is very important since if these sensors are eventually applied to human iPSC-derived cardiomyocytes, it could be used to gauge the maturity of these cells. Vunak-Novakovic's group (Ronaldson-Bouchard et al. Nature 2018) has shown that continuous pacing of engineered heart tissue can yield positive FFR. The authors' system will allow continuous monitoring of FFR over the time-course of culture, which would be a useful way to assess future approaches of maturing cardiomyocytes in vitro.

Pg 16 – the detachment of the cardiomyocytes over time may reflect higher static tension (this will also increase, along with contractile force). The detachment of the cells should be emphasized as a limitation of the current system and an opportunity for future improvements.

Figure 6 – the IC50 values for Verapamil should be calculated with standard deviation (this can be done by fitting curves to each technical or biological replicate to get individual IC50, then calculating the mean and standard deviation). These should be reported on the figure, not just in the text.

Manuscript ID: NCOMMS-19-01725B

Title: Highly durable crack sensor integrated with silicone rubber cantilever for measuring cardiac contractility

Author(s): Dong-Su Kim, Yong Whan Choi, Arunkumar Shanmugasundaram, Yun-Jin Jeong, Jongsung Park, Nomin-Erdene Oyunbaatar, Eung-Sam Kim, Mansoo Choi, and Dong-Weon Lee

Point-by-Point Response to the Reviewer's Comments

Reviewer #1:

Remarks to the Author: The authors have done great work for revising the paper with many additional analyses. I think this manuscript is ready to be published in Nature Communication as it is.

Response: We appreciate and thank the reviewer's effort and time to help us improve the quality of the manuscript. We also sincerely thank the reviewer for recommending our manuscript for publication in Nature Communications.

Reviewer #2

Remarks to the Author: The authors have responded appropriately to the previous critiques, and the manuscript is ready for acceptance after some very minor revisions to the text.

Response: We sincerely thank the reviewer for his kind words about our revised manuscript. In the following sections, we made our best efforts to address all the points and suggestions made by the reviewer. We are grateful to the reviewer as we have learned several fundamental and valuable key points from the reviewer's comments during the revision process, and it surely boosts us to continue our ongoing research efforts with a lot of confidence. We appreciate the reviewer for recommending our manuscript for publication in Nature Communications.

Comment-1: No new experiments are needed, and only two very minor analyses is warranted: 1) all rise and decay times for contractility analysis should be beat rate corrected using Fridericia's or other method; 2) 1-way ANOVA should be done to verify that there is a time-dependent effect on force

generation by these cardiomyocytes.

Response: As recommended by the reviewer, all the contractility analysis of the cultured cardiomyocytes is beat corrected using Fridericia's method. 1-way ANOVA was carried out the beat corrected data. The beat corrected contractility data now provided as a supplementary Figure in the revised version of the manuscript.

The rise time and decay time (RD) for contractility analysis of the cultured cardiomyocytes and different cardiovascular drugs treated cardiomyocytes were beat rate corrected (RDc) with Fridericia's formula ($RDc = RD / \text{interspike interval}^{1/3}$). The rise time (Rc) and decay time (Dc) were beat rate corrected with Fridericia's formula ($Rc = R / \text{interspike interval}^{1/3}$) and ($Dc = D / \text{interspike interval}^{1/3}$), respectively. The beat rate corrected data are provided with this response letter for the reviewer's kind consideration.

Figure 4. Long-term stability of the PDMS encapsulated crack sensor integrated with silicone rubber cantilever. **a, b**, Representative traces of real-time change in resistance ratio and displacement of the cardiomyocytes cultured PDMS-encapsulated crack sensor at day 5 and 26 of the culture days. **c**, Displacement of the cardiomyocytes seeded PDMS encapsulated crack sensor integrated with silicone rubber cantilever at different culture periods. **d, e**, Relative contraction force, and a beat rate of the cardiomyocytes at different culture periods. **f**, Rise time and decay time of cultured cardiomyocytes measured over 26 days. **c-e**, (** $p < 0.01$ measures by two-way ANOVA followed by Tukey's honest significant difference test). The rise time and decay time were analyzed, and beat rate corrected with Fridericia's formula (Rise time corrected (Rc) = $R/\text{interspike interval}^{1/3}$) and (Decay time corrected (Dc) = $D/\text{interspike interval}^{1/3}$), respectively. (* $p < 0.05$, ** $p < 0.01$ measures by one-way ANOVA followed by Tukey's honest significant difference test). Error bars are mean \pm s.d. (n=5 biologically independent samples). NS and D are representing displacement and non-significant, respectively.

Figure 6. Change in contractility of the Verapamil treated cultured cardiomyocytes. **a**, Representative real-time traces of change in sensor resistance ratio and cantilever displacement owing to the contraction and relaxation of different concentration of Verapamil treated cardiomyocytes. **b, c** Bar plot depicting the cantilever displacement and relative contraction force generated by cardiomyocytes (** $p < 0.01$ measures by one-way ANOVA followed by Tukey's honest significant difference test). **d**, Beat rate of cardiomyocytes at Verapamil concentrations of 0.01 nM and 1 μM . **e**, Beat rate of Verapamil treated cardiomyocytes at different concentrations (0.01 nM to 10 μM). **f**, Change in resistance of the cardiomyocytes seeded cantilever at different Verapamil concentrations (0.01 nM to 10 μM). **g**, Rise time and decay time of cardiomyocytes at Verapamil concentrations of 0.01 nM and 1 μM . The rise time and decay time were analyzed and beat rate corrected with Fridericia's formula (Rise time corrected (R_c) = $R/\text{interspike interval}^{1/3}$) and (Decay time corrected (D_c) = $D/\text{interspike interval}^{1/3}$), respectively. (** $p < 0.01$ measures by one-way ANOVA followed by Tukey's honest significant difference test). Error bars are mean \pm s.d. (n=5 biologically independent samples). NS and D are representing displacement and non-significant, respectively.

Figure 7. Change in contractility of the Quinidine treated cultured cardiomyocytes. **a**, Representative real-time traces of change in sensor resistance ratio and cantilever displacement owing to the contraction and relaxation of different concentration of Quinidine treated cardiomyocytes. **b, c** Bar plot depicting the cantilever displacement and relative contraction force generated by cardiomyocytes (** $p < 0.01$ measures by one-way ANOVA followed by Tukey's honest significant difference test.). **d**, Beat rate of cardiomyocytes at Quinidine concentrations of 1 nM and 10 μ M. **e**, Beat rate of Quinidine treated cardiomyocytes at different concentrations (1 nM to 100 μ M). **f**, Change in resistance of the cardiomyocytes seeded cantilever at different Quinidine concentrations (1 nM to 100 μ M). **g**, Rise time and decay time of cardiomyocytes at Quinidine concentrations of 1 nM and 10 μ M. The rise time and decay time were analyzed and beat rate corrected with Fridericia's formula (Rise time corrected (Rc) = $R/\text{interspike interval}^{1/3}$) and (Decay time corrected (Dc) = $D/\text{interspike interval}^{1/3}$), respectively. (* $p < 0.05$, ** $p < 0.01$ measures by one-way ANOVA followed by Tukey's honest significant difference test.). Error bars are mean \pm s.d. ($n=5$ biologically independent samples). D is representing displacement.

Figure 8. Real-time traces show the contractility and beat rate of 1 μM Verapamil treated cardiomyocytes. **a**, Displacement of the cantilever owing to the contraction and relaxation of the 1 μM Verapamil treated cardiomyocytes at a different time before and after drug washed out. **b**, **c**, Displacement of the cantilever after treating the cardiomyocytes with 1 μM Verapamil. **d**, Relative contraction force generated by 1 μM Verapamil treated cardiomyocytes at different drug treatment time. **e**, **f**, Rise time, and decay time of the 1 μM Verapamil treated cardiomyocytes. The rise time and decay time were analyzed and beat rate corrected with Fridericia's formula (Rise time corrected (R_c) = $R/\text{interspike interval}^{1/3}$) and (Decay time corrected (D_c) = $D/\text{interspike interval}^{1/3}$), respectively. (* $P < 0.05$, ** $P < 0.01$ measures by one-way ANOVA followed by Tukey's honest significant difference test. Error bars are mean \pm s.d.). (n=5 biologically independent samples).

Supplementary Figure. 22. Effect of Isoproterenol on the cultured cardiomyocytes. **a**, Representative real-time traces of change in sensor resistance ratio and cantilever displacement owing to the contraction and relaxation of different concentrations of Isoproterenol treated cardiomyocytes. **b**, **c**, Bar plot depicting the cantilever displacement and relative contraction force generated by cardiomyocytes (** $p < 0.01$ measures by one-way ANOVA followed by Tukey's honest significant difference test.). **d**, Beat rate of cardiomyocytes at Isoproterenol concentrations of 1 nM and 1 μ M. **e**, Beat rate of Isoproterenol treated cardiomyocytes at different concentrations (1 nM to 2 μ M). **f**, Change in resistance of the cardiomyocytes seeded cantilever at different Isoproterenol concentrations (1 nM to 2 μ M). **g**, Rise time and decay time of cardiomyocytes at Isoproterenol concentrations of 1 nM and 1 μ M. The rise time and decay time were analyzed and beat rate corrected with Fridericia's formula (Rise time corrected (Rc) = $R/\text{interspike interval}^{1/3}$) and (Decay time corrected (Dc) = $D/\text{interspike interval}^{1/3}$), respectively. (** $p < 0.01$ measures by one-way ANOVA followed by Tukey's honest significant difference test.). Error bars are mean \pm s.d. ($n=5$ biologically independent samples). NS and D are representing displacement and non-significant, respectively.

Supplementary Figure. 15. Long-term stability of the PDMS-encapsulated crack sensor integrated with silicone rubber cantilever. **a, b**, Representative traces of real-time change in resistance ratio and displacement of the cardiomyocytes cultured PDMS-encapsulated crack sensor at different culture days. Rise and decay time of cultured cardiomyocytes measured over 26 days. **c**, The rise and decay time (RD) for contractility analysis of the cultured cardiomyocytes were beat rate corrected (RD_c) with Fridericia's formula ($RD_c = RD / \text{interspike interval}^{1/3}$). (Non-significant (NS) and measures by two-way ANOVA followed by Tukey's honest significant difference test.). Error bars are mean \pm s.d. ($n=5$ biologically independent samples). D is representing displacement.

Supplementary Figure. 20. Real-time traces are representing the contractility of the cultured cardiomyocytes. **a**, Experimental results of cardiac contractile force and beating frequency according to Verapamil concentration. **b**, The rise and decay time (RD) for contractility analysis of the cultured cardiomyocytes were beat rate corrected (RDc) with Fridericia's formula ($RDc = RD / \text{interspike interval}^{1/3}$). (Non-significant (NS) and ** $p < 0.01$ measures by one-way ANOVA followed by Tukey's honest significant difference test.). Error bars are mean \pm s.d. ($n=5$ biologically independent samples). D is representing displacement.

Supplementary Figure. 21. Real-time traces are representing the contractility of the cultured cardiomyocytes. **a**, Experimental results of cardiac contractile force and beating frequency according to Quinidine concentration. **b**, The rise and decay time (RD) for contractility analysis of the cultured cardiomyocytes were beat rate corrected (RD_c) with Fridericia's formula (RD_c = RD/interspike interval^{1/3}). (Non-significant (NS) and ** p < 0.01 measures by one-way ANOVA followed by Tukey's honest significant difference test.). Error bars are mean ± s.d. (n=5 biologically independent samples). D is representing displacement.

Supplementary Figure. 24. Rise and decay time of the 1 μ M Verapamil treated cardiomyocytes at different drug treatment time. The rise and decay time (RD) for contractility analysis of the cultured cardiomyocytes were beat rate corrected (RDc) with Fridericia's formula ($RDc = RD / \text{interspike interval}^{1/3}$). (* $P < 0.05$, ** $P < 0.01$ measures by one-way ANOVA followed by Tukey's honest significant difference test. Error bars are mean \pm s.d.). (n=5 biologically independent samples).

Supplementary Figure. 23. Rise and decay time of cardiomyocytes at Isoproterenol concentrations of 1 nM and 1 μ M. The rise time and decay time were analyzed and beat rate corrected with Fridericia's formula (Rise time corrected (Rc) = $R / \text{interspike interval}^{1/3}$) and (Decay time corrected (Dc) = $D / \text{interspike interval}^{1/3}$), respectively. (** $p < 0.01$ measures by one-way ANOVA followed by Tukey's honest significant difference test.). Error bars are mean \pm s.d. (n=5 biologically independent samples). NS is representing non-significant.

Comment-2: Minor changes noted below: Figure 2 Caption: change “stage top bioreactor” to “stage top incubator”

Response: Complied with the reviewer’s recommendations. Figure 2 caption has been modified, as suggested by the reviewer.

Page no 9: Revised version of the manuscript.

Figure 2. Change in resistance of the crack sensors with and without PDMS encapsulation. **a-c**, Performances of the crack sensors as a function of humidity changes (45–95 %). **d-f**, Effect of temperature. **g-i**, Stability of the PDMS-encapsulated crack sensor in a stage top incubator at 37 °C. (NE and PE are representing non-encapsulation and PDMS-encapsulation). **c, f, and i** Error bars are mean \pm s.d. (n=5, * P < 0.05, ** P < 0.01).

Comment-3: Figure 2 – the data on R/R₀ show excellent performance of the PDMS encapsulated crack sensor, relative to a non-encapsulated sensor. Nevertheless, they do appear to show that the PDMS crack sensor is not completely impervious to environmentally induced change (i.e. if the authors compare R to R₀ for the sealed condition, it does appear that there would be a significant difference). This indicates that there is still room for future improvement of the sensor and/or the need to ensure that the sensors are properly calibrated under experimental operated conditions, which the authors can note.

Response: Complied with the reviewer’s suggestion and comment for future improvement is included in a discussion part of the revised manuscript.

Page no 25: Revised version of the manuscript.

The capabilities of the crack-based cantilever sensor proposed were experimentally verified; it can be used for a long time in a solution owing to the PDMS protection layer while also maintaining a high sensitivity. In particular, the chemical bonding of the proposed encapsulation layer, performed to maintain long-term stability in the same environment as the culture medium, showed that it is a very stable encapsulation method that does not affect sensitivity after bonding. Characteristic evaluation of the PDMS-encapsulated sensor in various environments (humidity, temperature, and culture medium) showed the advantage of this crack sensor. However, the PDMS-encapsulated crack sensor is not completely impervious to environmental change. Further research is needed to improve the durability of the PDMS-encapsulated crack sensor integrated silicone rubber cantilever. The proposed sensor can continuously monitor the FFR over the time-course of culture, which would be

a useful way to assess future approaches of maturing cardiomyocytes in vitro. It was possible to stably measure the contractile forces of cardiomyocytes for approximately four weeks in vitro. During the cultivation period, part of the cardiomyocytes was detached from the on the silicone rubber cantilever and not transmitting the resultant contractile force efficiently to the sensor. Therefore, the stable and long-term measurement of contractile force should be improved by reforming the cantilever design. The dose-response studies for Verapamil, Quinidine and Isoproterenol indicate that the toxicity of drugs can be screened in real-time using the proposed crack-based cantilever sensor with a PDMS encapsulation layer. The alignment of the cardiac tissue on the cantilever's longitudinal direction can induce a more significant displacement through the concentration of its contraction force. This can be expected to increase further the GF of the proposed crack sensor, which exhibits exponential behaviors in its resistance changes.

Comment-4: Pg. 11, line 184: Change “the home-made stage top incubator” to “a custom stage top incubator”

Response: Complied with the reviewer's suggestions.

Page no 9: Revised version of the manuscript.

Supplementary Figure 4 and Supplementary Note 4 briefly describes the temperature-dependent deformation of the cantilever composed of the silicone rubber cantilever, silicone rubber, and PDMS analyzed using the commercial finite element analysis program (ANSYS). Supplementary Figure 5 and Supplementary Note 5 describes the displacement change of the cantilever as a function of temperature. In comparison with the simulation results, small differences were negligible, and no further displacement changes of the cantilever were observed when the temperature was kept constant (Supplementary Figure 5b). Change in resistance of the PDMS-encapsulated crack sensor at different temperatures and displacement of the PDMS-encapsulated crack sensor integrated silicone rubber cantilever as a function of time **under a custom stage top incubator** at 40 °C are shown in Supplementary Figure 6 and Supplementary Note 6. The temperature characteristics of the crack sensor, according to encapsulation are shown in Figure 2(d-f). For the non-encapsulated crack sensor, the resistance change rate was ~170% at 35 °C. However, the PDMS-encapsulated crack sensor exhibited relatively stable changes in resistance with a change of ~120% at 35 °C. This can be explained by the fact that the crack sensor was protected from sudden temperature changes by using a polymer with a low thermal conductivity coefficient. Besides, because the proposed crack-based cantilever sensor used for measuring the contractility of

cardiomyocytes is to be used in an incubator environment in which temperature and humidity are maintained, changes in the initial resistance can be neglected (Figure 2(g-i)). The non-encapsulated crack sensor initial resistance value increased ~four times as the temperature increased from room temperature to 38 °C, as required by the stage top incubator. Exposing this crack sensor to electrolyte solutions had a significant impact on its reliability because its resistance value changed rapidly even in the culture media with a constant temperature. In contrast, the PDMS-encapsulated crack sensor saturated with a resistance change similar to that of merely increasing the temperature in air. The hydrophobic characteristics and mechanical sealing effect of the PDMS-encapsulation layer prevent the penetration of the culture media into the cracks even after the operating temperature was increased. Therefore, the crack-based cantilever sensor with the chemically bonded PDMS-encapsulation layer exhibited improved durability in a variety of environments in terms of humidity, temperature, and culture media.

Comment-5: Figure 4 – Currently, the most important data with this figure are buried in supplemental Figure 15. The authors should simply show the representative R/R_0 curves at the first and last time points (day 5 and 25) and move all other representative R/R_0 curves to the supplemental data. The data that are now in Supplemental Figure 15 (which do a much better job of reflecting the robustness of the measurement and the type of data that can be obtained) should be moved into Figure 4. 1-way ANOVA should be done on the data that is now Supplemental Figure 15 (to verify that there is a time-dependent change in force, frequency, etc.).

Response: Figure 4 and Supplementary Figure 15 have been modified based on the reviewer's suggestions.

Figure 4. Long-term stability of the PDMS encapsulated crack sensor integrated with silicone rubber cantilever. **a, b**, Representative traces of real-time change in resistance ratio and displacement of the cardiomyocytes cultured PDMS-encapsulated crack sensor at day 5 and 26 of the culture days. **c**, Displacement of the cardiomyocytes seeded PDMS encapsulated crack sensor integrated with silicone rubber cantilever at different culture periods. **d, e**, Relative contraction force, and a beat rate of the cardiomyocytes at different culture periods. **f**, Rise time and decay time of cultured cardiomyocytes measured over 26 days. **c-e**, (** $p < 0.01$ measures by two-way ANOVA followed by Tukey's honest significant difference test). The rise time and decay time were analyzed, and beat rate corrected with Fridericia's formula (Rise time corrected (Rc) = $R/\text{interspike interval}^{1/3}$) and (Decay time corrected (Dc) = $D/\text{interspike interval}^{1/3}$), respectively. (* $p < 0.05$, ** $p < 0.01$ measures by one-way ANOVA followed by Tukey's honest significant difference test). Error bars are mean \pm s.d. (n=5 biologically independent samples). NS and D are representing displacement and non-significant, respectively.

Supplementary Figure. 15. Long-term stability of the PDMS-encapsulated crack sensor integrated with silicone rubber cantilever. **a, b**, Representative traces of real-time change in resistance ratio and displacement of the cardiomyocytes cultured PDMS-encapsulated crack sensor at different culture days. Rise and decay time of cultured cardiomyocytes measured over 26 days. **c**, The rise and decay time (RD) for contractility analysis of the cultured cardiomyocytes were beat rate corrected (RD_c) with Fridericia's formula ($RD_c = RD / \text{interspike interval}^{1/3}$). (Non-significant (NS) and measures by two-way ANOVA followed by Tukey's honest significant difference test.). Error bars are mean \pm s.d. (n=5 biologically independent samples). D is representing displacement.

Comment-6: Page 15, lines 283 – 286 – the negative force-frequency relationship (FFR) these authors see is typical for neonatal rodent cells. The ability to measure the force-frequency is very important since if these sensors are eventually applied to human iPSC-derived cardiomyocytes, it could be used to gauge the maturity of these cells. Vunak-Novakovic's group (Ronaldson-Bouchard et al. Nature 2018) has shown that continuous pacing of engineered heart tissue can yield positive FFR. The authors' system will allow continuous monitoring of FFR over the time-course of culture, which would be a useful way to assess future approaches of maturing cardiomyocytes in vitro.

Response: We are thankful to the reviewer for his valuable suggestion. We have included the suggested sentence in the discussion part of the revised version of the manuscript.

Page no 25: Revised version of the manuscript.

The capabilities of the crack-based cantilever sensor proposed were experimentally verified; it can be used for a long time in a solution owing to the PDMS protection layer while also maintaining a high sensitivity. In particular, the chemical bonding of the proposed encapsulation layer, performed to maintain long-term stability in the same environment as the culture medium, showed that it is a very stable encapsulation method that does not affect sensitivity after bonding. Characteristic evaluation of the PDMS-encapsulated sensor in various environments (humidity, temperature, and culture medium) showed the advantage of this crack sensor. However, the PDMS-encapsulated crack sensor is not completely impervious to environmental change. Further research is needed to improve the durability of the PDMS-encapsulated crack sensor integrated silicone rubber cantilever. **The proposed sensor can continuously monitor the FFR over the time-course of culture, which would be a useful way to assess future approaches of maturing cardiomyocytes in vitro.** It was possible to stably measure the contractile forces of cardiomyocytes for approximately four weeks in vitro. During the cultivation period, part of the cardiomyocytes was detached from the on the silicone rubber cantilever and not transmitting the resultant contractile force efficiently to the sensor. Therefore, the stable and long-term measurement of contractile force should be improved by reforming the cantilever design. The dose-response studies for Verapamil, Quinidine and Isoproterenol indicate that the toxicity of drugs can be screened in real-time using the proposed crack-based cantilever sensor with a PDMS encapsulation layer. The alignment of the cardiac tissue on the cantilever's longitudinal direction can induce a more significant displacement through the concentration of its contraction force. This can be expected to increase further the GF of the proposed crack sensor, which exhibits exponential behaviors in its resistance changes.

Comment-7: Pg 16 – the detachment of the cardiomyocytes over time may reflect higher static tension (this will also increase, along with contractile force). The detachment of the cells should be emphasized as a limitation of the current system and an opportunity for future improvements.

Response: As recommended by the reviewer, we have included the suggested sentence in the discussion part of the revised manuscript.

Page no 25: Revised version of the manuscript.

The capabilities of the crack-based cantilever sensor proposed were experimentally verified; it can be used for a long time in a solution owing to the PDMS protection layer while also maintaining a high sensitivity. In particular, the chemical bonding of the proposed encapsulation layer, performed to maintain long-term stability in the same environment as the culture medium, showed that it is a very stable encapsulation method that does not affect sensitivity after bonding. Characteristic evaluation of the PDMS-encapsulated sensor in various environments (humidity, temperature, and culture medium) showed the advantage of this crack sensor. However, the PDMS-encapsulated crack sensor is not completely impervious to environmental change. Further research is needed to improve the durability of the PDMS-encapsulated crack sensor integrated silicone rubber cantilever. The proposed sensor can continuously monitor the FFR over the time-course of culture, which would be a useful way to assess future approaches of maturing cardiomyocytes in vitro. It was possible to stably measure the contractile forces of cardiomyocytes for approximately four weeks in vitro. During the cultivation period, part of the cardiomyocytes was detached from the on the silicone rubber cantilever and not transmitting the resultant contractile force efficiently to the sensor. Therefore, the stable and long-term measurement of contractile force should be improved by reforming the cantilever design. The dose-response studies for Verapamil, Quinidine and Isoproterenol indicate that the toxicity of drugs can be screened in real-time using the proposed crack-based cantilever sensor with a PDMS encapsulation layer. The alignment of the cardiac tissue on the cantilever's longitudinal direction can induce a more significant displacement through the concentration of its contraction force. This can be expected to increase further the GF of the proposed crack sensor, which exhibits exponential behaviors in its resistance changes.

Comment-8: Figure 6 – the IC_{50} values for Verapamil should be calculated with standard deviation (this can be done by fitting curves to each technical or biological replicate to get individual IC_{50} , then calculating the mean and standard deviation). These should be reported on the figure, not just in the text.

Response: We have followed the protocol suggested by the reviewer and calculated the IC_{50} value for Verapamil, Quinidine, and Isoproterenol. The obtained IC_{50} value reported in Figure 6 and other relevant figures in the revised version of the manuscript.

Figure 6. Change in contractility of the Verapamil treated cultured cardiomyocytes. **a**, Representative real-time traces of change in sensor resistance ratio and cantilever displacement owing to the contraction and relaxation of different concentration of Verapamil treated cardiomyocytes. **b, c** Bar plot depicting the cantilever displacement and relative contraction force generated by cardiomyocytes (** $p < 0.01$ measures by one-way ANOVA followed by Tukey's honest significant difference test). **d**, Beat rate of cardiomyocytes at Verapamil concentrations of 0.01 nM and 1 μM . **e**, Beat rate of Verapamil treated cardiomyocytes at different concentrations (0.01 nM to 10 μM). **f**, Change in resistance of the cardiomyocytes seeded cantilever at different Verapamil concentrations (0.01 nM to 10 μM). **g**, Rise time and decay time of cardiomyocytes at Verapamil concentrations of 0.01 nM and 1 μM . The rise time and decay time were analyzed and beat rate corrected with Fridericia's formula (Rise time corrected (R_c) = $R/\text{interspike interval}^{1/3}$) and (Decay time corrected (D_c) = $D/\text{interspike interval}^{1/3}$), respectively. (** $p < 0.01$ measures by one-way ANOVA followed by Tukey's honest significant difference test). Error bars are mean \pm s.d. (n=5 biologically independent samples). NS and D are representing displacement and non-significant, respectively.

Figure 7. Change in contractility of the Quinidine treated cultured cardiomyocytes. **a**, Representative real-time traces of change in sensor resistance ratio and cantilever displacement owing to the contraction and relaxation of different concentration of Quinidine treated cardiomyocytes. **b, c** Bar plot depicting the cantilever displacement and relative contraction force generated by cardiomyocytes (** $p < 0.01$ measures by one-way ANOVA followed by Tukey's honest significant difference test.). **d**, Beat rate of cardiomyocytes at Quinidine concentrations of 1 nM and 10 μ M. **e**, Beat rate of Quinidine treated cardiomyocytes at different concentrations (1 nM to 100 μ M). **f**, Change in resistance of the cardiomyocytes seeded cantilever at different Quinidine concentrations (1 nM to 100 μ M). **g**, Rise time and decay time of cardiomyocytes at Quinidine concentrations of 1 nM and 10 μ M. The rise time and decay time were analyzed and beat rate corrected with Fridericia's formula (Rise time corrected (Rc) = $R/\text{interspike interval}^{1/3}$) and (Decay time corrected (Dc) = $D/\text{interspike interval}^{1/3}$), respectively. (* $p < 0.05$, ** $p < 0.01$ measures by one-way ANOVA followed by Tukey's honest significant difference test.). Error bars are mean \pm s.d. ($n=5$ biologically independent samples). D is representing displacement.

Supplementary Figure. 22. Effect of Isoproterenol on the cultured cardiomyocytes. **a**, Representative real-time traces of change in sensor resistance ratio and cantilever displacement owing to the contraction and relaxation of different concentrations of Isoproterenol treated cardiomyocytes. **b**, **c**, Bar plot depicting the cantilever displacement and relative contraction force generated by cardiomyocytes (** $p < 0.01$ measures by one-way ANOVA followed by Tukey's honest significant difference test.). **d**, Beat rate of cardiomyocytes at Isoproterenol concentrations of 1 nM and 1 μ M. **e**, Beat rate of Isoproterenol treated cardiomyocytes at different concentrations (1 nM to 2 μ M). **f**, Change in resistance of the cardiomyocytes seeded cantilever at different Isoproterenol concentrations (1 nM to 2 μ M). **g**, Rise time and decay time of cardiomyocytes at Isoproterenol concentrations of 1 nM and 1 μ M. The rise time and decay time were analyzed and beat rate corrected with Fridericia's formula (Rise time corrected (Rc) = $R/\text{interspike interval}^{1/3}$) and (Decay time corrected (Dc) = $D/\text{interspike interval}^{1/3}$), respectively. (** $p < 0.01$ measures by one-way ANOVA followed by Tukey's honest significant difference test.). Error bars are mean \pm s.d. ($n=5$ biologically independent samples). NS and D are representing displacement and non-significant, respectively.